# BOTS: A Unified Framework for Bayesian Online Task Selection in LLM Reinforcement Finetuning

**Qianli Shen**[*]  **Daoyuan Chen**[*]  **Yilun Huang**  **Zhenqing Ling**
**Yaliang Li**[†]  **Bolin Ding**[†]  **Jingren Zhou**

Alibaba Group

## Abstract

Reinforcement finetuning (RFT) is a key technique for aligning Large Language Models (LLMs) with human preferences and enhancing reasoning, yet its effectiveness is highly sensitive to which tasks are explored during training. Uniform task sampling is inefficient, wasting computation on tasks that are either trivial or unsolvable, while existing task selection methods often suffer from high rollout costs, poor adaptivity, or incomplete evidence. We introduce **BOTS**, a unified framework for **B**ayesian **O**nline **T**ask **S**election in LLM reinforcement finetuning. Grounded in Bayesian inference, BOTS adaptively maintains posterior estimates of task difficulty as the model evolves. It jointly incorporates *explicit evidence* from direct evaluations of selected tasks and *implicit evidence* inferred from these evaluations for unselected tasks, with Thompson sampling ensuring a principled balance between exploration and exploitation for task selection. To make implicit evidence practical, we instantiate it with an ultra-light interpolation-based plug-in that estimates difficulties of tasks without extra rollouts, adding negligible overhead. Empirically, across diverse domains and LLM scales, BOTS consistently improves data efficiency and performance over baselines and ablations, providing a practical and extensible solution for dynamic task selection in RFT[1].

## 1 Introduction

Reinforcement finetuning (RFT) has become a key technique for aligning Large Language Models (LLMs) with human preferences and enhancing their reasoning capabilities (Jaech et al., 2024; Guo et al., 2025; Luo et al., 2025; Hu et al., 2025; Zeng et al., 2025). However, the effectiveness of RFT is highly sensitive to task selection (Parashar et al., 2025; Shen et al., 2025; Zhu et al., 2025; Wen et al., 2025; Li et al., 2025a). Naively training on a static, uniformly sampled dataset is inefficient: the model spends excessive computation on tasks that are either already mastered (too easy) or beyond reach (too hard) (Yu et al., 2025; Bae et al., 2025; Chen et al., 2025b). This inefficiency not only inflates training costs but also destabilizes optimization by reducing the effective batch size. The central challenge, therefore, is to dynamically select tasks of "just right" difficulty to maximize learning efficiency as the model's capability evolves.

Existing methods to this challenge face several limitations. Offline task selection (Parashar et al., 2025; Shen et al., 2025; Zhu et al., 2025; Wen et al., 2025; Li et al., 2025a), which pre-schedules tasks from easy to hard, is too rigid and does not adapt to the evolving trajectory of the model. In response, a few online selection methods have been proposed, aiming to adaptively choose tasks based on model's current capability. Core challenge of these methods lies in the tradeoff between the computational cost of collecting information and the accuracy of the resulting performance estimates. We argue that existing solutions are not sufficiently efficient: some expend excessive computation on information gathering, undermining efficiency, while others fail to fully exploit collected information, leading to suboptimal selection. On one hand, oversampling-based methods (Yu et al., 2025; Bae et al., 2025) find suitable tasks by rolling out oversized batches, introducing substantial extra

---

[*]Equal contribution

[†]Corresponding Authors. Email to {shenqianli.sql, yaliang.li, bolin.ding}@alibaba-inc.com

[1]Our code is released at https://github.com/modelscope/Trinity-RFT/tree/main/examples/bots

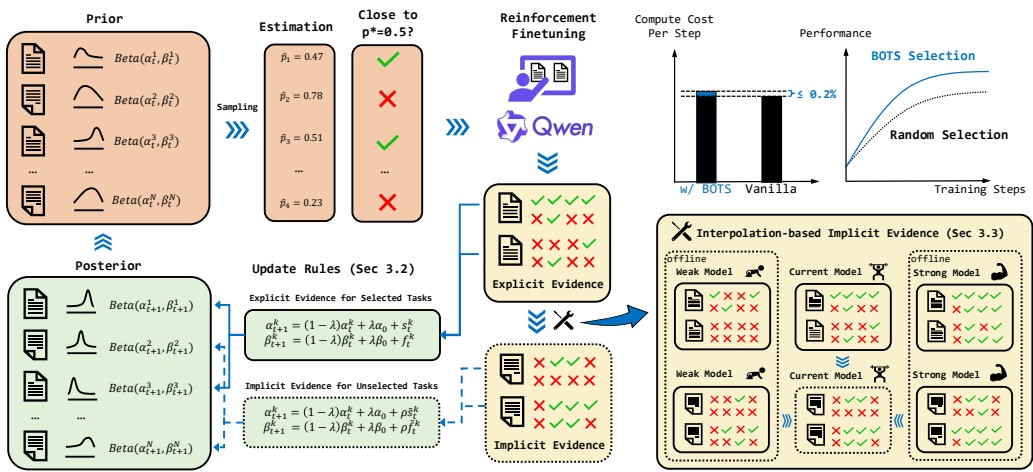

Figure 1: **Overview of the BOTS framework.** BOTS operates in a continuous loop of task selection, model training, and posterior updating. (1) **Selection:** Thompson sampling from the posterior beliefs selects a batch of tasks whose estimated success probabilities are near a target difficulty (e.g., $p^* = 0.5$). (2) **Training & Evidence Collection:** The LLM is finetuned, yielding direct success/failure counts (*explicit evidence*) for the selected batch. For unselected tasks, predicted counts (*implicit evidence*) are produced by a plug-in; in Section 3.3, we introduce an ultra-lightweight interpolation-based variant with negligible overhead. (3) **Posterior Updating:** Explicit and implicit evidence are fused using our generalized Bayesian update rule (Section 3.2). The BOTS framework and the Interpolation-based Implicit Evidence are summarized as Algorithm 1∼ 2 in Appendix C.

cost. On the other hand, non-oversampling approaches typically rely on a single source of information—either leveraging historical evaluations as *explicit evidence* (Chen et al., 2025b; Qu et al., 2025) or exploiting inter-task correlations as *implicit evidence* (Sun et al., 2025). Our empirical results reveal a clear complementarity: explicit evidence provides stable and accurate task-difficulty estimates but suffers from a slow warm-up when historical evaluations are scarce in early training, whereas implicit evidence quickly guides early-stage selection yet becomes less reliable in later stages. These findings indicate that relying solely on one type of evidence leaves information underutilized and leads to suboptimal task selection. Therefore, a principled framework to *fuse* these complementary evidence sources is essential for robust and efficient online task selection.

In this work, we introduce **BOTS**, the first unified and extensible framework for **B**ayesian **O**nline **T**ask **S**election in LLM reinforcement finetuning. BOTS recasts online task selection as a principled Bayesian inference problem over the model's evolving capabilities. By doing so, it naturally addresses the core challenges of non-stationarity and partial observability, featuring with three key design elements: (1) **Bayesian foundation**: Grounded in Bayesian inference, the framework naturally adapts to the evolving capability of the model, allowing task difficulty to be continuously re-estimated. (2) **Integration of two evidence sources**: Tunable update rules jointly incorporate *explicit evidence* from direct evaluations and *implicit evidence* inferred from related tasks, leveraging their complementary strengths. (3)**Thompson sampling**: Task selection is guided by posterior sampling, ensuring a principled balance between exploration and exploitation.

For implicit evidence, we further instantiate the framework with an extremely efficient interpolation-based plugin that estimates the difficulty of unevaluated tasks without additional rollouts, making the overhead negligible. We demonstrate empirically, across diverse domains and model scales, that our method significantly improves data efficiency and model performance over baselines and ablations, offering a practical, effective, and extensible solution for online task selection in RFT.

## 2 RELATED WORKS

The impact of task difficulty on RFT of LLMs has become an active research topic. Inspired by the seminal idea of curriculum learning (Bengio et al., 2009), researchers have proposed various strategies for selecting appropriate tasks in LLM RFT. For instance, Parashar et al. (2025); Shen

et al. (2025) advocate scheduling tasks from easy to hard, enabling LLMs to gradually acquire reasoning skills. Similar ideas have been extended to multi-modal LLMs (Zhu et al., 2025; Wen et al., 2025; Li et al., 2025a), though these methods mainly focus on offline task selection following an easy-to-hard trajectory.

More recently, online selection strategies have emerged, often targeting tasks of moderate difficulty. One line of work adopts *sampling-based task filtering*, where tasks with consistently trivial rewards (all zeros or ones) are considered uninformative and are down-weighted or filtered (Yu et al., 2025; Bae et al., 2025). While effective, these methods require additional rollouts, incurring non-trivial overhead. To avoid extra rollouts, several works attempt to *predict* task passing rates without direct rollouts. Chen et al. (2025b) formulate task selection as a non-stationary multi-armed bandit problem, treating each problem category (e.g., difficulty level or type) as an arm and using absolute advantage as a reward proxy, with posterior estimation based on historical outcomes. Qu et al. (2025) extend this framework to the task level. Although these methods eliminate extra rollouts, they rely solely on direct evaluations and overlook cross-task relationships. In contrast, Sun et al. (2025) propose evaluating a small set of reference tasks and predicting the passing rates of others using an attention-inspired kernel over embeddings. However, this approach still requires additional rollouts for the reference set and discards historical evaluation information.

More comprehensive discussion and comparison on related works are provided in Appendix E.

## 3 BAYESIAN ONLINE TASK SELECTION

### 3.1 PRELIMINARIES: MODELING TASK DIFFICULTY

A task $\mathcal{T} = (Q, R)$ is defined as a tuple consisting of a query $Q$, expressed in natural language, and a reward function $R$ that maps any natural language response $O$ to a binary reward $R(Q, O) \in \{0, 1\}$, which is common in domains like math, coding such that $1$ indicates correct and $0$ indicates incorrect. Consider RFT of a parameterized language model $\mathcal{M}_\theta$, which maps a query $Q$ to a response $O$, on a set of $N$ tasks $\{\mathcal{T}^k\}_{k=1}^N$. The binary reward obtained by executing the model on a task $\mathcal{T}$ follows a Bernoulli distribution $\texttt{Bernoulli}(p_{\theta, \mathcal{T}})$, where $p_{\theta, \mathcal{T}} = \mathbb{E}_{o \sim \mathcal{M}(\cdot|\mathcal{T}; \theta)} R(O, \mathcal{T})$ denotes the model's success probability on $\mathcal{T}$. With a slight abuse of notation, we denote the reward distribution for a given model and task as $R(\cdot|\mathcal{T}; \theta) := \texttt{Bernoulli}(p_{\theta, \mathcal{T}})$. Since we focus on online task selection over a fixed set of tasks, we simplify the notation by letting $p_t^k$ denote $p_{\theta_t, \mathcal{T}^k}$ and $R_t^k$ denote $R(\cdot|\mathcal{T}^k; \theta_t)$, whenever the context is clear. All notations are summarized in Table 5.

### 3.2 CORE MECHANISM: FUSING EVIDENCE IN A UNIFIED POSTERIOR

Our goal is to estimate the success probability $p_t^k$ of the online-adapted model on $k$-th task $\mathcal{T}^k$. For efficiency, direct evaluations are only performed after a task is selected. As statistical evidence, at time step $t$, we obtain online samples $r_{1:n}^k \overset{\text{i.i.d.}}{\sim} R_t^k(\cdot)$ for each selected task $\mathcal{T}^k$ in the training batch $\mathcal{B}_t$, where $n$ corresponds to the number of rollouts per task.

A natural way to model the estimation is via a Beta distribution, $\texttt{Beta}(\alpha_t^k, \beta_t^k)$, where the posterior parameters $\alpha_t^k$ and $\beta_t^k$ represent the accumulated counts of successes and failures, respectively, for model $\theta_t$ on task $\mathcal{T}^k$. The problem then reduces to designing online adaptation rules for $\alpha_t^k$ and $\beta_t^k$. We propose the following online adaptation rules: Given a batch of direct evaluation results $\mathcal{B}_t = \{(\mathcal{T}_{\mathcal{B}_t[i]}, r_{1:n}^{\mathcal{B}_t[i]})\}_{i=1}^{|\mathcal{B}_t|}$, we define the adaptation rules as

$$\alpha_{t+1}^k = (1-\lambda)\alpha_t^k + \lambda\alpha_0^k + (1-\rho)\, s_t^k + \rho\, \tilde{s}_t^k, \qquad \beta_{t+1}^k = (1-\lambda)\beta_t^k + \lambda\beta_0^k + (1-\rho)\, f_t^k + \rho\, \tilde{f}_t^k, \quad (1)$$

where $\alpha_0, \beta_0$ denote the prior parameter set for the Beta distribution, and the coefficient $\lambda \in [0, 1]$ discounts historical information by interpolating the counts with the prior (Raj & Kalyani, 2017);

$$s_t^k = \sum_{k' \in \mathcal{B}_t} \mathbb{I}[k' = k] \sum_{i=1}^n r_i^{k'}, \qquad f_t^k = \sum_{k' \in \mathcal{B}_t} \mathbb{I}[k' = k] \sum_{i=1}^n (1 - r_i^{k'}) \qquad (2)$$

denote the *explicit* success and failure counts from direct evaluations, by slightly abusing the notation $k \in \mathcal{B}_t$ to represent task $\mathcal{T}^k$ received direct evaluation at time step $t$. Notice when direct evaluation

results are not available ($k \notin \mathcal{B}_t$), $s_t^k = f_t^k = 0$; and

$$\tilde{s}_t^k = s_t^k + \mathbb{I}[k \notin \mathcal{B}_t] \, \tilde{p}(k, \mathcal{B}_t) \, n, \qquad \tilde{f}_t^k = f_t^k + \mathbb{I}[k \notin \mathcal{B}_t] \, (1 - \tilde{p}(k, \mathcal{B}_t)) \, n. \qquad (3)$$

Here, $\rho \in [0, 1]$ balances the contributions of explicit and implicit evidence, $\tilde{s}_t^k$ and $\tilde{f}_t^k$ coincide with $s_t^k$ and $f_t^k$ when direct evaluation results are available for task $\mathcal{T}_k$, and otherwise represent the *pseudo* success and failure counts. These are derived from an estimator $\tilde{p}(k, \mathcal{B}_t)$, which uses inter-task relationships to infer difficulty for tasks *not present* in the current evaluation batch $\mathcal{B}_t$. Our framework places no restrictions on the specific form of $\tilde{p}(k, \mathcal{B}_t)$, while in Sec. 3.3, we introduce a lightweight interpolation-based instance to produce the pseudo counts. Additionally, to manage the equivalent total sample size—and hence the uncertainty of the estimate—the pseudo sample size is ensured to satisfy $\tilde{s}_t^k + \tilde{f}_t^k = n$.

The following proposition indicates that the update in Equation (1) preserves the Beta family as the (generalized) posterior for a Bernoulli parameter under a tempered/prior-mixing update.

**Proposition 1.** *Let $p \in (0, 1)$ be the Bernoulli success probability at time $t$. Suppose the current belief is $\pi_t(p) = \text{Beta}(p \mid \alpha_t, \beta_t)$, and let $\pi_0(p) = \text{Beta}(p \mid \alpha_0, \beta_0)$ be a base prior. Given counts $(s_t, f_t)$ and pseudo counts $(\tilde{s}_t, \tilde{f}_t)$ with $s_t, f_t, \tilde{s}_t, \tilde{f}_t \geq 0$, define the generalized-Bayes update*

$$\pi_{t+1}(p) \; \propto \; \underbrace{\pi_t(p)^{1-\lambda} \pi_0(p)^{\lambda}}_{\text{prior mixing / discounting}} \times \; \underbrace{\left[ p^{s_t}(1-p)^{f_t} \right]^{1-\rho}}_{\text{tempered explicit likelihood}} \times \; \underbrace{\left[ p^{\tilde{s}_t}(1-p)^{\tilde{f}_t} \right]^{\rho}}_{\text{tempered implicit evidence}}, \qquad (4)$$

*with $\lambda \in (0, 1)$ and $\rho \in [0, 1]$. Then $\pi_{t+1}$ is exactly $\text{Beta}(\alpha_{t+1}, \beta_{t+1})$ with*

$$\alpha_{t+1} = (1 - \lambda)\alpha_t + \lambda\alpha_0 + (1 - \rho)s_t + \rho\tilde{s}_t, \qquad \beta_{t+1} = (1 - \lambda)\beta_t + \lambda\beta_0 + (1 - \rho)f_t + \rho\tilde{f}_t.$$

The proof is placed in Appendix B.1.

### 3.3 Ultra-Light Interpolation Plug-in for Implicit Evidence

Given a batch of online evaluation results $\mathcal{B}_t = \{(\mathcal{T}_{\mathcal{B}_t[i]}, r_{1:n}^{\mathcal{B}_t[i]})\}_{i=1}^{|\mathcal{B}_t|}$, we aim to estimate the passing rate $p_t^k$ for any task $\mathcal{T}_k$ using an estimator $\tilde{p}(\mathcal{B}_t, k)$. In this work, we adopt an ultra-lightweight interpolation-based estimator to minimize additional computational overhead for online task selection. Notably, the adaptation rules in Sec. 3.2 place no restrictions on the specific form of $\tilde{p}(\mathcal{B}_t, k)$.

Assume that for each task $\mathcal{T}^k$, we have empirical success rates $\bar{p}_w^k$ and $\bar{p}_s^k$ from two reference models of distinct capability (weak vs. strong). Define the average empirical success rates of the current, weak, and strong models on $\mathcal{B}_t$ as

$$\bar{p}_t^{\text{ref}}(\mathcal{B}_t) := \frac{1}{|\mathcal{B}_t|} \sum_{k \in \mathcal{B}_t} \frac{1}{n} \sum_{j=1}^n r_j^k, \quad \bar{p}_w^{\text{ref}}(\mathcal{B}_t) := \frac{1}{|\mathcal{B}_t|} \sum_{k \in \mathcal{B}_t} \bar{p}_w^k, \quad \bar{p}_s^{\text{ref}}(\mathcal{B}_t) := \frac{1}{|\mathcal{B}_t|} \sum_{k \in \mathcal{B}_t} \bar{p}_s^k.$$

We estimate the relative capability coefficient of the current model as $\mu_t(\mathcal{B}_t) = \left( \bar{p}_t^{\text{ref}}(\mathcal{B}_t) - \bar{p}_w^{\text{ref}}(\mathcal{B}_t) \right) / \left( \bar{p}_s^{\text{ref}}(\mathcal{B}_t) - \bar{p}_w^{\text{ref}}(\mathcal{B}_t) \right)$, which locates the current model between the weak and strong reference models on the batch $\mathcal{B}_t$ (we assume $\bar{p}_s^{\text{ref}} > \bar{p}_w^{\text{ref}}$; otherwise one may add a small $\varepsilon$ to the denominator). To reduce variance from stochastic rollouts, we maintain a momentum version of the coefficient $\tilde{\mu}_t = \gamma\tilde{\mu}_{t-1} + (1 - \gamma)\mu_t$.

Finally, the passing rate of the current model on task $\mathcal{T}^k$ is obtained via linear interpolation between the weak and strong references, followed by clipping to $[0, 1]$:

$$\tilde{p}(k, \mathcal{B}_t) = \text{clip}\left( \tilde{\mu}_t(\mathcal{B}_t) \, \bar{p}_s^k + \left( 1 - \tilde{\mu}_t(\mathcal{B}_t) \right) \bar{p}_w^k, \, 0, \, 1 \right). \qquad (5)$$

### 3.4 Thompson Sampling for Task Selection

Having established a task difficulty posterior $\text{Beta}(p_t^k \mid \alpha_t^k, \beta_t^k)$ over the success probability of each task $\mathcal{T}^k$, we now turn to the crucial step of selecting tasks for the next training batch.

The first question is: at what difficulty level does the current model benefit most from training? Prior works (Chen et al., 2025b; Sun et al., 2025) show that, under binary rewards, tasks with success probability around $0.5$ are most informative for learning as they lead to gradients with larger

expected magnitude than tasks with success probability close to $0$ or $1$. We define the utility of a task as the absolute deviation of its posterior mean from a target success probability $p^* \in (0, 1)$, with $p^* = 0.5$ as the canonical choice.

Given this target success probability, the problem of online task selection naturally reduces to a non-stationary bandit problem. The central challenge is the tradeoff between *exploitation* and *exploration* for task selection: the model must decide whether to select tasks with high-confidence estimates close to the target rate in order to maximize immediate utility, or to select tasks with high uncertainty to gather information that may improve future decisions. A purely exploitative strategy might select tasks whose posterior mean $\tilde{p}_t^k$ is closest to $p^*$, but this risks overlooking tasks whose difficulty is currently uncertain but potentially optimal. To naturally balance exploration and exploitation, we employ *Thompson Sampling* (Thompson, 1933), a strategy renowned for both its empirical effectiveness and theoretical guarantees. More specifically, at each selection step $t$, we perform the following: (1) **Posterior Sampling:** Draw a sample of the passing rate from its current posterior distribution $\hat{p}_k \sim \text{Beta}(\alpha_t^k, \beta_t^k)$ for each task $\mathcal{T}^k$ in the pool. (2) **Selection:** Select tasks with the highest estimated utilities $\{\hat{u}_k := -|\hat{p}_k - p^*|\}_{k=1}^N$ to form the training batch $\mathcal{B}_{t+1}$. This procedure elegantly prioritizes tasks likely to be near the target difficulty, while the inherent variance in sampling from the posterior ensures that tasks with higher uncertainty are naturally explored.

## 3.5 Hyperparameters

The hyperparameters $\lambda$ and $\rho$ play distinct but complementary roles in shaping both the *difficulty estimation* process and the *uncertainty profile* that drives Thompson Sampling–based task selection. Their effects can be summarized along two axes:

**Impact on difficulty estimation.** The parameter $\lambda$ controls the balance between adaptivity and stability in Bayesian belief updating. A *larger* $\lambda$ places more weight on recent evaluations, enabling the posterior to rapidly adapt to the model's evolving capability. However, this increased adaptivity also makes the estimate more sensitive to noise in the most recent observations, reducing stability. Conversely, a *smaller* $\lambda$ assigns greater weight to accumulated history, reducing variance in the estimated probability of success. This leads to more stable but less responsive difficulty estimates. A toy example illustrating these effects is provided in Figure 4.

The parameter $\rho$ governs how explicit and implicit evidence are fused. A *larger* $\rho$ biases the update toward implicit evidence, causing tasks without direct evaluations to inherit stronger difficulty signals inferred from tasks with direct evaluations. A *smaller* $\rho$, in contrast, favors explicit evidence, making the posterior rely primarily on direct observations and thus behave more conservatively when a task has not been recently evaluated.

**Impact on selection-time uncertainty.** Proposition 2 characterizes how the hyperparameters $\lambda$ and $\rho$ jointly determine the effective sample size $n_t = \alpha_t + \beta_t$ of each task's Beta posterior, thereby controlling the confidence of the estimated probability of success.

**Proposition 2.** *Let $n_t := \alpha_t + \beta_t$. Suppose the updates follow Equation (1)–(3) with $\lambda \in (0, 1)$, $\rho \in [0, 1]$, we have*

$$\liminf_{t \to \infty} n_t = n_0 + \tfrac{\rho}{\lambda} n, \qquad \limsup_{t \to \infty} n_t = n_0 + \tfrac{1}{\lambda} n.$$

The proof is given in Appendix B.2. In terms of balancing exploration toward tasks whose current success probability estimates deviate from the target and exploitation of tasks estimated to be near the target success probability, $\lambda$ controls the overall scale of the effective sample size $n_t = \alpha_t + \beta_t$, which directly dictates the variance of the posterior distribution. A *larger* $\lambda$ accelerates discounting of older evidence, reducing $n_t$ and increasing posterior uncertainty. Under Thompson Sampling, this results in more exploratory behavior, particularly toward tasks whose current success-probability estimates deviate from the target difficulty. A *smaller* $\lambda$ enlarges $n_t$, decreasing posterior variance and nudging the strategy toward exploitation of tasks estimated to be near the target success probability.

The parameter $\rho$ primarily affects the effective sample sizes of tasks that *have not* been directly evaluated at the current step. A *larger* $\rho$ increases these tasks' effective sample sizes via stronger pseudo-count updates, reducing their posterior uncertainty and making them less likely to be explored. A *smaller* $\rho$ decreases their effective sample sizes, increasing posterior variance and allow-

ing tasks with difficulty estimates farther from the target to be selected, thereby improving long-run exploration.

## 3.6 PRACTICAL CONSIDERATIONS

**Hyperparameters.** We recommend using $\lambda = 0.1$ and $\rho = 0.1$ as the default configuration for BOTS. This setting consistently performs well across all model scales and task domains evaluated in Section 4. For new applications, practitioners may start from this default and subsequently adjust the hyperparameters using the insights provided in Section 3.5.

**Computational Cost of Interpolation-based Task Difficulty Estimation.** The main cost of the interpolation-based estimator (Section 3.3) comes from evaluating reference models on all tasks. Fortunately, such evaluations are increasingly common in modern RL datasets—for example, the GURU dataset (Cheng et al., 2025) provides Qwen2.5-7B-Instruct (Yang et al., 2024) and Qwen3-30B-A3B (Yang et al., 2025) scores as difficulty tags for offline filtering. Our estimator simply reuses these tags for *online* task selection. In settings where reference-model evaluations are not available, enabling implicit evidence requires a one-time rollout cost to compute them. Given the substantial gains that implicit evidence brings, this introduces a natural computation–efficiency tradeoff. Unlike oversampling-based online methods, this cost does not recur during training and amortizes well when training multiple models or checkpoints. Additional practical benefits of this computation are discussed in Appendix F.3. Once the reference scores are available, the *online* overhead of the interpolation-based estimator is negligible (see Appendix G.1), since no extra rollouts are required and all updates reduce to simple vector operations.

**Choice of Reference Models.** When a dataset provides pre-computed base model evaluation tags, as in GURU, we recommend directly using these tags as reference-model signals. This avoids any additional rollout cost, even though it may introduce extrapolation (e.g., when both reference models are stronger or weaker than the training model). Our empirical results in Appendix G.2 show that while extrapolation can reduce the accuracy of implicit evidence, BOTS remains robust, making the computational savings generally worthwhile. When multiple reference-model pairs are available, we suggest choosing models with a *clear capability gap*, which provides stronger discrimination for interpolating task difficulty. For instance, GURU's choice of Qwen2.5-7B-Instruct and Qwen3-30B-A3B yields informative difficulty signals across a wide range. Finally, although the interpolation-based estimator supports extrapolation in principle, our results show that avoiding extrapolation produces more accurate implicit evidence and slightly stronger overall BOTS performance.

## 4 EXPERIMENTS

We begin with Section 4.1, which introduces datasets, reinforcement finetuning protocols, evaluation metrics, and computational cost. Section 4.2 and Section 4.3 analyze the impact of hyperparameters $(\rho)$, $(\lambda)$ on task selection respectively. Section 4.4 then compares BOTS with competitive baselines across model scales and domains, while Section 4.5 summarizes additional experiments provided in the Appendix G.

### 4.1 SETUPS

**Dataset.** We conduct experiments on GURU (Cheng et al., 2025), a well-curated cross-domain RL dataset. Each subset is deduplicated, verified, and filtered. We use its math, code, and logic subsets (excluding the Zebra Puzzle due to its non-binary reward). Detailed information about the used datasets is provided in Appendix D.1 (for training) and Appendix D.2 (for evaluation).

**RFT Setting.** We adopt GRPO (Shao et al., 2024), Qwen2.5-1.5B-Instruct and Qwen2.5-7B. Key hyperparameters include a learning rate of 1e-6, 16 rollouts per task, and a temperature of $1.0$. Comprehensive training details and used RL algorithm are provided in Appendix D.3 and D.5.

**Evaluation.** We report the following metrics to evaluate our framework and its ablations, with formal definitions given in Appendix D.4.

• *Effective Task Ratio (ETR).* It evaluates tasks selection. The fraction of sampled tasks whose *empirical success rate*, estimated from $n = 16$ independent rollouts, falls strictly within the $(0, 1)$ range. A higher ETR indicates a more efficient task selection that successfully filters out tasks that are either already mastered ($p = 1$) or currently unsolvable ($p = 0$).

• *Time-to-Baseline (TTB).* It measures training acceleration relative to the random baseline to achieve a specific performance. Let the baseline start from performance $P_{\text{init}}$ and reach the best performance $P_{\text{best}}$ within the training window. For a target fraction $\tau \in \{50\%, 75\%, 100\%\}$, we define the target performance as $P_{\tau} = P_{\text{init}} + \tau \cdot (P_{\text{best}} - P_{\text{init}})$. $\text{TTB}(\tau)$ is the ratio of steps required by a method to reach $P_{\tau}$ compared to the baseline. For example, if the baseline starts at $0.1$ and reaches $0.3$ by step 100, then the 50% target is $0.2$. If the baseline first reaches $0.2$ at step 40 while another method reaches it at step 30, then $\text{TTB}(50\%) = 30/40 = 0.75$. By definition, the baseline has $\text{TTB} = 1$; smaller values indicate greater acceleration.

• *Best-so-far (BSF).* It measures the performance gain relative to the random baseline under a fixed budget. Within a specific ratio (25%, 50%, 100%) of total training steps, BSF is the ratio between a method's best-so-far performance and the baseline's best-so-far performance. For example, at step 50 (total steps 100), if the baseline's best is 0.4 and a method's best is 0.6, then BSF(50%) = $0.6/0.4 = 1.5$. The baseline always has BSF = 1; larger values indicate greater gains.

**Computational Overhead.** We note that BOTS introduces negligible train-time additional computation, with overhead measured at $\leq 0.2\%$ of total training time (see analysis in Section 3.3 and empirical results in Appendix G.1). Thus, in the main results, we report TTB and BSF, in terms of training steps rather than wall-clock time, as their difference is practically insignificant.

## 4.2 ANALYZING THE IMPACT OF $\rho$

In Section 3.5, we analyzed the role of the hyperparameter $\lambda$ and its influence on BOTS. In this section, we empirically validate these analyses. We investigate the role of $\rho$ in task selection by varying $\rho \in \{0.0, 0.05, 0.1, 0.2, 0.5, 1.0\}$, while keeping other hyperparameters fixed at their default values ($\lambda = 0.1$, posterior sampling enabled).

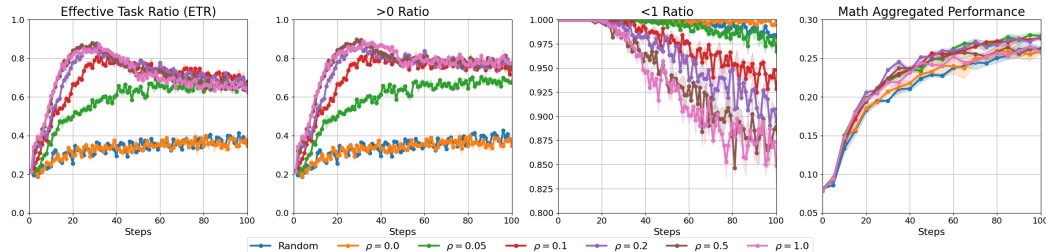

Figure 2: **Qwen2.5-1.5B-Instruct** on **Math** with $\lambda = 0.1, \rho \in \{0.0, 0.05, 0.1, 0.2, 0.5, 1.0\}$. Ratio of sampled training tasks (measured over 16 rollouts) with passing rates: strictly between 0 and 1 (L1), strictly greater than 0 (L2), and strictly less than 1 (L3), along with aggregated performance (MATH500, AMC23 and AIME24), plotted against training steps. All results are averaged over 3 random seeds, and 95% confidence intervals are reported. More detailed results are placed in Appendix G.6.

**Analysis and Takeaways.** Our central finding is that *implicit evidence is crucial for rapid cold-starts, while explicit evidence is vital for long-term accuracy. A principled fusion of both is key to effective task selection.* Relying solely on implicit evidence ($\rho = 1$) leads to error accumulation, while ignoring it ($\rho = 0$) suffers from severe data sparsity in early training, especially on large-scale datasets. This conclusion is supported by the following observations from Figure 2 and Table 1:

(1) **Implicit evidence provides an essential early boost.** In the initial training phase, all settings with $\rho > 0$ demonstrate a sharp increase in the Effective Task Ratio (ETR) over the random baseline, primarily by filtering out unsolvable tasks ($p = 0$). In contrast, the $\rho = 0$ setting, which relies solely on sparse explicit feedback, behaves almost identically to random sampling. This highlights the critical role of implicit evidence in overcoming the cold-start problem.

| Benchmark | MATH500 | | | | | | AMC23 | | | | | | AIME24 | | | | | | Math Aggregated Performance | | | | | |
|---|---|---|---|---|---|---|---|---|---|---|---|---|---|---|---|---|---|---|---|---|---|---|---|---|
| Metric | TTB (↓) | | | BSF (↑) | | | TTB (↓) | | | BSF (↑) | | | TTB (↓) | | | BSF (↑) | | | TTB (↓) | | | BSF (↑) | | |
| Target Fraction | 50% | 75% | 100% | 25% | 50% | 100% | 50% | 75% | 100% | 25% | 50% | 100% | 50% | 75% | 100% | 25% | 50% | 100% | 50% | 75% | 100% | 25% | 50% | 100% |
| Random | 1.00 | 1.00 | 1.00 | 1.00 | 1.00 | 1.00 | 1.00 | 1.00 | 1.00 | 1.00 | 1.00 | 1.00 | 1.00 | 1.00 | 1.00 | 1.00 | 1.00 | 1.00 | 1.00 | 1.00 | 1.00 | 1.00 | 1.00 | 1.00 |
| $\rho = 0.0$ | 0.95 | 0.88 | 1.04 | 1.00 | 1.01 | 1.01 | 1.13 | 0.98 | - | 0.98 | 0.99 | 0.98 | 1.16 | 0.78 | 0.76 | 0.93 | 1.18 | 1.13 | 0.98 | 0.94 | - | 0.99 | 1.02 | 0.98 |
| $\rho = 0.05$ | 0.77 | 0.66 | **0.66** | 1.05 | 1.08 | 1.04 | 0.86 | 0.78 | 0.72 | 1.11 | 1.08 | **1.12** | **0.66** | 0.77 | 0.61 | **1.20** | 1.05 | **1.32** | 0.78 | 0.68 | **0.64** | 1.06 | 1.09 | **1.06** |
| $\rho = 0.1$ | 0.86 | 0.66 | 0.78 | 1.05 | **1.09** | **1.06** | 0.86 | 0.68 | 0.74 | 1.10 | 1.16 | 1.06 | 0.86 | 0.85 | 0.50 | **1.20** | 1.25 | 1.23 | 0.85 | 0.66 | 0.72 | 1.06 | **1.12** | 1.05 |
| $\rho = 0.2$ | **0.76** | **0.61** | 0.74 | **1.08** | 1.08 | 1.04 | 0.87 | 0.62 | **0.70** | 1.05 | **1.20** | 1.04 | 1.00 | 0.71 | **0.47** | 1.02 | **1.63** | 1.31 | 0.78 | 0.63 | 0.69 | 1.07 | 1.10 | 1.05 |
| $\rho = 0.5$ | 0.82 | 0.68 | 0.98 | 1.07 | 1.04 | 1.01 | **0.78** | **0.61** | - | 1.14 | 1.16 | 1.00 | 0.70 | 0.80 | 0.67 | 1.16 | 1.24 | 1.16 | 0.79 | 0.64 | 0.89 | **1.09** | 1.09 | 1.00 |
| $\rho = 1.0$ | 0.78 | 0.70 | - | 1.06 | 1.04 | 0.98 | 0.85 | 0.86 | - | 1.04 | 1.13 | 1.00 | 0.96 | **0.66** | 0.48 | 1.02 | 1.43 | 1.22 | 0.78 | 0.69 | 0.99 | 1.05 | 1.07 | 1.01 |

Table 1: **Qwen2.5-1.5B-Instruct** on **Math** with $\lambda = 0.1, \rho \in \{0.0, 0.05, 0.1, 0.2, 0.5, 1.0\}$. For TTB, notation "-" indicates that the target performance is never achieved within the evaluation window. The **best** and second best results are marked accordingly.

(2) **Over-reliance on implicit evidence degrades long-term performance.** As training progresses, settings with a large $\rho$ (e.g., 0.5, 1.0) show a declining ETR. This is driven by an increased tendency to select tasks whose success probability is 1, as shown in the curve of $< 1$ ratio. Consequently, the performance is negatively affected, as seen in the aggregated TTB and BSF metrics where large $\rho$ values underperform.

(3) **A small positive $\rho$ achieves the best of both worlds.** A $\rho \in \{0.05, 0.1, 0.2\}$ strikes an optimal balance. It leverages implicit evidence for initial acceleration while allowing more accurate, accumulating explicit evidence to dominate in the long run. This fusion leads to robust performance gains throughout entire training process, achieving best overall TTB and BSF scores.

## 4.3 ANALYZING THE IMPACT OF $\lambda$

In Section 3.5, we analyzed the role of the hyperparameter $\rho$ and its influence on BOTS. In this section, we empirically validate these analyses. We investigate the role of $\lambda$ in task selection by varying $\lambda \in \{0.0, 0.05, 0.1, 0.2, 0.5, 1.0\}$, while keeping other hyperparameters fixed at their default values ($\rho = 0.1$, posterior sampling enabled).

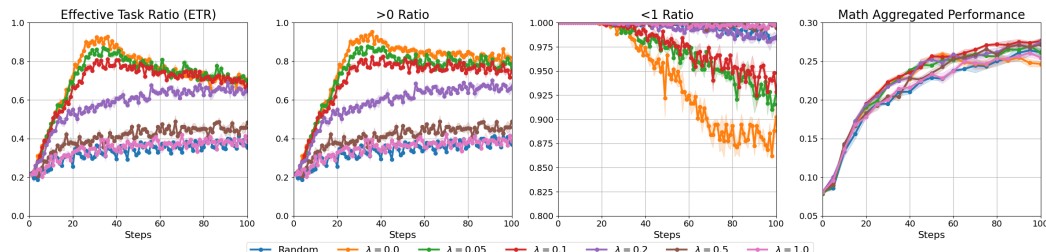

Figure 3: **Qwen2.5-1.5B-Instruct** on **Math** with $\lambda \in \{0.0, 0.05, 0.1, 0.2, 0.5, 1.0\}, \rho = 0.1$. Ratio of sampled training tasks (measured over 16 rollouts) with passing rates: strictly between 0 and 1 (L1), strictly greater than 0 (L2), and strictly less than 1 (L3), along with aggregated performance (MATH500, AMC23, and AIME24), plotted against training steps. All results are averaged over 3 random seeds, and 95% confidence intervals are reported. More detailed results are placed in Appendix G.7.

**Analysis and Takeaways.** The hyperparameter $\lambda$ governs (i) the adaptivity of difficulty estimation under a non-stationary learning process, and (ii) the exploration–exploitation balance during task selection. Our key takeaway is that *both of these aspects are crucial for strong long-term performance.* An overly small $\lambda$ prevents the system from adapting to the model's improving capability, whereas an overly large $\lambda$ injects excessive randomness into task selection. This conclusion is supported by the following observations from Figure 3 and Table 2:

(1) **Small $\lambda$ fails to track capability improvement.** Settings with small $\lambda$ (e.g., 0.0, 0.05) exhibit a declining, especially $< 1$ Ratio, in the mid-to-late training phase. This is because their preference to historical evaluations makes them slow to update the difficulty of tasks, causing ineffective tasks to be wastefully selected. This ultimately limits late-stage performance gains.

| Benchmark | MATH500 | | | | | | AMC23 | | | | | | AIME24 | | | | | | Math Aggregated Performance | | | | | |
|---|---|---|---|---|---|---|---|---|---|---|---|---|---|---|---|---|---|---|---|---|---|---|---|---|
| Metric | TTB (↓) | | | BSF (↑) | | | TTB (↓) | | | BSF (↑) | | | TTB (↓) | | | BSF (↑) | | | TTB (↓) | | | BSF (↑) | | |
| Target Fraction | 50% | 75% | 100% | 25% | 50% | 100% | 50% | 75% | 100% | 25% | 50% | 100% | 50% | 75% | 100% | 25% | 50% | 100% | 50% | 75% | 100% | 25% | 50% | 100% |
| Random | 1.00 | 1.00 | 1.00 | 1.00 | 1.00 | 1.00 | 1.00 | 1.00 | 1.00 | 1.00 | 1.00 | 1.00 | 1.00 | 1.00 | 1.00 | 1.00 | 1.00 | 1.00 | 1.00 | 1.00 | 1.00 | 1.00 | 1.00 | 1.00 |
| λ = 0.0 | 0.86 | 0.69 | - | **1.06** | 1.07 | 0.97 | 0.91 | **0.64** | **0.60** | 1.06 | **1.19** | 1.01 | 0.77 | **0.55** | 0.54 | 1.43 | 1.17 | **1.27** | 0.86 | 0.66 | - | 1.07 | 1.11 | 0.98 |
| λ = 0.05 | 0.86 | 0.67 | 0.96 | 1.05 | 1.04 | 1.00 | **0.82** | 0.74 | 1.04 | 1.02 | 1.11 | 1.01 | 1.04 | 0.67 | 0.53 | 0.95 | 1.23 | 1.26 | **0.83** | 0.70 | 0.89 | 1.02 | 1.07 | 1.02 |
| λ = 0.1 | 0.86 | **0.66** | 0.78 | 1.05 | 1.09 | 1.06 | 0.86 | 0.68 | 0.74 | 1.10 | 1.16 | 1.06 | 0.86 | 0.85 | 0.50 | 1.20 | 1.25 | 1.23 | 0.85 | **0.66** | 0.72 | 1.06 | **1.12** | 1.05 |
| λ = 0.2 | **0.83** | 0.76 | 0.92 | 1.05 | 1.07 | 1.01 | 0.90 | 0.69 | 0.66 | 1.04 | 1.13 | 1.05 | 0.83 | 0.89 | **0.47** | **1.27** | **1.28** | 1.21 | 0.83 | 0.71 | 0.86 | 1.05 | 1.10 | 1.04 |
| λ = 0.5 | 0.93 | 0.84 | 0.97 | 0.97 | 1.04 | 1.01 | 1.01 | 1.01 | 0.81 | **1.06** | 0.99 | **1.06** | **0.56** | 0.90 | 0.60 | 1.05 | 1.07 | 1.21 | 0.93 | 0.88 | 0.78 | 0.99 | 1.03 | 1.03 |
| λ = 1.0 | 0.91 | 0.81 | - | 0.99 | 1.02 | 0.97 | 1.18 | 0.96 | - | 1.01 | 0.99 | 1.00 | 0.91 | 1.18 | 0.79 | 1.10 | 0.88 | 1.08 | 0.96 | 0.96 | - | 1.00 | 1.02 | 0.99 |

Table 2: **Qwen2.5-1.5B-Instruct** on **Math** with $\lambda \in \{0.0, 0.05, 0.1, 0.2, 0.5, 1.0\}, \rho = 0.1$. For TTB, notation "-" indicates that the target performance is never achieved within the evaluation window. The **best** and second best results are marked accordingly.

(2) **Large $\lambda$ induces near-random task selection.** Conversely, large $\lambda$ produce task selection behavior that is almost fully random. This arises because large $\lambda$ aggressively discounts historical evidence, yielding very small effective sample sizes and thus high posterior uncertainty, which in turn drives overly exploratory Thompson sampling behavior.

(3) **Moderate $\lambda$ achieves the best trade-off.** A moderate $\lambda \in \{0.05, 0.1, 0.2\}$ strikes a balance: the difficulty estimates remain sufficiently adaptive while the uncertainty stays controlled, enabling a healthy exploration–exploitation balance. This leads to strong and consistent performance gains across the entire training process, achieving the best TTB and BSF results.

## 4.4 PERFORMANCE COMPARISON ACROSS MODELS AND DOMAINS

| Domain | Math | | | | | | Code | | | | | | Logic | | | | | |
|---|---|---|---|---|---|---|---|---|---|---|---|---|---|---|---|---|---|---|
| Metric | TTB (↓) | | | BSF (↑) | | | TTB (↓) | | | BSF (↑) | | | TTB (↓) | | | BSF (↑) | | |
| Target Fraction | 50% | 75% | 100% | 25% | 50% | 100% | 50% | 75% | 100% | 25% | 50% | 100% | 50% | 75% | 100% | 25% | 50% | 100% |
| Random | 1.00 | 1.00 | 1.00 | 1.00 | 1.00 | 1.00 | 1.00 | 1.00 | 1.00 | 1.00 | 1.00 | 1.00 | 1.00 | 1.00 | 1.00 | 1.00 | 1.00 | 1.00 |
| Offline | 0.77 | **0.66** | 0.85 | **1.09** | 1.11 | 1.04 | 0.68 | 1.03 | - | 1.09 | 1.00 | 0.99 | 1.23 | 1.36 | - | 0.67 | 0.78 | 0.89 |
| BOTS-MoPPS | 1.02 | 0.90 | 0.89 | 0.98 | 1.03 | 1.00 | 0.78 | 0.96 | 1.09 | 1.09 | 1.02 | 1.02 | 0.87 | 1.04 | - | 1.13 | 0.96 | 0.96 |
| BOTS-DOTS | **0.70** | 0.70 | 0.73 | 1.08 | 1.08 | 1.01 | 0.67 | **0.65** | 0.79 | 1.17 | **1.09** | 1.02 | 0.66 | 1.32 | - | **1.28** | 0.92 | 0.93 |
| **BOTS** | 0.85 | 0.66 | **0.72** | 1.06 | **1.12** | **1.05** | **0.58** | 0.90 | **0.77** | 1.17 | 1.08 | **1.03** | 0.85 | **0.94** | 1.05 | 1.19 | **1.01** | **1.00** |

Table 3: **BOTS-Qwen2.5-1.5B-Instruct Across Domains.** The recommended setting outperforms both out-of-framework and within-framework baselines, achieving 10 **first**-place and 6 second-place finishes out of 18 reported metrics. Full results are in Appendix G.8∼G.10.

| Domain | Math | | | | | | Code | | | | | | Logic | | | | | |
|---|---|---|---|---|---|---|---|---|---|---|---|---|---|---|---|---|---|---|
| Metric | TTB (↓) | | | BSF (↑) | | | TTB (↓) | | | BSF (↑) | | | TTB (↓) | | | BSF (↑) | | |
| Target Fraction | 50% | 75% | 100% | 25% | 50% | 100% | 50% | 75% | 100% | 25% | 50% | 100% | 50% | 75% | 100% | 25% | 50% | 100% |
| Random | 1.00 | 1.00 | 1.00 | **1.00** | 1.00 | 1.00 | 1.00 | 1.00 | 1.00 | 1.00 | 1.00 | 1.00 | 1.00 | 1.00 | 1.00 | 1.00 | 1.00 | 1.00 |
| Offline | 0.91 | **0.73** | 0.76 | 0.99 | 1.00 | **1.05** | 0.97 | 1.24 | - | 0.99 | 0.99 | 0.99 | 1.06 | 1.23 | 0.95 | 0.96 | 0.95 | 1.04 |
| BOTS-MoPPS | 1.07 | 1.15 | 0.70 | 0.94 | 1.04 | 1.04 | 0.84 | **0.69** | 0.84 | 1.04 | 1.02 | 1.00 | **0.75** | 0.92 | 0.69 | 1.07 | 1.00 | **1.05** |
| BOTS-DOTS | **0.83** | 0.80 | **0.61** | 0.98 | 1.02 | 1.03 | 0.86 | 0.96 | **0.76** | 1.02 | **1.04** | 1.02 | 0.79 | 0.91 | 0.91 | 1.02 | 0.95 | 1.03 |
| **BOTS** | 0.86 | 0.77 | 0.63 | 0.99 | **1.04** | 1.04 | **0.82** | 0.79 | 1.06 | **1.04** | 1.01 | 1.00 | 0.79 | **0.78** | **0.50** | 1.03 | **1.01** | 1.00 |

Table 4: **BOTS-Qwen2.5-7B Across Domains.** The recommended setting outperforms both out-of-framework and within-framework baselines, achieving 6 **first**-place and 8 second-place finishes out of 18 reported metrics. Full results are in Appendix G.8∼G.10.

We conduct extended experiments across model sizes (1.5B and 7B) and task domains (math, code, logic). We compare BOTS under the default setting ($\lambda = 0.1, \rho = 0.1$, posterior sampling enabled) against four baselines. Two are out-of-framework baselines: (i) **Random**, where tasks are uniformly sampled, and (ii) **Offline**, where tasks are ranked from easy to hard based on success probabilities of Qwen2.5-7B-Instruct and Qwen3-30B-A3B (for tie-breaking). Two are within-framework baselines: (iii) **BOTS-MoPPS**, with $\lambda = 0.0, \rho = 0.0$ and posterior sampling enabled, which reduces our framework to MoPPS (Qu et al., 2025) and thus enables direct evaluation of a purely explicit-evidence strategy; and (iv) **BOTS-DOTS**, with $\lambda = 1.0, \rho = 1.0$ and posterior sampling disabled,

serving as a proxy inspired by DOTS (Sun et al., 2025), which evaluates the long-term effectiveness of relying almost entirely on implicit evidence without corrective feedback. Implementation details and further discussion of these baselines are provided in Appendix D.6.

**Analysis and Takeaways.** The default BOTS setting consistently outperforms both out-of-framework and within-framework baselines, validating the principle of fusing explicit and implicit evidence. Moreover, BOTS-DOTS emerges as the strongest baseline, confirming that our interpolation-based implicit evidence provides useful guidance for task selection. These conclusions are supported by the following observations from Table 3 and Table 4:

(1) **BOTS achieves the best overall performance.** For the 1.5B model, BOTS obtains 10 first-place and 6 second-place finishes out of 18 reported metrics, with a notable 28% acceleration (TTB(100%) = 0.72) in the math domain. For the 7B model, BOTS secures 6 first-place and 8 second-place finishes, including a remarkable 50% acceleration (TTB(100%) = 0.50) in the logic domain.

(2) **BOTS-DOTS ranks second.** For the 1.5B model, BOTS-DOTS achieves 6 first-place and 4 second-place finishes. For the 7B model, it achieves 5 first-place and 1 second-place finishes, outperforming the remaining baselines.

In summary, these results demonstrate that BOTS not only delivers consistent gains across different domains and model scales, but also achieves substantial acceleration in training efficiency. The superiority of BOTS over both BOTS-MoPPS and BOTS-DOTS highlights the necessity of combining explicit and implicit evidence, while the strength of BOTS-DOTS underscores the practical value of our interpolation-based implicit evidence. Together, these findings establish BOTS as a principled, effective, and scalable solution for online task selection in LLM RFT.

## 4.5 Overview of Additional Experiments

We provide extended experiments in Appendix G for a deeper understanding of BOTS: (1) A wall-clock breakdown (Appendix G.1) shows task selection adds less than 0.2% overhead. (2) Evaluation of the interpolation-based implitict evidence with various combination of reference models (Appendix G.2) confirms its effectiveness and robustness, and provides insights for practice. (3) An ablation on Thompson sampling (Appendix G.3) shows posterior sampling yields more stable selection. (4) A fine-grained analysis of selected task dynamics (Appendix G.4) illustrates how BOTS shifts computation away from trivial ($p = 1$) and impossible ($p = 0$) tasks toward the informative mid-difficulty region. (5) Comparison against an external baseline Dynamic Sampling Yu et al. (2025) (Appendix G.5) further demonstrates the superiority of BOTS. (6) The extended experimental results in Appendix G.6~G.10 provide additional details for the experiments in this section.

## 5 Conclusion and Discussion

We introduced **BOTS**, a unified Bayesian framework for online task selection in LLM reinforcement finetuning. BOTS formulates task difficulty estimation as Bayesian belief updating, fusing two complementary evidence sources: stable but sparse *explicit evidence* from direct rollouts and dense but less precise *implicit evidence* from inter-task relationships. Instantiated with a lightweight interpolator and Thompson sampling, BOTS achieves adaptive and efficient online data selection with negligible overhead. Experiments show consistent gains in data efficiency and model performance across domains and model scales, surpassing methods that rely on a single evidence source. We envision BOTS as a practical foundation for dynamic, model-aware data selection, advancing efficient and effective LLM training.

This work opens several promising directions. First, we focused mainly on binary-reward tasks; extending and validating BOTS in non-binary reward settings is an important next step. Second, BOTS currently uses fixed update rules, though our results show that different values of $\lambda$ and $\rho$ work best at different training stages. Developing adaptive update strategies that adjust to training dynamics would further improve robustness. Third, while our lightweight interpolator efficiently provides implicit evidence, designing stronger plug-in alternatives and systematically studying the trade-off between predictive accuracy and computational cost remain open challenges. A more detailed discussion of these directions is provided in Appendix F.4.

**Ethics Statement**    All authors have read and adhered to the ICLR 2026 Code of Ethics. Our research focuses on the algorithmic efficiency of reinforcement finetuning for Large Language Models and does not involve human subjects, animal experiments, or the processing of personally identifiable information. The datasets used in our experiments are publicly available and established benchmarks within the research community; all software, datasets, and frameworks utilized are governed by the permissive Apache-2.0 open-source license. Our method aims to make AI research more sustainable and accessible, and we do not foresee any direct negative societal impacts or ethical concerns arising from our proposed methodology. The authors declare no conflict of interest.

**Reproducibility Statement**    We are committed to ensuring the full reproducibility of our research. To facilitate this, we will release our source code, which includes the implementation of the **BOTS** framework and the scripts required to replicate all experiments presented in this paper. The appendix contains a comprehensive description of our experimental setup, detailing all model configurations, dataset processing steps, and hyperparameter settings.

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

APPENDIX

# A    NOTATION SUMMARY

For ease of reading and reference, we present the symbols used in this paper in Table 5.

| Symbol | Description |
|---|---|
| ***General & Problem Setup*** | |
| $\mathcal{T}_k, \mathcal{T}$ | The $k$-th task, or a generic task. |
| $Q, O$ | A query (input) and an outcome (output) of a task. |
| $R(\cdot)$ | A binary reward function, mapping an outcome to $\{0, 1\}$. |
| $\mathcal{M}_\theta$ | The language model parameterized by $\theta$. |
| $p_{\theta,\mathcal{T}}$ | Success probability of model $\mathcal{M}_\theta$ on task $\mathcal{T}$. |
| $t$ | Index for the training step or iteration. |
| $p_t^k$ | Shorthand for $p_{\theta_t,\mathcal{T}_k}$, success probability on task $k$ at step $t$. |
| $R_t^k$ | Shorthand for the reward distribution (Bernoulli) of task $k$ at step $t$. |
| $N$ | Total number of tasks in the pool. |
| $n$ | Number of rollouts per task in a batch. |
| $\mathcal{B}_t$ | The batch of tasks selected for training at step $t$. |
| ***Bayesian Estimation Framework*** | |
| $\alpha_t^k, \beta_t^k$ | Parameters of the Beta posterior for task $k$ at step $t$. |
| $\alpha_0^k, \beta_0^k$ | Parameters of the base prior Beta distribution for task $k$. |
| $n_t^k$ | Equivalent sample size for task $k$ at step $t$, $n_t^k = \alpha_t^k + \beta_t^k$. |
| $n_0^k$ | Equivalent sample size of the prior for task $k$, $n_0^k = \alpha_0^k + \beta_0^k$. |
| $\lambda$ | Discounting factor in $[0, 1]$ for historical information. |
| $\rho$ | Balance coefficient in $[0, 1]$ for explicit vs. implicit evidence. |
| $r_i^k$ | The $i$-th binary reward obtained for task $k$. |
| $s_t^k, f_t^k$ | Explicit success and failure counts for task $k$ at step $t$. |
| $\tilde{s}_t^k, \tilde{f}_t^k$ | Pseudo success and failure counts from implicit evidence. |
| ***Interpolation-based Implicit Evidence*** | |
| $\bar{p}_w^k, \bar{p}_s^k$ | Pre-computed success rates of a weak and a strong reference model on task $k$. |
| $\bar{p}_t^{\text{ref}}, \bar{p}_w^{\text{ref}}, \bar{p}_s^{\text{ref}}$ | Avg. success rates of current, weak, and strong models on a reference batch. |
| $\mu_t$ | Relative capability coefficient of the current model at step $t$. |
| $\tilde{\mu}_t$ | Momentum-updated relative capability coefficient. |
| $\gamma$ | Momentum coefficient for updating $\tilde{\mu}_t$. |
| $\tilde{p}(k, \mathcal{B}_t)$ | Estimated success probability for task $k$ using implicit evidence. |
| ***Task Selection (Thompson Sampling)*** | |
| $p^*$ | The target success rate for optimal learning (e.g., 0.5). |
| $\hat{p}_k$ | Success rate sampled from the posterior $\text{Beta}(\alpha_t^k, \beta_t^k)$ |
| $\hat{u}_k$ | Utility of a task, computed as $|\hat{p}_k - p^*|$. |
| ***Evaluation Metrics*** | |
| $\tau$ | Target fraction (e.g., 0.5, 0.75) for defining a performance milestone. |
| $P_{\text{init}}, P_{\text{best}}$ | Initial and best performance of the baseline method. |
| $P_\tau$ | Target performance, $P_{\text{init}} + \tau \cdot (P_{\text{best}} - P_{\text{init}})$. |
| $\tau_M$ | Training step (hitting time) for method $M$ to first reach performance $P_\tau$. |
| $\beta$ | Budget ratio (e.g., 0.25, 0.5) for evaluating Best-so-far (BSF). |
| ***Generalizations & Alternatives (Appendix)*** | |
| $\mathcal{K}(\cdot, \cdot)$ | Kernel function between two tasks. |
| $\tau$ | Temperature parameter for the kernel function. |
| $\eta$ | Natural parameter of an exponential family. |
| $T(r), A(\eta)$ | Sufficient statistic and log-partition function of an exponential family. |
| $\chi, \nu$ | Hyperparameters of the conjugate prior for an exponential family. |

Table 5: Symbol Notation

# B PROOFS

## B.1 PROOF FOR PROPOSITION 1

**Proposition 1.** *Let $p \in (0,1)$ be the Bernoulli success probability at time $t$. Suppose the current belief is $\pi_t(p) = \text{Beta}(p \mid \alpha_t, \beta_t)$, and let $\pi_0(p) = \text{Beta}(p \mid \alpha_0, \beta_0)$ be a base prior. Given counts $(s_t, f_t)$ and pseudo counts $(\tilde{s}_t, \tilde{f}_t)$ with $s_t, f_t, \tilde{s}_t, \tilde{f}_t \geq 0$, define the generalized-Bayes update*

$$\pi_{t+1}(p) \;\propto\; \underbrace{\pi_t(p)^{1-\lambda} \pi_0(p)^{\lambda}}_{\text{prior mixing / discounting}} \;\times\; \underbrace{\left[p^{s_t}(1-p)^{f_t}\right]^{1-\rho}}_{\text{tempered explicit likelihood}} \;\times\; \underbrace{\left[p^{\tilde{s}_t}(1-p)^{\tilde{f}_t}\right]^{\rho}}_{\text{tempered implicit evidence}}, \qquad (4)$$

*with $\lambda \in (0,1)$ and $\rho \in [0,1]$. Then $\pi_{t+1}$ is exactly $\text{Beta}(\alpha_{t+1}, \beta_{t+1})$ with*

$$\alpha_{t+1} = (1-\lambda)\alpha_t + \lambda\alpha_0 + (1-\rho)s_t + \rho\tilde{s}_t, \qquad \beta_{t+1} = (1-\lambda)\beta_t + \lambda\beta_0 + (1-\rho)f_t + \rho\tilde{f}_t.$$

*Proof.* Write the Beta densities (up to normalization) as $\pi_t(p) \propto p^{\alpha_t-1}(1-p)^{\beta_t-1}$ and $\pi_0(p) \propto p^{\alpha_0-1}(1-p)^{\beta_0-1}$. Raising these to powers and multiplying by the (tempered) likelihood terms in equation (4) yields

$$\pi_{t+1}(p) \;\propto\; p^{(1-\lambda)(\alpha_t-1)+\lambda(\alpha_0-1)+(1-\rho)s_t+\rho\tilde{s}_t} (1-p)^{(1-\lambda)(\beta_t-1)+\lambda(\beta_0-1)+(1-\rho)f_t+\rho\tilde{f}_t}.$$

Collecting exponents shows that $\pi_{t+1}$ is Beta with parameters exactly as stated (add $+1$ back to exponents), completing the proof. $\qquad\square$

## B.2 PROOF FOR PROPOSITION 2

**Proposition 2.** *Let $n_t := \alpha_t + \beta_t$. Suppose the updates follow Equation (1)–(3) with $\lambda \in (0,1)$, $\rho \in [0,1]$, we have*

$$\liminf_{t\to\infty} n_t = n_0 + \frac{\rho}{\lambda}n, \qquad \limsup_{t\to\infty} n_t = n_0 + \frac{1}{\lambda}n.$$

*Proof.* We drop the task index for clarity. From Equation (1) and Equation (3),

$$n_{t+1} = (1-\lambda)n_t + \lambda n_0 + (1-\rho)(s_t + f_t) + \rho(\tilde{s}_t + \tilde{f}_t) = (1-\lambda)n_t + \lambda n_0 + (1-\rho)n\,\mathbb{I}[\mathcal{E}_t] + \rho n,$$

where $\mathcal{E}_t$ is the event that the task receives a direct evaluation at time $t$ (so $s_t + f_t = n$ iff $\mathcal{E}_t$ holds; otherwise $s_t + f_t = 0$). Unrolling the linear recurrence for $t \geq 0$,

$$n_t = (1-\lambda)^t n_0 + \sum_{u=0}^{t-1}(1-\lambda)^u(\lambda n_0 + \rho n + (1-\rho)n\,\mathbb{I}[\mathcal{E}_{t-1-u}])$$

$$= n_0 + \frac{\rho n}{\lambda}\left(1 - (1-\lambda)^t\right) + (1-\rho)n\sum_{u=0}^{t-1}(1-\lambda)^u\,\mathbb{I}[\mathcal{E}_{t-1-u}],$$

where we used $\sum_{u=0}^{t-1}(1-\lambda)^u\lambda = 1 - (1-\lambda)^t$. Because $\mathbb{I}[\mathcal{E}_.] \in \{0,1\}$,

$$0 \leq \sum_{u=0}^{t-1}(1-\lambda)^u\,\mathbb{I}[\mathcal{E}_{t-1-u}] \leq \sum_{u=0}^{t-1}(1-\lambda)^u = \frac{1 - (1-\lambda)^t}{\lambda}.$$

Multiplying by $(1-\rho)n$ and adding the common term $n_0 + \frac{\rho n}{\lambda}(1 - (1-\lambda)^t)$ gives the uniform bounds

$$n_0 + \frac{\rho n}{\lambda}\left(1 - (1-\lambda)^t\right) \leq n_t \leq n_0 + \frac{n}{\lambda}\left(1 - (1-\lambda)^t\right).$$

Since $(1-\lambda)^t \to 0$ exponentially, the $\liminf / \limsup$ statements follow immediately.

Moreover, these bounds are tight: if $\mathbb{I}[\mathcal{E}_t] \equiv 0$ (never directly evaluated), then $n_t = n_0 + \frac{\rho n}{\lambda}(1 - (1-\lambda)^t)$ and $\lim_{t\to\infty} n_t = n_0 + \frac{\rho n}{\lambda}$; if $\mathbb{I}[\mathcal{E}_t] \equiv 1$ (always directly evaluated), then $n_t = n_0 + \frac{n}{\lambda}(1 - (1-\lambda)^t)$ and $\lim_{t\to\infty} n_t = n_0 + \frac{n}{\lambda}$. $\qquad\square$

## C PSEUDO CODES

---

**Algorithm 1 BOTS**: **B**ayesian **O**nline **T**ask **S**election

---

**Require:** Task set $\{\mathcal{T}^k\}_{k=1}^N$, task batch size $B$, target success probability $p^*$, implicit evidence constructor $\tilde{p}(\cdot, \cdot)$, LLM parameter $\theta_0$, RFT trainer $RFT(\cdot, \cdot)$, $\lambda, \rho$.
**Ensure:** Trained model parameters $\theta$
1: Initialize $\boldsymbol{\alpha}_0, \boldsymbol{\beta}_0$.                                         ▷ If not specified, $\boldsymbol{\alpha}_0 = \mathbf{1}, \boldsymbol{\beta}_0 = \mathbf{1}$
2: **for** $t = 0$ to $T - 1$ **do**
3:     **if** Thompson Sampling **then**
4:         Sample $\tilde{p}_t^k \sim \text{Beta}(\alpha_t^k, \beta_t^k), k = 1, \ldots, N$
5:     **else**
6:         $\tilde{p}_t^k = \frac{\alpha_t^k}{\alpha_t^k + \beta_t^k}, k = 1, \ldots, N$
7:     **end if**
8:     Select a batch of tasks $\{\mathcal{T}_{\mathcal{B}_t[i]}\}_{i=1}^B$ with top-$B$ utilities $\{\hat{u}_k := -|\hat{p}_k - p^*|\}$.
9:     RFT with the selected tasks: $\theta_{t+1} = RFT(\theta_t, \{\mathcal{T}_{\mathcal{B}_t[i]}\}_{i=1}^B)$.
10:    Collect the online evaluation results of the selected tasks $\{r_{1:n}^{\mathcal{B}_t[i]}\}_{i=1}^B$.
11:    Compute explicit evidence $\{(s_t^k, f_t^k)\}_{k=1}^N$, with $\{r_{1:n}^{\mathcal{B}_t[i]}\}_{i=1}^B$.               ▷ Eq. 2
12:    Compute implicit evidence $\{(\tilde{s}_t^k, \tilde{f}_t^k)\}_{k=1}^N$, with $\tilde{p}, \{r_{1:n}^{\mathcal{B}_t[i]}\}_{i=1}^B$.       ▷ Eq. 3
13:    Update posterior $\boldsymbol{\alpha}_{t+1}, \boldsymbol{\beta}_{t+1}$, with $\lambda, \rho$.                       ▷ Eq. 1
14: **end for**
15: **return** $\theta_T$

---

**Algorithm 2** Interpolation-based Implicit Evidence Estimator

---

**Require:** Task index $k$, task batch $\{\mathcal{T}_{\mathcal{B}[i]}\}_{i=1}^B$, online evaluation results $\{r_{1:n}^{\mathcal{B}[i]}\}_{i=1}^B$, momentum coefficient $\gamma$.
**Ensure:** Estimated success rate $\tilde{p}(k, \{\mathcal{T}_{\mathcal{B}[i]}, r_{1:n}^{\mathcal{B}[i]}\}_{i=1}^B)$ for $\mathcal{T}_k$.
1: Retrieve success rates of weak/strong reference models $\{(\bar{p}_w^{\mathcal{B}[i]}, \bar{p}_s^{\mathcal{B}[i]})\}_{i=1}^B$.     ▷ Pre-computed.
2: Compute the average empirical success rates of the current, weak, and strong models on $\mathcal{B}_t$ as

$$\bar{p}^{\text{ref}}(\mathcal{B}) = \frac{1}{|\mathcal{B}|} \sum_{i=1}^B \frac{1}{n} \sum_{j=1}^n r_j^{\mathcal{B}[i]}, \quad \bar{p}_w^{\text{ref}}(\mathcal{B}) = \frac{1}{|\mathcal{B}|} \sum_{i=1}^B \bar{p}_w^{\mathcal{B}[i]}, \quad \bar{p}_s^{\text{ref}}(\mathcal{B}) = \frac{1}{|\mathcal{B}|} \sum_{i=1}^B \bar{p}_s^{\mathcal{B}[i]}.$$

3: Estimate the relative capability coefficient of the current model as

$$\mu(\mathcal{B}) = \left(\bar{p}^{\text{ref}}(\mathcal{B}) - \bar{p}_w^{\text{ref}}(\mathcal{B})\right) / \left(\bar{p}_s^{\text{ref}}(\mathcal{B}) - \bar{p}_w^{\text{ref}}(\mathcal{B})\right).$$

4: Momentum estimation: $\tilde{\mu} \leftarrow \gamma\tilde{\mu} + (1 - \gamma)\mu(\mathcal{B})$.
5: Retirve success rates of weak/strong reference models on $\mathcal{T}_k$: $(\bar{p}_w^k, \bar{p}_s^k)$.
6: Linear interpolation with clipping: $\tilde{p}(k, \{\mathcal{T}_{\mathcal{B}[i]}, r_{1:n}^{\mathcal{B}[i]}\}_{i=1}^B) = \text{clip}\left(\tilde{\mu}\,\bar{p}_s^k + (1 - \tilde{\mu})\,\bar{p}_w^k, 0, 1\right)$.
7: **return** $\tilde{p}(k, \{\mathcal{T}_{\mathcal{B}[i]}, r_{1:n}^{\mathcal{B}[i]}\}_{i=1}^B)$

---

## D IMPLEMENTATIONAL DETAILS

### D.1 TRAINING DATA

The training data for our experiments is sourced from GURU (Cheng et al., 2025), a well-curated, cross-domain RL dataset covering mathematics, code, logic, science, simulation, and tabular tasks. Each subset has been rigorously deduplicated, verified, and filtered. Our training utilizes the **math, code, and logic** subsets, with the Zebra Puzzle excluded from the logic portion due to its non-binary reward structure.

**Mathematics (54.4k)**   The Mathematics subset comprises data from **Skywork OR1** (He et al., 2025a), **DAPO** (Yu et al., 2025), and **DeepScaler** (Luo et al., 2025). For all math problems, models are instructed to provide the final answer within a \boxed{} environment.

**Code (18.1k)**   The Code subset includes programming challenges from **LeetCode** (Xia et al., 2025), **TACO-verified** (Li, 2024), **PrimeIntellect** (Mattern et al., 2025), and **LiveCodeBench** (Jain et al., 2024). For PrimeIntellect and LiveCodeBench, it incorporates pre-filtered versions provided by DeepCoder[2].

**Logic (5.0k)**   The Logic subset is composed of several benchmarks designed to test structured reasoning. It includes **Ordering Puzzles** (relational ordering), **Graph Puzzles** (implicit graph traversal), the public training splits of **ARC-AGI** (Chollet et al., 2024) and **ARC-AGI-2** (Chollet et al., 2025) (abstract grid transformations), and a 3.4k-sample from **BARC** (Li et al., 2024b) (synthetic ARC-style tasks). For these tasks, predictions are extracted from <answer> tags for reward calculation via exact match.

### D.2 EVALUATION BENCHMARKS

Followed by GURU (Cheng et al., 2025), the evaluation suite consists of a set of established benchmarks to rigorously assess the model's performance in a zero-shot setting across the same domains.

**Math** We evaluate mathematical reasoning on **AIME24** (MAA, 2024) and **MATH500** (Hendrycks et al., 2021), which cover a wide range of competition-level math problems.

**Code** Programming capabilities are assessed using **HumanEval** (Chen et al., 2021), **MBPP** (Austin et al., 2021), and a subset of **LiveCodeBench** (Jain et al., 2024). These benchmarks span from basic function generation to complex algorithmic challenges.

**Logic** Logical and abstract reasoning are measured using **Ordering Puzzles** for general reasoning, and **ARC-AGI** (Chollet et al., 2024) for grid-based abstract reasoning.

### D.3 TRAINING DETAILS

The RFT setup follows standard practices (Cheng et al., 2025; Qu et al., 2025; Sun et al., 2025), employing GRPO (Shao et al., 2024) as the RL algorithm (detailed in Appendix D.5). All experiments are conducted on verl framework (Sheng et al., 2025) with 8 NVIDIA A100 (80GB) GPUs. We utilize PyTorch's FSDP for distributed training, with vLLM (Kwon et al., 2023) employed to accelerate the response generation during the rollout phase. The actor model is optimized using a learning rate of $1 \times 10^{-6}$ and weight decay of 0.1. We apply a constant learning rate warmup for the first 10 steps and use gradient clipping with a maximum norm of 1.0. For the GRPO algorithm, we set the clipping ratio $\epsilon$ to 0.2 and generate 16 rollouts with a temperature of 1.0. The maximum prompt and response lengths are set to 4,096 and 8,192 tokens, respectively.

The training configurations are tailored for two LLMs: **Qwen2.5-1.5B-Instruct** and **Qwen2.5-7B** (Yang et al., 2024), across three reasoning domains: Math, Code, and Logic. The primary differences across settings lie in batch sizes and memory optimization strategies to accommodate different model sizes. Specifically, the 7B model experiments utilize a larger training batch size (512 vs. 256) and GRPO mini-batch size (64 vs. 32). To fit the larger model on the same hardware,

---

[2]https://www.together.ai/blog/deepcoder

we enable FSDP's CPU offloading for all 7B model experiments. Key hyperparameters for all six experimental settings are summarized in Table 6.

| Hyperparameter | 1.5B-Math | 7B-Math | 1.5B-Code | 7B-Code | 1.5B-Logic | 7B-Logic |
|---|---|---|---|---|---|---|
| Base Model | Qwen2.5-1.5B-Inst. | Qwen2.5-7B | Qwen2.5-1.5B-Inst. | Qwen2.5-7B | Qwen2.5-1.5B-Inst. | Qwen2.5-7B |
| Optimizer | AdamW | AdamW | AdamW | AdamW | AdamW | AdamW |
| Learning Rate | $1 \times 10^{-6}$ | $1 \times 10^{-6}$ | $1 \times 10^{-6}$ | $1 \times 10^{-6}$ | $1 \times 10^{-6}$ | $1 \times 10^{-6}$ |
| Weight Decay | 0.1 | 0.1 | 0.1 | 0.1 | 0.1 | 0.1 |
| Global Steps | 100 | 100 | 70 | 70 | 100 | 100 |
| Batch Size (Prompts) | 256 | 512 | 256 | 512 | 256 | 512 |
| GRPO Mini-batch Size | 32 | 64 | 32 | 64 | 32 | 64 |
| Rollouts per Prompt ($n$) | 16 | 16 | 16 | 16 | 16 | 16 |
| GRPO Clip Ratio ($\epsilon$) | 0.2 | 0.2 | 0.2 | 0.2 | 0.2 | 0.2 |
| Rollout Temperature | 1.0 | 1.0 | 1.0 | 1.0 | 1.0 | 1.0 |
| Max Prompt Length | 4,096 | 4,096 | 4,096 | 4,096 | 4,096 | 4,096 |
| Max Response Length | 8,192 | 8,192 | 8,192 | 8,192 | 8,192 | 8,192 |
| FSDP CPU Offload | Disabled | Enabled | Disabled | Enabled | Disabled | Enabled |

Table 6: Key hyperparameters for RFT across different models and domains. "Inst." is an abbreviation for Instruct.

### D.4 FORMAL DEFINITIONS OF METRICS

We provide formal definitions of the two key metrics used in our experiments: *Time-to-Baseline (TTB)* and *Best-so-far (BSF)*.

**Time-to-Baseline (TTB).** We define *Time-to-Baseline (TTB)* as a metric to measure the acceleration of a method relative to the random baseline. Let the random baseline start from performance $P_{\text{init}}$ and reach its best performance $P_{\text{best}}$ within a fixed training window of $K$ steps. For a target fraction $\tau \in \{50\%, 75\%, 100\%\}$, the target performance is defined as the corresponding interpolation between the initial and best performance:

$$P_\tau = P_{\text{init}} + \tau \cdot \left( P_{\text{best}} - P_{\text{init}} \right).$$

Denote by $\tau_M$ the (possibly interpolated) training step at which method $M$ first achieves performance $P_\tau$. Then the TTB of method $M$ is

$$\text{TTB}_M(\tau) = \frac{\tau_M}{\tau_{\text{baseline}}}.$$

By definition, $\text{TTB}_{\text{baseline}}(\tau) = 1$. Smaller values of TTB indicate that a method reaches the target improvement faster, and thus achieves greater acceleration relative to the baseline.

*Example.* Suppose the baseline starts at performance $P_{\text{init}} = 0.1$ and reaches $P_{\text{best}} = 0.3$ within 100 training steps. The total improvement is therefore 0.2. For $\tau = 50\%$, the target performance is

$$P_{50\%} = 0.1 + 0.5 \times (0.3 - 0.1) = 0.2.$$

If the baseline first reaches 0.2 at step 40, then $\tau_{\text{baseline}} = 40$. If another method $M$ reaches 0.2 earlier, at step 30, then $\tau_M = 30$, and thus

$$\text{TTB}_M(50\%) = \frac{30}{40} = 0.75.$$

This indicates that method $M$ achieves the same relative improvement (50% of the baseline's maximum gain) using only 75% of the training steps required by the baseline, reflecting a 25% acceleration.

**Best-so-far (BSF).** We define *Best-so-far (BSF)* to measure the relative performance gain of a method against the random baseline under the same training budget. Let the total training window be $T$ steps. At a budget ratio $\beta \in (0, 1]$, corresponding to step $t = \lfloor \beta T \rfloor$, denote by $\text{best}_M(t)$ the best performance achieved by method $M$ up to step $t$, and by $\text{best}_{\text{rand}}(t)$ the best performance achieved by the random baseline up to the same step. Then the BSF of method $M$ is defined as

$$\text{BSF}_\beta(M) := \frac{\text{best}_M(t)}{\text{best}_{\text{rand}}(t)}.$$

By definition, $\mathrm{BSF}_\beta(\mathrm{rand}) = 1$. Larger values indicate that the method has achieved stronger absolute performance under the same budget, i.e., it delivers better best-so-far outcomes than the baseline, not just relative improvement.

*Example.* Suppose the total training window is $T = 100$ steps, and we consider $\beta = 0.5$ ($t = 50$). If the random baseline achieves $\mathrm{best}_{\mathrm{rand}}(50) = 0.4$, while another method $M$ achieves $\mathrm{best}_M(50) = 0.6$, then

$$\mathrm{BSF}_{50\%}(M) = \frac{0.6}{0.4} = 1.5.$$

This means that by step 50, method $M$ has achieved a best-so-far performance 1.5 times that of the random baseline.

## D.5 Policy Optimization Algorithm: GRPO

The policy is optimized using **Group Relative Policy Optimization (GRPO)** (Shao et al., 2024), a policy gradient method that operates without a learned value function. In GRPO, the advantage for a given response is computed by normalizing its reward against the statistics of a group of candidate responses sampled for the same prompt.

Specifically, for each prompt $x$, a set of $G$ responses $Y = \{y_1, \ldots, y_G\}$ is sampled from the policy $\pi_{\theta_{\mathrm{old}}}$. After obtaining the reward $R(y_k)$ for each response, the group-relative advantage $A(y_k)$ is defined as:

$$A(y_k) = \frac{R(y_k) - \mu_Y}{\sigma_Y} \tag{6}$$

where $\mu_Y$ and $\sigma_Y$ represent the mean and standard deviation of the rewards $\{R(y_1), \ldots, R(y_G)\}$ for the group.

This advantage is then incorporated into a clipped surrogate objective function to update the policy parameters $\theta$:

$$\mathcal{L}_{\mathrm{GRPO}}(\theta) = \mathbb{E}_{y \sim \pi_{\theta_{\mathrm{old}}}} \left[ \min \left( r_\theta(y) A(y), \mathrm{clip}(r_\theta(y), 1 - \epsilon, 1 + \epsilon) A(y) \right) \right] \tag{7}$$

Here, $r_\theta(y) = \frac{\pi_\theta(y|x)}{\pi_{\theta_{\mathrm{old}}}(y|x)}$ denotes the probability ratio between the current and old policies. The clip function constrains this ratio within the interval $[1 - \epsilon, 1 + \epsilon]$, limiting the magnitude of policy updates during optimization.

## D.6 Baseline Details

We detail the configurations of the baselines and ablations used for comparison.

- **Random:** Tasks are sampled uniformly at random from the entire training pool. This represents a no-curriculum scenario and serves as the fundamental baseline for measuring performance gains.

- **Offline Baseline:** Tasks are pre-sorted once from easy to hard based on the success rates of external models (Qwen2.5-7B-Instruct, with Qwen3-32B-A3B for tie-breaking). This baseline represents a static curriculum and is used to benchmark our adaptive method against a fixed task sequence.

- **Setting from MoPPS (Qu et al., 2025):** This ablation uses $\lambda = 0.0, \rho = 0.0$ with posterior sampling, relying solely on explicit evidence for task selection. Under this configuration, our framework reduces exactly to the setting studied in Qu et al. (2025), enabling a direct evaluation of a purely explicit-evidence-based strategy. Note that Qu et al. (2025) did not study the code or logic domains, nor did they provide principles for determining $\lambda$ ($1 - \lambda$ in BOTS). We therefore reuse their hyperparameter choice for math ($\lambda = 1$, i.e., $\lambda = 0$ in BOTS) when evaluating other domains. This comparison is fair since no hyperparameter tuning is performed for either MoPPS or BOTS.

- **Proxy for DOTS (Sun et al., 2025):** This ablation uses $\lambda = 1.0, \rho = 1.0$ with posterior sampling disabled, so task selection is driven entirely by implicit evidence from our estimator. It serves as a proxy for the approach in Sun et al. (2025), though our construction of pseudo-counts differs, see Appendix F.2 for more details. This baseline allows us to evaluate the long-term efficacy of a strategy that relies almost exclusively on implicit evidence without direct corrective feedback. We

note that Sun et al. (2025) additionally explored reusing historical trajectories, which lies beyond the scope of this paper. To ensure clarity, we isolate such tricks from DOTS and focus solely on the online task selection component.

## E  MORE RELATED WORKS

**Reinforcement Finetuning for LLMs**    Reinforcement Finetuning (RFT) has become a pivotal technique for aligning LLMs with human values and enhancing their capabilities in complex reasoning tasks (OpenAI, 2023; Guo et al., 2025; Zeng et al., 2025; He et al., 2025a). Initial RLHF methods using PPO (Schulman et al., 2017) have been complemented by preference optimization techniques like DPO (Rafailov et al., 2023) and its variants such as KTO (Ethayarajh et al., 2024), ORPO (Hong et al., 2024) and SimPO (Meng et al., 2024). More recently, attention has shifted to rejection-sampling-based finetuning, popularized by frameworks like GPRO (Shao et al., 2024) for its effectiveness in verifiable reasoning. This has spurred variants like DAPO (Yu et al., 2025) and GSPO (Zheng et al., 2025), with further work focusing on algorithmic refinements (Cui et al., 2025; Wang et al., 2025b) and reward shaping (Chen et al., 2025a; Pan et al., 2025). Despite these algorithmic advances, the data curriculum, that how tasks are selected and presented, remains a critical bottleneck for RFT efficiency.

**The Role of Data Curriculum**    The importance of data difficulty in RFT is increasingly recognized, leading to the creation of highly challenging benchmarks (Albalak et al., 2025; He et al., 2025b; Gao et al., 2025). Beyond dataset creation, data selection and curriculum design has also become a central topic of investigation. For Supervised Finetuning (SFT), a variety of data selection and curriculum strategies have been widely studied based on quality (Liu et al., 2024; Li et al., 2024a), diversity (Ling et al., 2025; Lu et al., 2024), or pre-assessed difficulty (Xu et al., 2025; Ye et al., 2025). However, such static methods are ill-suited for the dynamic nature of RFT. This has motivated a focus on dynamic curricula that adapt to the model's evolving capabilities, typically by leveraging the notion of **task difficulty** (Cheng et al., 2025; Pikus et al., 2025; Li et al., 2025b; Toloubidokhti et al., 2023; Wang et al., 2025a).

**Task Selection Strategies for RFT**    Existing task selection strategies for RFT vary in their approach to leveraging task difficulty. An early line of work, directly inspired by curriculum learning, employs **offline curricula** that schedule tasks along a fixed easy-to-hard trajectory (Parashar et al., 2025; Shen et al., 2025; Zhu et al., 2025; Wen et al., 2025; Li et al., 2025a). While simple and intuitive, these methods are non-adaptive and cannot respond to the model's real-time learning progress. To address this, **online selection strategies** have emerged. A straightforward approach is *sampling-based task filtering*, which uses extra rollouts to evaluate and discard tasks that are too easy or too hard, thereby incurring significant computational overhead (Yu et al., 2025; Bae et al., 2025). To avoid this cost, recent works attempt to *predict* task success rates. Some frame task selection as a non-stationary multi-armed bandit (MAB) problem, but often treat tasks as independent arms, thus overlooking cross-task relationships (Chen et al., 2025b; Qu et al., 2025). Others use a small reference set to predict the difficulty of other tasks via similarity kernels, but this still requires extra rollouts and discards valuable historical information (Sun et al., 2025; Wang et al., 2025c).

**Positioning Our Work.**    The current landscape reveals a clear need for a task selection method that is simultaneously adaptive, computationally efficient, and information-complete. Our framework, **BOTS**, is designed to fill this gap. It introduces a unified Bayesian framework that is fully online and adaptive. Critically, BOTS is the first to jointly incorporate *explicit evidence* from direct evaluations and *implicit evidence* inferred from related tasks, without requiring any additional model rollouts. BOTS provides a principled, low-overhead, and effective solution for dynamic task selection in RFT.

## F    EXTENDED DISCUSSION

### F.1    A TOY EXAMPLE ILLUSTRATING THE ROLE OF $\lambda$ IN DIFFICULTY ESTIMATION

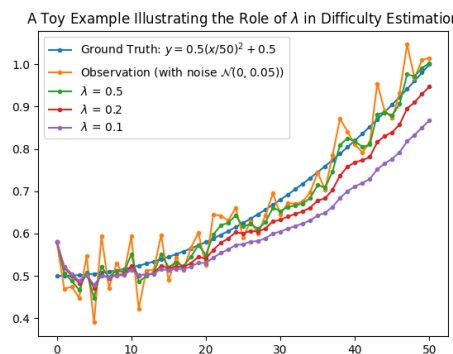

Figure 4: A toy example illustrating the role of $\lambda$ in difficulty estimation. **Ground Truth** simulates a continuously evolving success probability, while **Observations** add Gaussian noise to mimic estimation errors from explicit/implicit evidence. We plot the posterior means obtained by BOTS over 50 update steps for three settings of $\lambda \in \{0.1, 0.2, 0.5\}$. A larger $\lambda$ (e.g., 0.5) reacts quickly to rapid changes in the underlying difficulty but is highly sensitive to noise. A smaller $\lambda$ (e.g., 0.1) yields smoother and more stable estimates but lags behind when the true difficulty increases rapidly. The intermediate setting $\lambda = 0.2$ balances the two behaviors, offering better stability than $\lambda = 0.5$ and better adaptivity than $\lambda = 0.1$.

### F.2    ATTENTION-BASED ADAPTIVE DIFFICULTY ESTIMATE

Given a batch of online evaluation results $\mathcal{B}_t = \{(\mathcal{T}_{\mathcal{B}_t[i]}, r_{1:m}^{\mathcal{B}_t[i]})\}_{i=1}^{|\mathcal{B}_t|}$, we need to estimate $\tilde{s}_t^k$ and $\tilde{f}_t^k$ for each task $\mathcal{T}^k$. Besides the interplation-based estimate discussed in 3.3, another straightforward idea is to construct a kernel-based estimator:

$$\tilde{s}_t^k = \sum_{i=1}^{|B_t|} \frac{\mathcal{K}(\mathcal{T}_i, \mathcal{T}_k)}{\sum_{j=1}^{|B_t|} \mathcal{K}(\mathcal{T}_j, \mathcal{T}_k)}\, s_t^i, \qquad \tilde{f}_t^k = \sum_{i=1}^{|B_t|} \frac{\mathcal{K}(\mathcal{T}_i, \mathcal{T}_k)}{\sum_{j=1}^{|B_t|} \mathcal{K}(\mathcal{T}_j, \mathcal{T}_k)}\, f_t^i. \tag{8}$$

For example, Sun et al. (2025) proposed to use an attention-like kernel structure:

$$\mathcal{K}(\mathcal{T}, \mathcal{T}') = \exp\left( \frac{\langle \mathtt{Embd}(\mathcal{T}), \mathtt{Embd}(\mathcal{T}') \rangle}{\tau} \right). \tag{9}$$

However, the limitations of such approaches are two-folds: (1) Due to its convex structure, the kernel estimator does not support extrapolation. Specifically, given a batch of tasks with passing rates bounded within $[p_{\min}, p_{\max}]$, the estimated passing rate for any task is also restricted to this range. (2) Its effectiveness depends heavily on the quality of the task embeddings, which often require additional training, may be high-dimensional, and thus introduce non-negligible storage overhead.

### F.3    INTERPOLATION-BASED IMPLICIT EVIDENCE

**When linear interpolation is exact and *why* it becomes imperfect in practice.**

**Assumption 1** (Linear Interpolation Assumption). *Let $p(\mathcal{T})$, $p_w^{\mathrm{ref}}(\mathcal{T})$, and $p_s^{\mathrm{ref}}(\mathcal{T})$ denote the success probability of the current model, the weak reference model, and the strong reference model, respectively. We assume that all tasks share a common interpolation coefficient, i.e., for any pair of tasks $\mathcal{T}^1, \mathcal{T}^2$,*

$$\frac{p(\mathcal{T}^1) - p_w^{\mathrm{ref}}(\mathcal{T}^1)}{p_s^{\mathrm{ref}}(\mathcal{T}^1) - p_w^{\mathrm{ref}}(\mathcal{T}^1)} = \frac{p(\mathcal{T}^2) - p_w^{\mathrm{ref}}(\mathcal{T}^2)}{p_s^{\mathrm{ref}}(\mathcal{T}^2) - p_w^{\mathrm{ref}}(\mathcal{T}^2)}.$$

*We denote the (task-invariant) interpolation coefficient by $\mu^\star$.*

This assumption states that the current model's success probabilities lie on the same affine transformation of the weak and strong reference models; hence linear interpolation is a perfect estimator of task difficulty.

**Proposition 3.** *Consider a batch of tasks $\mathcal{B} := \{\mathcal{T}^k\}_{k=1}^K$ with ground-truth success probabilities $\{p(\mathcal{T}^k), p_w^{\mathrm{ref}}(\mathcal{T}^k), p_s^{\mathrm{ref}}(\mathcal{T}^k)\}_{k=1}^K$. For any task $\mathcal{T} \notin \mathcal{B}$, define the linear interpolation estimator*

$$\tilde{p}(\mathcal{T}) = \mu(\mathcal{B})\, p_s^{\mathrm{ref}}(\mathcal{T}) + \big(1 - \mu(\mathcal{B})\big)\, p_w^{\mathrm{ref}}(\mathcal{T}),$$

*where*

$$\mu(\mathcal{B}) = \frac{\frac{1}{K}\sum_{k=1}^K p(\mathcal{T}^k) - \frac{1}{K}\sum_{k=1}^K p_w^{\mathrm{ref}}(\mathcal{T}^k)}{\frac{1}{K}\sum_{k=1}^K p_s^{\mathrm{ref}}(\mathcal{T}^k) - \frac{1}{K}\sum_{k=1}^K p_w^{\mathrm{ref}}(\mathcal{T}^k)}.$$

*Then $\tilde{p}(\mathcal{T}) = p(\mathcal{T})$ for all $\mathcal{T} \notin \mathcal{B}$ if and only if Assumption 1 holds.*

*Proof.* **(If)** Assume Assumption 1 holds. Let $\mu^\star$ be the common interpolation coefficient for all tasks. Then for every $\mathcal{T}^k \in \mathcal{B}$,

$$p(\mathcal{T}^k) = \mu^\star p_s^{\mathrm{ref}}(\mathcal{T}^k) + (1 - \mu^\star)p_w^{\mathrm{ref}}(\mathcal{T}^k).$$

Taking averages over $\mathcal{B}$, we obtain

$$\frac{1}{K}\sum_{k=1}^K p(\mathcal{T}^k) = \mu^\star \frac{1}{K}\sum_{k=1}^K p_s^{\mathrm{ref}}(\mathcal{T}^k) + (1 - \mu^\star)\frac{1}{K}\sum_{k=1}^K p_w^{\mathrm{ref}}(\mathcal{T}^k).$$

Rearranging gives $\mu(\mathcal{B}) = \mu^\star$.

Now for any $\mathcal{T} \notin \mathcal{B}$, Assumption 1 implies

$$p(\mathcal{T}) = \mu^\star p_s^{\mathrm{ref}}(\mathcal{T}) + (1 - \mu^\star)p_w^{\mathrm{ref}}(\mathcal{T}).$$

Since $\mu(\mathcal{B}) = \mu^\star$, the estimator satisfies

$$\tilde{p}(\mathcal{T}) = p(\mathcal{T}).$$

Thus linear interpolation is perfect under the assumption.

**(Only If)** Suppose $\tilde{p}(\mathcal{T}) = p(\mathcal{T})$ for all $\mathcal{T} \notin \mathcal{B}$. In particular, choose two arbitrary tasks $\mathcal{T}^1, \mathcal{T}^2 \notin \mathcal{B}$. By construction of $\tilde{p}(\cdot)$, we have

$$p(\mathcal{T}^i) = \mu(\mathcal{B})\, p_s^{\mathrm{ref}}(\mathcal{T}^i) + \big(1 - \mu(\mathcal{B})\big)\, p_w^{\mathrm{ref}}(\mathcal{T}^i), \quad i = 1, 2.$$

Solving for $\mu(\mathcal{B})$ in each equation yields

$$\frac{p(\mathcal{T}^1) - p_w^{\mathrm{ref}}(\mathcal{T}^1)}{p_s^{\mathrm{ref}}(\mathcal{T}^1) - p_w^{\mathrm{ref}}(\mathcal{T}^1)} = \mu(\mathcal{B}) = \frac{p(\mathcal{T}^2) - p_w^{\mathrm{ref}}(\mathcal{T}^2)}{p_s^{\mathrm{ref}}(\mathcal{T}^2) - p_w^{\mathrm{ref}}(\mathcal{T}^2)}.$$

Thus the interpolation ratio is identical for all tasks, which is precisely Assumption 1. This completes the proof. $\square$

However, in practice, different tasks do not progress at the same learning pace, and thus Assumption 1 may not hold. This mismatch naturally introduces estimation error into the linear interpolation–based difficulty estimator. Nevertheless, our empirical results show that even with imperfect estimates, the interpolation-based implicit evidence is sufficiently informative to substantially improve training efficiency within BOTS. A more fine-grained theoretical analysis of interpolation error under richer and more realistic assumptions is an interesting direction for future work.

**Extrapolation Capability.** Unlike kernel-based estimators (Sun et al., 2025) (see Appendix F.2 for a brief introduction), which are confined to the convex hull of observed tasks, our interpolation-based estimator naturally supports *extrapolation*. For example, if the reference pair consists of a strong and a stronger model, the capability coefficient $\mu_t(\mathcal{B}_t)$ defined in Section 3.3 may fall outside $[0, 1]$, yielding extrapolated predictions. The clipping step in Equation (5) ensures these estimates remain within the feasible range $[0, 1]$.

**Additional Practical Benefits of Evaluating Reference Models** We highlight two additional practical benefits of evaluating reference models: (1) as demonstrated in GURU, reference-model evaluations can directly support *offline difficulty-aware filtering*, removing tasks that are too trivial or too hard before training even begins; and (2) these evaluations can also assist RL algorithms that rely on online difficulty estimation, such as SPO (Xu & Ding, 2025).

**Discussion on the Potential Limitations.** The simplicity of our interpolation-based approach, while a key strength, also entails two potential limitations. (i) *Expressive power:* The linear interpolation assumes a linear relationship between a model's global capability and its per-task success rate. While our empirical results suggest this is a powerful approximation (see Appendix G.2 for a validation), the true learning dynamics of LLMs may be more complex. (ii) *Distributional shift in capability estimation:* The capability coefficient $\mu_t$ is estimated on the selected batch $\mathcal{B}_t$, which is not a uniform sample from the task pool. This introduces a bias, as the model is likely to perform better on this adaptively chosen batch than on the entire dataset. Consequently, $\mu_t$ might be an overestimate of the model's true global capability. However, we argue this bias is not fatal: the primary role of $\mu_t$ is to track the *progression* of the model's capability, and even a biased estimate can provide a valuable monotonic signal for this purpose, see Section 4.4 for empirical validation. (iii) *Potential Failure Mode:* If a task exhibits identical success probabilities under both reference models, then the interpolation collapses to that shared value, making the estimated difficulty entirely independent of the current model's capability.

## F.4 DISCUSSION ON FUTURE WORKS

### F.4.1 GENERALIZATION TO OTHER REWARD DISTRIBUTIONS

While our main exposition focuses on binary rewards with a Beta-Bernoulli model, the core principles of BOTS extend naturally to any reward distribution within the exponential family that admits a conjugate prior. This generality makes BOTS a versatile blueprint for online task selection algorithms.

**The General Framework.** Let the reward $r$ for a task follow a distribution from a one-parameter exponential family: $f(r \mid \eta) = h(r) \exp(\eta T(r) - A(\eta))$, where $\eta$ is the natural parameter and $T(r)$ is the sufficient statistic. The conjugate prior for $\eta$ takes the form $p(\eta \mid \chi, \nu) \propto \exp(\chi\eta - \nu A(\eta))$, where $(\chi, \nu)$ are hyperparameters. Here, $\chi$ can be seen as a pseudo-sum of sufficient statistics from prior observations, and $\nu$ as a pseudo-count of those observations.

Our generalized Bayesian update rule from Eq. equation (4) can be directly mapped to this setting. The update for the posterior hyperparameters $(\chi_t, \nu_t)$ becomes:

$$\chi_{t+1} = (1 - \lambda)\chi_t + \lambda\chi_0 + (1 - \rho)T_{\text{explicit}} + \rho T_{\text{implicit}}, \tag{10}$$

$$\nu_{t+1} = (1 - \lambda)\nu_t + \lambda\nu_0 + (1 - \rho)n_{\text{explicit}} + \rho n_{\text{implicit}}, \tag{11}$$

where $(\chi_0, \nu_0)$ are the base prior's parameters. For $n_{\text{explicit}}$ direct observations $\{r_i\}$, the explicit evidence is $T_{\text{explicit}} = \sum_{i=1}^{n_{\text{explicit}}} T(r_i)$. For implicit evidence, we assume a pseudo-observation of size $n_{\text{implicit}}$ with an estimated total sufficient statistic $T_{\text{implicit}}$. This structure precisely mirrors our update for the Beta parameters, preserving conjugacy across iterations.

**Revisiting the Bernoulli Case.** For a Bernoulli reward $r \in \{0, 1\}$ with success probability $p$, the natural parameter is the logit $\eta = \log(p/(1 - p))$, and the sufficient statistic is $T(r) = r$. The conjugate Beta$(\alpha, \beta)$ prior corresponds to hyperparameters $\chi = \alpha - 1$ and $\nu = \alpha + \beta - 2$. With $n$ rollouts, $T_{\text{explicit}} = s_t$ (success counts), $n_{\text{explicit}} = n$, $T_{\text{implicit}} = \tilde{s}_t$, and $n_{\text{implicit}} = n$. Plugging these into Equation 10- 11 and transforming back to $(\alpha, \beta)$ parameters precisely recovers our update rule in Equation 1. This confirms that our proposed update is a specific instance of this general principle.

**Example: Gaussian Rewards.** Consider a continuous reward $r$, like a score from a powerful critic model, modeled as $R \sim \mathcal{N}(\mu, \sigma^2)$ with known variance $\sigma^2$. The conjugate prior for the mean $\mu$ is Gaussian, $\mu \sim \mathcal{N}(\mu_0, \sigma_0^2)$. In the exponential family form, $\eta = \mu/\sigma^2$ and $T(r) = r$. The prior hyperparameters are $\chi_0 = \mu_0/\sigma_0^2$ and $\nu_0 = \sigma^2/\sigma_0^2$. The BOTS update would apply directly to $(\chi_t, \nu_t)$, where $T_{\text{explicit}}$ is the sum of observed rewards and $T_{\text{implicit}}$ is an estimated sum from the

interpolator. The posterior for $\mu$ remains Gaussian, allowing for Thompson sampling by drawing a sample of the mean $\hat{\mu}$ and selecting tasks whose $\hat{\mu}$ is closest to some target score $\mu^*$.

**Example: Categorical Rewards.**   For tasks with $K$ discrete outcomes (e.g., multi-level ratings), the reward is a one-hot vector, and the distribution is categorical. The conjugate prior is the Dirichlet distribution, a multivariate generalization of the Beta. BOTS would maintain a vector of $K$ Dirichlet parameters $(\alpha_1, \ldots, \alpha_K)$ for each task. The update rules would apply component-wise to each $\alpha_j$ based on explicit and implicit counts for that outcome.

This generality significantly broadens the applicability of our framework beyond binary success/failure tasks. It provides a principled and extensible blueprint for difficulty-aware online task selection across a wide spectrum of RFT problems involving diverse reward structures.

### F.4.2   SELF-ADAPTIVE UPDATE RULES

In the main paper, we set the belief update coefficients $\lambda$ and $\rho$ as fixed hyperparameters. However, our empirical study (Section 4.3 and Section 4.2) shows that different settings benefit different training stages: smaller $\lambda$ accelerates adaptation in early training but may cause instability later, while moderate $\rho$ effectively leverages implicit evidence early on but can reduce accuracy in later stages. A natural extension is to design *self-adaptive update rules* that automatically adjust $\lambda$ and $\rho$ according to training dynamics—for example, by monitoring posterior uncertainty, validation performance, or the variance of estimated success probabilities. Such adaptive schemes would allow BOTS to dynamically balance exploration and exploitation, potentially improving robustness across diverse tasks and model scales.

### F.4.3   ALTERNATIVE PLUG-IN FOR IMPLICIT EVIDENCE

Our interpolation-based estimator provides an extremely lightweight way to generate implicit evidence without additional rollouts, but it is not the only possible choice. In fact, it also comes with an inherent limitation, see discussions in Appendix F.3. More expressive alternatives could be explored, such as kernel-based predictors (Sun et al., 2025), task-embedding regressors, or small auxiliary models trained jointly with the main model. These alternatives may improve predictive accuracy, especially when the reference models poorly bracket the training model's capability. However, they also introduce a trade-off: richer implicit evidence often requires higher computational and storage costs. Systematically characterizing this trade-off—between accuracy and efficiency in implicit evidence—remains an open research question and a promising direction for future work.

# G    ADDITIONAL EMPIRICAL RESULTS

We provide extended experiments in this section for a deeper understanding of BOTS: (1) A wall-clock breakdown (Appendix G.1) shows task selection adds less than 0.2% overhead. (2) Evaluation of the interpolation-based implitict evidence with various combination of reference models (Appendix G.2) confirms its effectiveness and robustness, and provides insights for practice. (3) An ablation on Thompson sampling (Appendix G.3) shows posterior sampling yields more stable selection. (4) A fine-grained analysis of selected task dynamics (Appendix G.4) illustrates how BOTS shifts computation away from trivial ($p = 1$) and impossible ($p = 0$) tasks toward the informative mid-difficulty region. (5) The extended experimental results in Appendix G.6~G.10 provide additional details for the experiments in this section.

## G.1    COMPUTATIONAL OVERHEAD

We examine the computational overhead introduced by task selection. The breakdown of wall-clock time across training phases is shown in Figure 5.

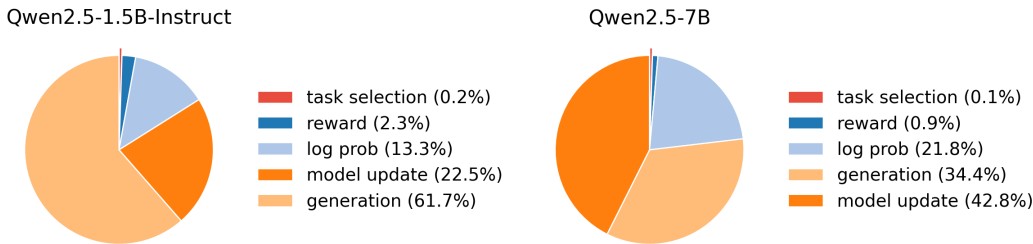

Figure 5:   Wall-clock time breakdown across training phases for **Qwen2.5-1.5B-Instruct** (Left) and **Qwen2.5-7B** (Right) on GURU-Math, trained on 8 A100 GPUs. Runtime is averaged over the first 100 training steps. The cost of task selection—including posterior sampling, index sorting, and distribution parameter updates—is negligible compared to overall training.

As illustrated, the dominant cost arises from generation and model updates, which together account for more than 75% of runtime. By contrast, the overhead of task selection—including posterior sampling, index sorting, and distribution parameter updates—is negligible (0.2% or less). Importantly, unlike generation and model updates, this cost does not increase with model size. Overall, our Bayesian framework and the chosen practical instantiation remain extremely lightweight, adding almost no extra burden to training.

## G.2    INTERPOLATION-BASED IMPLICIT EVIDENCE: EMPIRICAL RESULTS

To empirically validate our interpolation-based implicit evidence estimator, we assess its predictive quality against the evolving empirical success probabilities of the training model. We examine the trajectory under vanilla training, *i.e.*, uniformly sampling tasks for training. At each step, we compare the predictions from our interpolation-based estimator with the ground-truth online task success probabilities.

**Evaluation Metrics.** Two metrics are used: (i) **Pearson Correlation**, measuring the linear relationship between estimated and empirical difficulties, and (ii) **ROC AUC**, evaluating the ability to distinguish effective tasks (success strictly between 0 and 1) from ineffective tasks (success equal to 0 or 1).

**Training Models.** Evaluations are conducted throughout training for two models of different scales: **Qwen2.5-1.5B-Instruct** and **Qwen2.5-7B**.

**Reference Models.** Reference models are selected among three models: Qwen2.5-1.5B-Instruct (**1.5B**), Qwen2.5-7B-Instruct (**7B**), and Qwen3-30B-A3B (**30B**). Besides the default choice of ref-

erence models (**7B** × **30B**) used in Section 4, we additionally include two reference-model pairs: **1.5B** × **7B** and **1.5B** × **30B**.

The resulting Pearson Correlation and ROC AUC curves are shown in Figure 6.

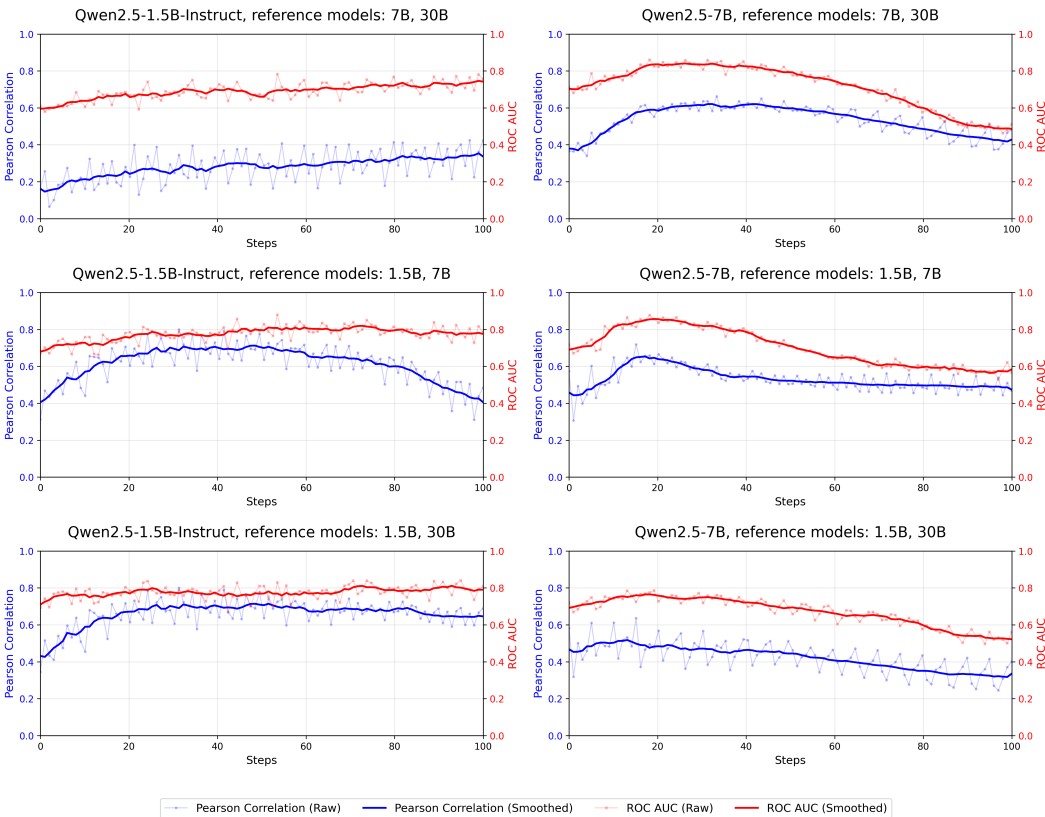

Figure 6: Performance of the interpolation-based implicit evidence estimator during training of **Qwen2.5-1.5B-Instruct** (Left) and **Qwen2.5-7B** (Right), under different choices of reference model pairs: **7B & 30B** (Top), **1.5B & 7B** (Middle), and **1.5B & 30B** (Bottom). Predictive quality is measured by **Pearson Correlation** and **ROC AUC**.

**Observations.** The consistently positive correlation and ROC AUC above 0.5 demonstrate that interpolation-based implicit evidence effectively captures task difficulty, even in the the extrapolation-heavy setting, where the pair {7B, 30B} is used to predict task success probabilities for Qwen2.5-1.5B-Instruct. All other reference-model combinations maintain even stronger predictive quality throughout training. A mild performance drop is observed in the {1.5B, 30B} → 7B setting, which is the only non-extrapolative scenario that both reference models differ substantially in capability from the training model. Additionally, we observe both Pearson Correlation and ROC AUC decline in later stages in several cases, indicating that implicit evidence becomes less informative for difficulty prediction as the model matures. This highlights the necessity of incorporating direct evaluations as explicit evidence to maintain accurate estimation.

**Analysis and Takeaways.** These results indicate that choosing well-bracketed reference models—particularly avoiding strong extrapolation regimes and, when possible, including a reference model whose capability is close to the training model—substantially improves the quality of implicit evidence within BOTS.

Next, we evaluate how different choices of reference models affect the performance of BOTS. We apply three reference model pairs to BOTS and train **Qwen2.5-1.5B-Instruct** on the **GURU-Math** dataset. The results are presented in Figure 7.

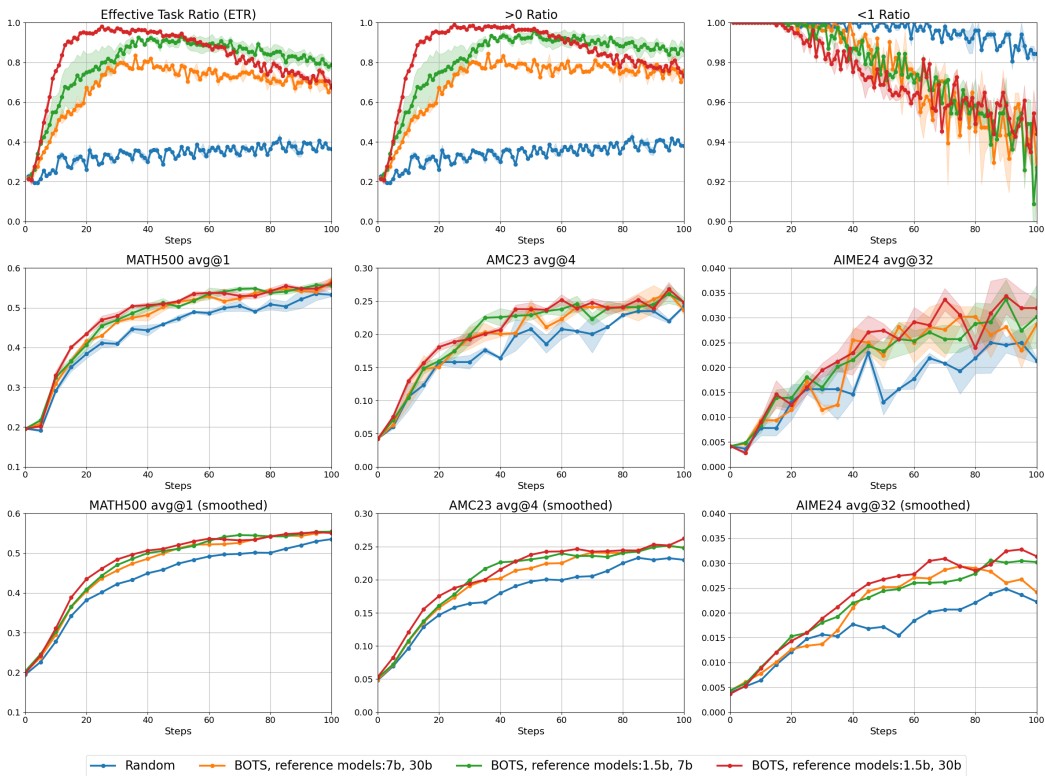

Figure 7: **Qwen2.5-1.5B-Instruct** on **Math** with various combinations of reference models. Ratios of sampled training tasks (measured with 16 rollouts) whose empirical success probabilities are strictly between 0 and 1 (LU), strictly greater than 0 (MU), and strictly less than 1 (RU), together with performance curves averaged over 3 random runs with 95% confidence intervals on MATH500 (LM), AMC23 (MM), and AIME24 (RM), plotted against training steps. The bottom row shows smoothed versions (running average, window size 3) of the curves in the middle row.

**Observations.** We observe that the reference model pairs with stronger implicit evidence estimation quality (Figure 6) – namely **1.5B × 7B** and **1.5B × 30B** – achieve clearly better ETR and more efficient training curves compared to the {7B & 30B} pair. Even though **7B × 30B** performs slightly worse than the other combinations, it still delivers performance that significantly surpasses the random baseline.

**Analysis and Takeaways.** Overall, BOTS exhibits low sensitivity to the choice of reference models: even imperfect implicit-evidence estimates (e.g., under extrapolation) still yield substantial training improvements. At the same time, better-chosen reference models produce higher-quality implicit evidence, which in turn further strengthens training efficiency and end performance.

### G.3 SAMPLING FROM POSTERIOR

We now investigate the impact of posterior sampling. As discussed in Section 3.4, posterior sampling naturally balances exploration and exploitation in bandit-style problems. Without sampling, tasks with the closest estimated success rates are greedily selected for training, which risks over-exploitation and insufficient exploration.

To examine this effect, we compare our default setting ($\lambda = 0.1, \rho = 0.1$) with posterior sampling enabled versus disabled. The valid ratio metrics and benchmark performance metrics are reported in Figure 8 and Table G.3.

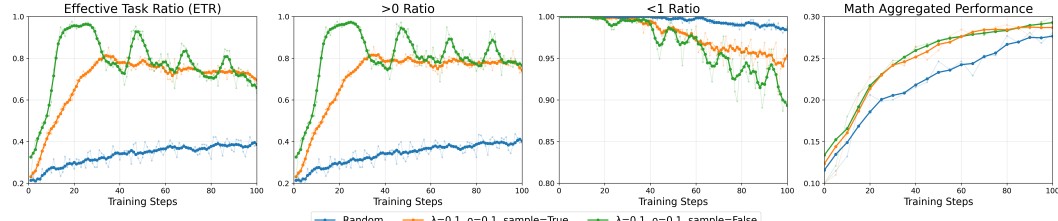

Figure 8: **Qwen2.5-1.5B-Instruct** on **Math**. Ratio of sampled training tasks (measured over 16 rollouts) with passing rates: strictly between 0 and 1, strictly greater than 0, and strictly less than 1, along with aggregated performance (MATH500 and AIME24), plotted against training steps.

| Benchmark | MATH500 | | | | | | AIME24 | | | | | | Math Aggregated Performance | | | | | |
|---|---|---|---|---|---|---|---|---|---|---|---|---|---|---|---|---|---|---|
| Metric | TTB (↓) | | | BSF (↑) | | | TTB (↓) | | | BSF (↑) | | | TTB (↓) | | | BSF (↑) | | |
| Method (↓), % (→) | 50% | 75% | 100% | 25% | 50% | 100% | 50% | 75% | 100% | 25% | 50% | 100% | 50% | 75% | 100% | 25% | 50% | 100% |
| Random | 1.00 | 1.00 | 1.00 | 1.00 | 1.00 | 1.00 | 1.00 | 1.00 | **1.00** | 1.00 | 1.00 | **1.00** | 1.00 | 1.00 | 1.00 | 1.00 | 1.00 | 1.00 |
| $\lambda = 0.1, \rho = 0.1$, sample=True | 0.89 | **0.49** | 0.57 | **1.13** | 1.05 | **1.05** | 0.51 | 1.00 | **1.00** | 1.00 | **1.75** | **1.00** | 0.89 | 0.56 | 0.64 | **1.12** | 1.07 | 1.05 |
| $\lambda = 0.1, \rho = 0.1$, sample=False | 0.73 | 0.54 | **0.54** | 1.10 | **1.09** | 1.03 | 0.77 | **0.92** | 1.25 | 1.25 | 1.50 | **1.00** | 0.79 | 0.55 | 0.81 | **1.12** | **1.10** | **1.07** |

Table 7: **Qwen2.5-1.5B-Instruct** on **Math**. TTB (lower better) and BSF (higher better) evaluated on MATH500, AIME24, and aggregated performance. For TTB, notation "-" indicates the the target performance is never achieved within the evaluation window. The **best** and second best results are marked accordingly.

**Observations.** When posterior sampling is disabled, the valid ratio exhibits a faster and higher boost in the early phase due to the removal of randomness, but fluctuations appear as training progresses. In contrast, enabling posterior sampling yields a smoother valid ratio trajectory over time. Notably, these differences in valid ratio do not translate into significant differences in benchmark performance: both settings outperform the random baseline and achieve very similar performance levels. Given the improved stability of the valid ratio, we recommend enabling posterior sampling, though this conclusion is less pronounced compared to the effects of $\lambda$ and $\rho$. We leave a larger-scale study for future work to obtain more reliable evidence.

### G.4 DYNAMICS OF SELECTED TASKS

In addition to reporting the Effective Task Ratio (ETR), which reflects the proportion of selected tasks with success probabilities strictly between 0 and 1, we conduct a finer-grained analysis to capture more detailed dynamics. Specifically, we visualize the distribution of success probabilities for selected tasks along the training trajectory, for both Qwen2.5-1.5B-Instruct and Qwen2.5-7B models. This analysis complements ETR by revealing how the quality of selected tasks evolves beyond the binary effective/ineffective distinction.

We use a heatmap where the x-axis represents training steps, the y-axis represents the empirical success rate (discretized from 0/16 to 16/16), and the color intensity indicates the proportion of tasks sampled at that success rate. This allows us to compare the distributional dynamics of BOTS against the random baseline.

The resulting heatmaps in Figure 9 reveal starkly different behaviors. The random baseline (Top) exhibits a largely static distribution, with a persistent, high-density band at the 0/16 success rate, indicating continuous wasted computation on unsolvable tasks. In contrast, BOTS (Bottom) demonstrates a highly dynamic curriculum. The initial concentration of tasks at the 0/16 success rate diminishes rapidly. Concurrently, the sampling density shifts upward, concentrating in the intermediate difficulty range.

This visualization directly illustrates how BOTS actively filters out overly easy or hard tasks and focuses computational resources on the most informative ones, which explains the superior Effective Task Ratio and overall performance gains observed previously.

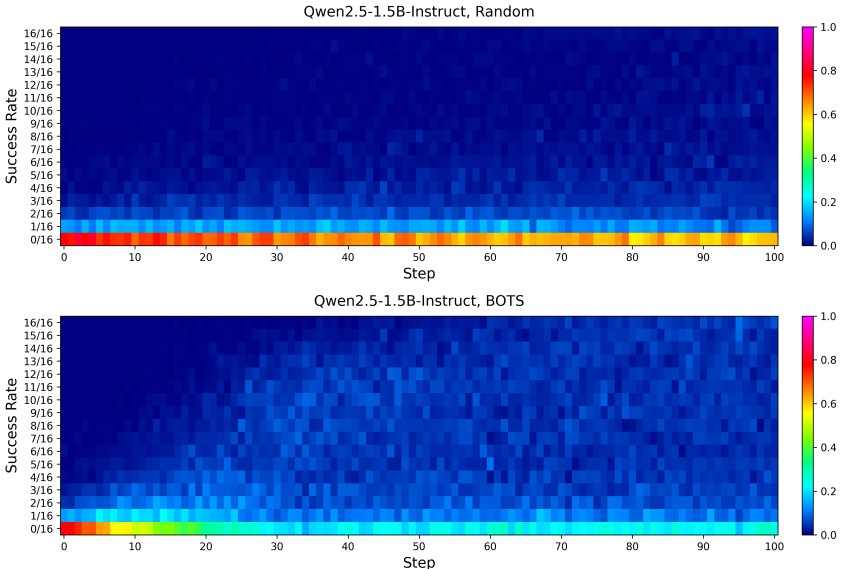

Figure 9: Heatmap visualizing the distribution of sampled task success rates over training steps for Random sampling (Top) and BOTS (Bottom) on Qwen2.5-1.5B-Instruct.

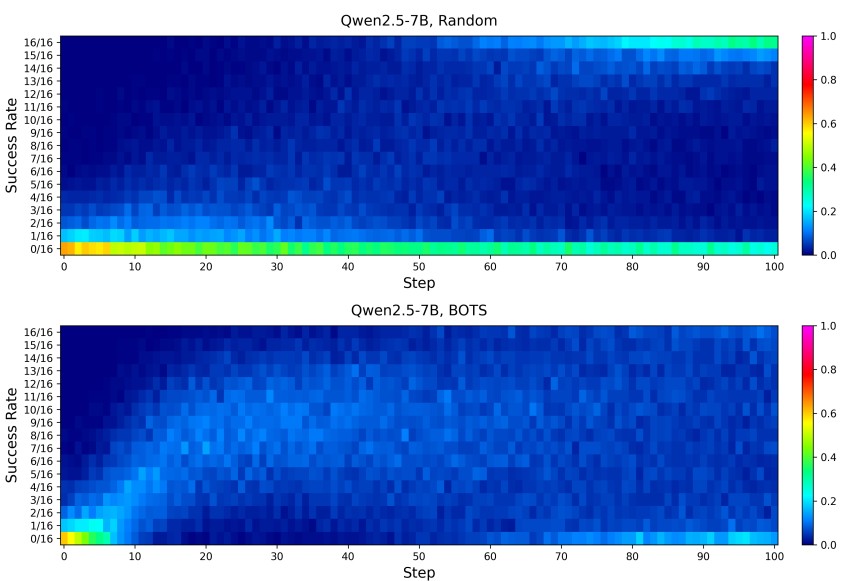

Figure 10: Heatmap visualizing the distribution of sampled task success rates over training steps for Random sampling (Top) and BOTS (Bottom) on Qwen2.5-7B.

The analysis on the 7B model, shown in Figure 10, reinforces our findings. Consistent with the 1.5B results, BOTS (Bottom) rapidly diminishes sampling of unsolvable tasks (0/16 success rate) and progressively shifts its focus to the intermediate difficulty range. However, an interesting phenomenon emerges in the later training stages due to the 7B model's stronger capability. For the random baseline (Top), a high-density band appears at the 16/16 success rate, indicating that significant computation is wasted on tasks the model has already mastered. In stark contrast, BOTS effectively avoids this region, maintaining a broad distribution across the intermediate success rates.

This demonstrates BOTS's advanced adaptivity: it not only filters out tasks that are too hard but also dynamically avoids those that become too easy, thereby maximizing learning efficiency throughout the entire training process.

## G.5 EXTERNAL BASELINE: DYNAMIC SAMPLING

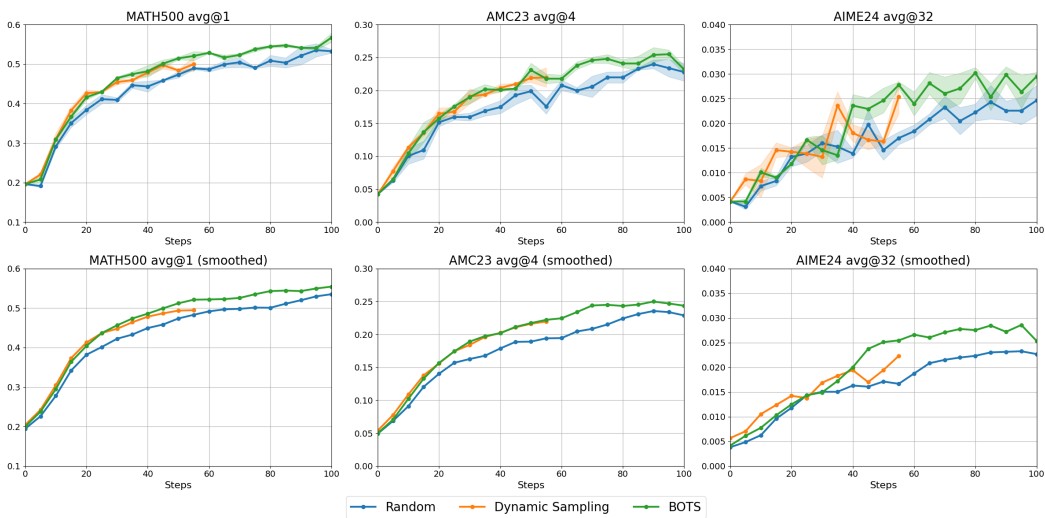

Figure 11: **Qwen2.5-1.5B-Instruct** on **Math**. Performance curves averaged over 3 random runs with 95% confidence intervals on MATH500 (LU), AMC23 (MU), and AIME24 (RU), plotted against training steps. The bottom row shows smoothed versions (running average, window size 3) of the curves in the upper row.

We conduct additional experiments comparing BOTS with *Dynamic Sampling* (Yu et al., 2025), an oversampling-based online task selection strategy. Results are shown in Figure 11. For Dynamic Sampling, we follow the original setup and train for one full epoch (55 steps, 54.4K tasks). For reference, both BOTS and the random baseline consume 25.6K tasks over 100 training steps (batch size 256). Although the performance curves are plotted step-wise for consistency, it is important to note that *each step of Dynamic Sampling incurs roughly 2.5× the time cost of a BOTS or random baseline step.*

**Observations.** Interestingly, while Dynamic Sampling consistently outperforms the random baseline, it slightly underperforms BOTS even under a step-wise comparison—despite achieving the ideal effective task ratio (ETR = 1).

**Analysis and Takeaways.** A likely explanation is that Dynamic Sampling only filters out invalid tasks (success probability 0 or 1) but does not differentiate among the remaining ones. In contrast, BOTS explicitly prioritizes tasks whose success probabilities lie near the ideal difficulty (e.g., 0.5), which are most beneficial for learning. Consequently, BOTS achieves both *higher step-wise efficiency* and *lower computational cost*. Overall, in the studied setting, BOTS outperforms Dynamic Sampling on both axes.

## G.6    EXTENDED EXPERIMENTAL RESULTS: IMPACT OF $\rho$

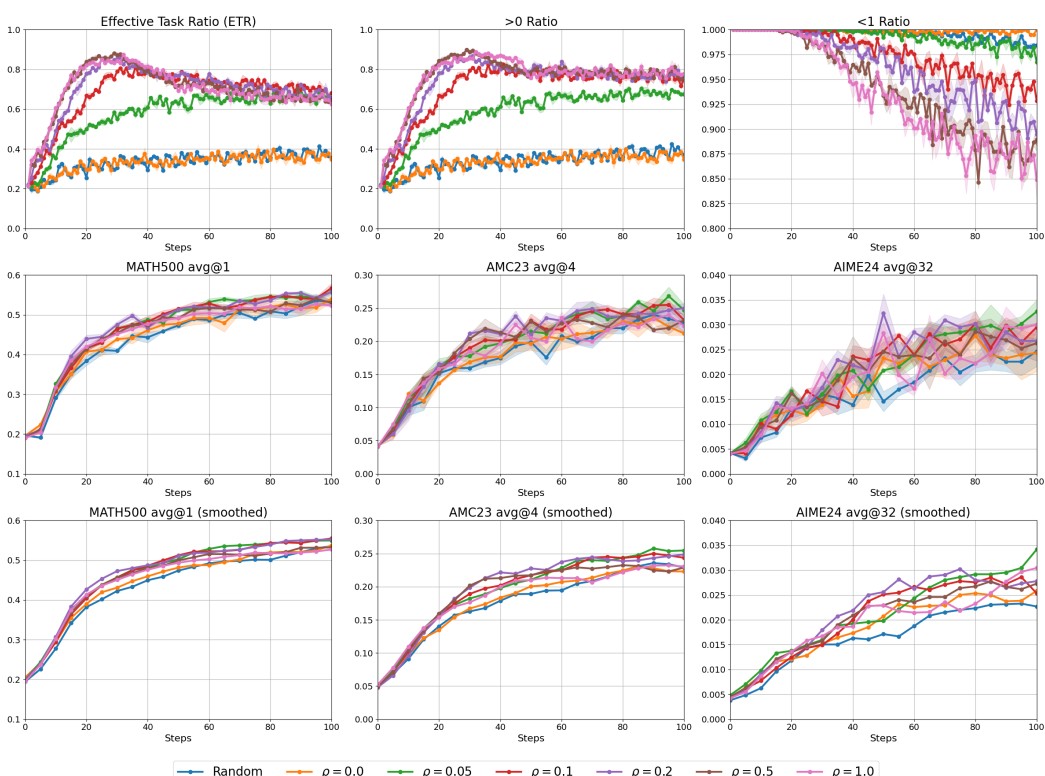

Figure 12: **Qwen2.5-1.5B-Instruct** on **Math** with $\lambda = 0.1, \rho \in \{0.0, 0.05, 0.1, 0.2, 0.5, 1.0\}$ Ratios of sampled training tasks (measured with 16 rollouts) whose empirical success probabilities are strictly between 0 and 1 (LU), strictly greater than 0 (MU), and strictly less than 1 (RU), together with performance curves averaged over 3 random runs with 95% confidence intervals on MATH500 (LM), AMC23 (MM), and AIME24 (RM), plotted against training steps. The bottom row shows smoothed versions (running average, window size 3) of the curves in the middle row.

## G.7 Additional Experimental Results: Impact of $\lambda$

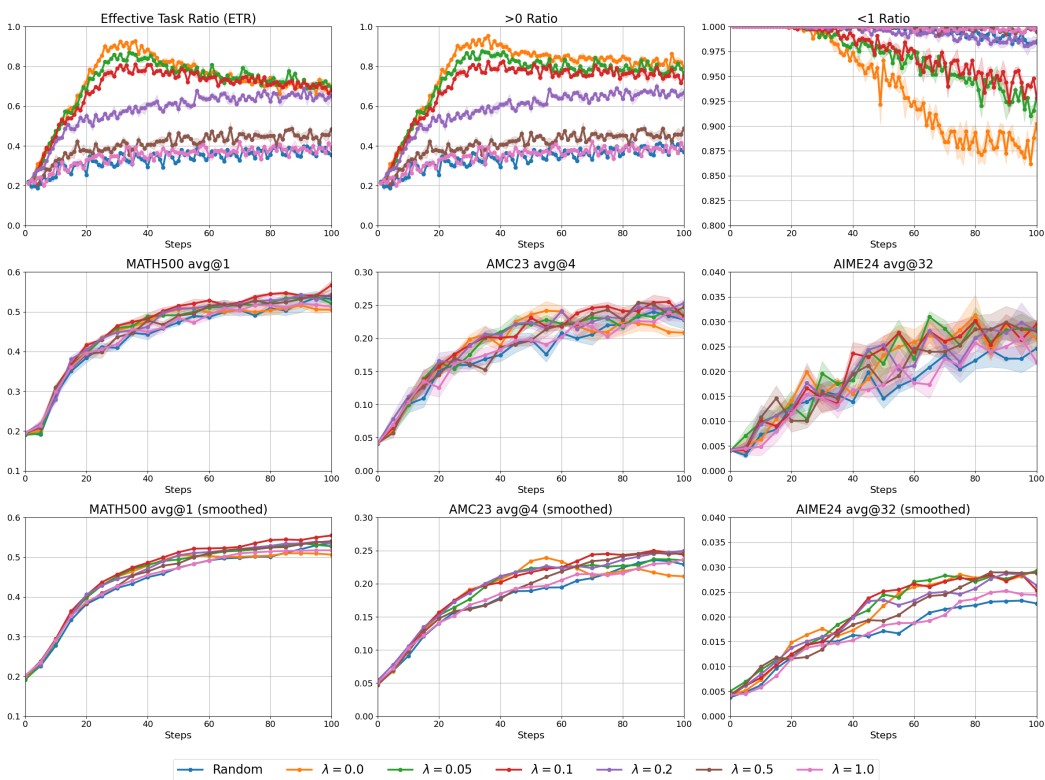

Figure 13: **Qwen2.5-1.5B-Instruct** on **Math** with $\lambda \in \{0.0, 0.05, 0.1, 0.2, 0.5, 1.0\}, \rho = 0.1$. Ratios of sampled training tasks (measured with 16 rollouts) whose empirical success probabilities are strictly between 0 and 1 (LU), strictly greater than 0 (MU), and strictly less than 1 (RU), together with performance curves averaged over 3 random runs with 95% confidence intervals on MATH500 (LM), AMC23 (MM), and AIME24 (RM), plotted against training steps. The bottom row shows smoothed versions (running average, window size 3) of the curves in the middle row.

## G.8 EXTENDED EXPERIMENTAL RESULTS: MATH

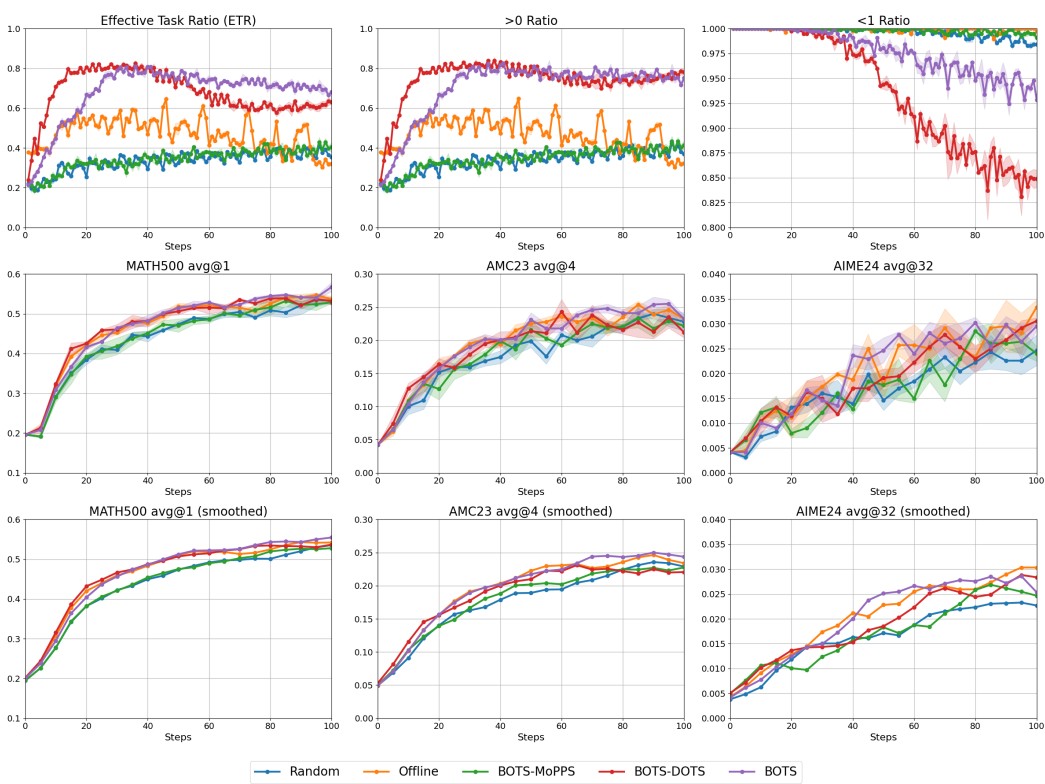

Figure 14: **Qwen2.5-1.5B-Instruct** on **Math**. Ratios of sampled training tasks (measured with 16 rollouts) whose empirical success probabilities are strictly between 0 and 1 (LU), strictly greater than 0 (MU), and strictly less than 1 (RU), together with performance curves averaged over 3 random runs with 95% confidence intervals on MATH500 (LM), AMC23 (MM), and AIME24 (RM), plotted against training steps. The bottom row shows smoothed versions (running average, window size 3) of the curves in the middle row.

| Benchmark | MATH500 | | | | | | AMC23 | | | | | | AIME24 | | | | | | Math Aggregated Performance | | | | | |
|---|---|---|---|---|---|---|---|---|---|---|---|---|---|---|---|---|---|---|---|---|---|---|---|---|
| Metric | TTB (↓) | | | BSF (↑) | | | TTB (↓) | | | BSF (↑) | | | TTB (↓) | | | BSF (↑) | | | TTB (↓) | | | BSF (↑) | | |
| Target Fraction | 50% | 75% | 100% | 25% | 50% | 100% | 50% | 75% | 100% | 25% | 50% | 100% | 50% | 75% | 100% | 25% | 50% | 100% | 50% | 75% | 100% | 25% | 50% | 100% |
| Random | 1.00 | 1.00 | 1.00 | 1.00 | 1.00 | 1.00 | 1.00 | 1.00 | 1.00 | 1.00 | 1.00 | 1.00 | 1.00 | 1.00 | 1.00 | 1.00 | 1.00 | 1.00 | 1.00 | 1.00 | 1.00 | 1.00 | 1.00 | 1.00 |
| Offline | 0.76 | 0.67 | 0.88 | 1.08 | **1.10** | 1.02 | 0.89 | **0.65** | 0.90 | 1.11 | 1.13 | 1.06 | 0.93 | **0.77** | **0.45** | 1.07 | **1.26** | **1.35** | 0.77 | **0.66** | 0.85 | **1.09** | 1.11 | 1.04 |
| BOTS-MoPPS | 0.98 | 0.94 | - | 0.99 | 1.00 | 0.99 | 1.20 | 0.85 | - | 0.98 | 1.09 | 0.97 | 1.26 | 1.41 | 0.77 | 0.95 | 0.93 | 1.15 | 1.02 | 0.90 | 0.89 | 0.98 | 1.03 | 1.00 |
| BOTS-DOTS | **0.72** | **0.56** | 0.74 | **1.12** | 1.07 | 1.01 | **0.75** | 0.76 | **0.66** | 1.03 | 1.08 | 1.01 | **0.88** | 1.23 | 0.64 | **1.18** | 0.96 | 1.24 | **0.70** | 0.70 | 0.73 | 1.08 | 1.08 | 1.01 |
| BOTS | 0.86 | 0.66 | 0.78 | 1.05 | 1.09 | **1.06** | 0.86 | 0.68 | 0.74 | **1.10** | **1.16** | **1.06** | 0.86 | 0.85 | 0.50 | **1.20** | 1.25 | 1.23 | 0.85 | 0.66 | **0.72** | 1.06 | **1.12** | **1.05** |

Table 8: TTB and BSF evaluated on **Math** with **Qwen2.5-1.5B-Instruct**. For TTB, notation "-" indicates that the target performance is never achieved within the evaluation window. The **best** and second best results are marked accordingly.

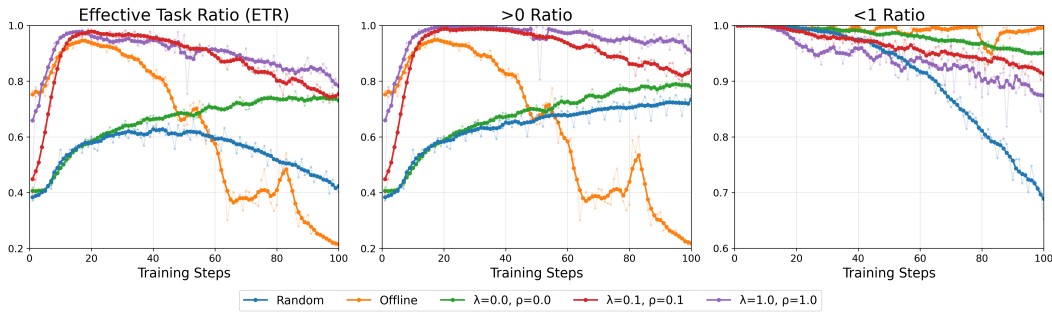

Figure 15: **Qwen2.5-7B** on **Math**. Ratio of sampled training tasks with different passing rates over training steps.

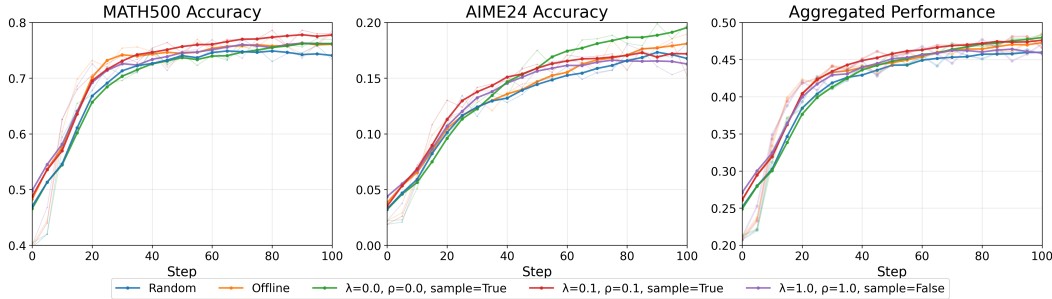

Figure 16: **Qwen2.5-7B** on **Math**. Performance on downstream math benchmarks (MATH500, AIME24) and their aggregation.

| Benchmark | MATH500 | | | | | | AIME24 | | | | | | Math | | | | | |
|---|---|---|---|---|---|---|---|---|---|---|---|---|---|---|---|---|---|---|
| Metric | TTB (↓) | | | BSF (↑) | | | TTB (↓) | | | BSF (↑) | | | TTB (↓) | | | BSF (↑) | | |
| Method (↓), % (→) | 50% | 75% | 100% | 25% | 50% | 100% | 50% | 75% | 100% | 25% | 50% | 100% | 50% | 75% | 100% | 25% | 50% | 100% |
| Random | 1.00 | 1.00 | 1.00 | 1.00 | 1.00 | 1.00 | 1.00 | 1.00 | 1.00 | **1.00** | 1.00 | 1.00 | 1.00 | 1.00 | 1.00 | **1.00** | 1.00 | 1.00 |
| Offline | 0.96 | **0.70** | **0.79** | **1.02** | 1.01 | 1.01 | 0.86 | 0.90 | 0.74 | 0.88 | 1.00 | 1.05 | 0.91 | **0.73** | 0.76 | 0.99 | 1.00 | **1.05** |
| BOTS-MoPPS | 1.10 | 1.10 | 1.35 | 0.96 | 1.00 | 1.00 | 1.14 | 0.88 | **0.64** | 0.85 | **1.19** | **1.10** | 1.07 | 1.15 | 0.70 | 0.94 | 1.04 | 1.04 |
| BOTS-DOTS | **0.89** | 0.72 | 1.07 | 1.01 | 1.00 | 1.02 | 0.96 | 0.76 | - | 0.89 | 1.07 | 0.99 | **0.83** | 0.80 | **0.61** | 0.98 | 1.02 | 1.03 |
| **BOTS** | 0.91 | 0.74 | 0.93 | 1.01 | **1.02** | **1.02** | **0.79** | **0.63** | 0.94 | 0.97 | 1.11 | 1.01 | 0.86 | 0.77 | 0.63 | 0.99 | **1.04** | 1.04 |

Table 9: **Qwen2.5-7B** on **Math**. TTB (lower better) and BSF (higher better) evaluated on downstream math benchmarks.

## G.9 EXTENDED EXPERIMENTAL RESULTS: CODE

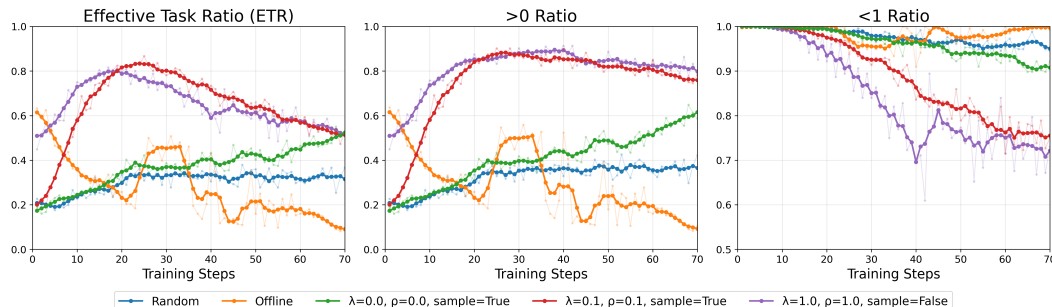

Figure 17: **Qwen2.5-1.5B-Instruct** on **Code**. Ratio of sampled training tasks with different passing rates.

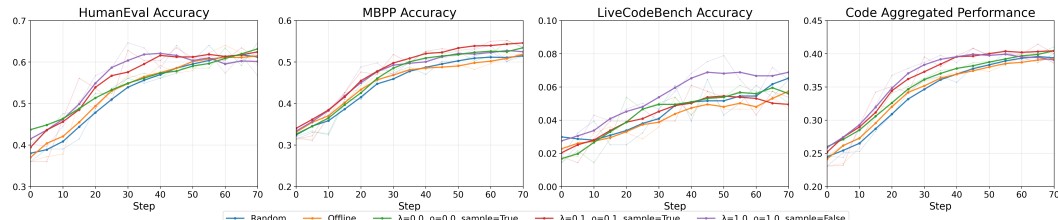

Figure 18: **Qwen2.5-1.5B-Instruct** on **Code**. Performance on downstream code benchmarks (HumanEval, MBPP, LiveCodeBench) and their aggregation.

| Benchmark | HumanEval | | | | | | MBPP | | | | | | LiveCodeBench | | | | | | Aggregated | | | | | |
|---|---|---|---|---|---|---|---|---|---|---|---|---|---|---|---|---|---|---|---|---|---|---|---|---|
| Metric | TTB (↓) | | | BSF (↑) | | | TTB (↓) | | | BSF (↑) | | | TTB (↓) | | | BSF (↑) | | | TTB (↓) | | | BSF (↑) | | |
| Method (↓), % (→) | 50% | 75% | 100% | 25% | 50% | 100% | 50% | 75% | 100% | 25% | 50% | 100% | 50% | 75% | 100% | 25% | 50% | 100% | 50% | 75% | 100% | 25% | 50% | 100% |
| Random | 1.00 | 1.00 | 1.00 | 1.00 | 1.00 | 1.00 | 1.00 | 1.00 | 1.00 | 1.00 | 1.00 | 1.00 | 1.00 | 1.00 | 1.00 | 1.00 | **1.00** | 1.00 | 1.00 | 1.00 | 1.00 | 1.00 | 1.00 | 1.00 |
| Offline | 0.69 | 0.83 | 1.09 | 1.12 | 1.01 | 1.00 | 0.69 | 1.03 | - | 1.05 | 0.99 | 0.99 | 1.12 | 1.08 | - | 0.91 | 0.88 | 0.89 | 0.68 | 1.03 | - | 1.09 | 1.00 | 0.99 |
| BOTS-MoPPS | 0.81 | 0.77 | 1.26 | 1.11 | 1.02 | **1.01** | 0.81 | 0.81 | **0.67** | 1.03 | 1.02 | **1.05** | 0.62 | 0.62 | - | 1.05 | 0.88 | 0.94 | 0.78 | 0.96 | 1.09 | 1.09 | 1.02 | 1.02 |
| BOTS-DOTS | 0.71 | **0.48** | **0.54** | 1.18 | **1.15** | **1.01** | 0.68 | **0.63** | 0.89 | 1.11 | 1.03 | 1.03 | **0.34** | 1.02 | **0.85** | **1.14** | 0.94 | **1.22** | 0.67 | **0.65** | 0.79 | 1.17 | **1.09** | 1.02 |
| **BOTS** | **0.56** | 0.63 | - | **1.24** | 1.12 | 0.99 | **0.68** | 0.66 | 0.67 | 1.10 | **1.06** | 1.05 | 0.76 | 1.18 | - | 1.00 | 0.76 | 0.94 | **0.58** | 0.90 | **0.77** | **1.17** | 1.08 | **1.03** |

Table 10: **Qwen2.5-1.5B-Instruct** on **Code**. TTB and BSF evaluated on HumanEval, MBPP, LiveCodeBench, and aggregated performance.

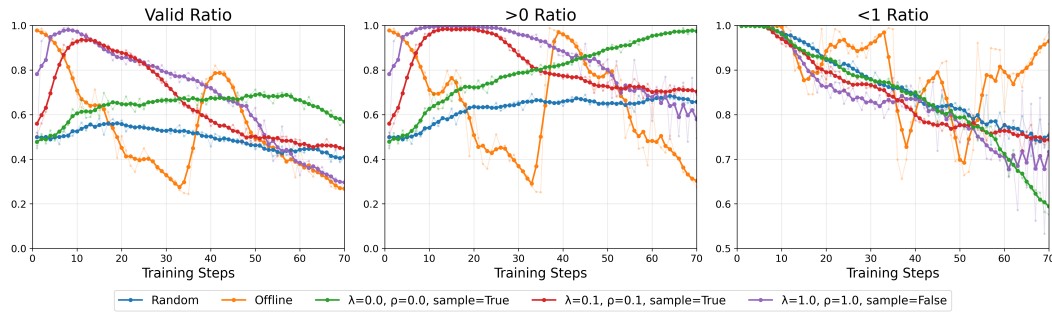

Figure 19: **Qwen2.5-7B** on **Code**. Ratio of sampled training tasks with different passing rates.

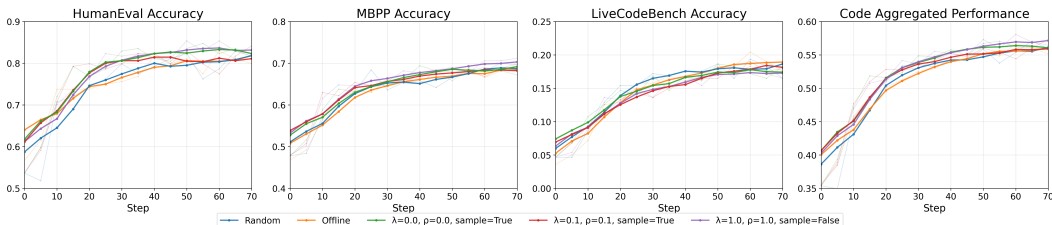

Figure 20: **Qwen2.5-7B** on **Code**. Performance on downstream code benchmarks and aggregation.

| Benchmark | HumanEval | | | | | | MBPP | | | | | | LiveCodeBench | | | | | | Aggregated | | | | | |
|---|---|---|---|---|---|---|---|---|---|---|---|---|---|---|---|---|---|---|---|---|---|---|---|---|
| Metric | TTB (↓) | | | BSF (↑) | | | TTB (↓) | | | BSF (↑) | | | TTB (↓) | | | BSF (↑) | | | TTB (↓) | | | BSF (↑) | | |
| Method (↓), % (→) | 50% | 75% | 100% | 25% | 50% | 100% | 50% | 75% | 100% | 25% | 50% | 100% | 50% | 75% | 100% | 25% | 50% | 100% | 50% | 75% | 100% | 25% | 50% | 100% |
| Random | 1.00 | 1.00 | 1.00 | 1.00 | 1.00 | 1.00 | 1.00 | 1.00 | 1.00 | 1.00 | **1.00** | 1.00 | 1.00 | 1.00 | 1.00 | **1.00** | **1.00** | 1.00 | 1.00 | 1.00 | 1.00 | 1.00 | 1.00 | 1.00 |
| Offline | **0.83** | 1.29 | 0.92 | 1.01 | 0.96 | 1.00 | 1.13 | 1.29 | - | 0.99 | 0.97 | 0.98 | 1.04 | **0.96** | **0.96** | 0.90 | 0.98 | **1.06** | 0.97 | 1.24 | - | 0.99 | 0.99 | 0.99 |
| BOTS-MoPPS | 0.91 | **0.54** | **0.46** | 1.08 | **1.05** | 1.02 | 0.83 | 1.21 | - | 1.03 | 0.96 | 0.99 | 0.97 | 1.09 | - | 0.96 | 0.98 | 0.96 | 0.84 | **0.69** | 0.84 | 1.04 | 1.02 | 1.00 |
| BOTS-DOTS | 0.97 | 0.80 | 0.73 | 1.03 | 1.03 | **1.03** | 0.73 | 0.68 | **0.91** | **1.04** | 1.00 | **1.02** | **0.95** | 1.32 | - | 0.96 | 0.94 | 0.94 | 0.86 | 0.96 | **0.76** | 1.02 | **1.04** | **1.02** |
| **BOTS** | 0.96 | 0.55 | 0.77 | **1.09** | 1.03 | 1.01 | **0.69** | 0.87 | - | 1.03 | 0.96 | 1.00 | 1.18 | 1.47 | - | 0.87 | 0.87 | 0.98 | **0.82** | 0.79 | 1.06 | **1.04** | 1.01 | 1.00 |

Table 11: **Qwen2.5-7B** on **Code**. TTB and BSF evaluated on downstream code benchmarks.

## G.10 EXTENDED EXPERIMENTAL RESULTS: LOGIC

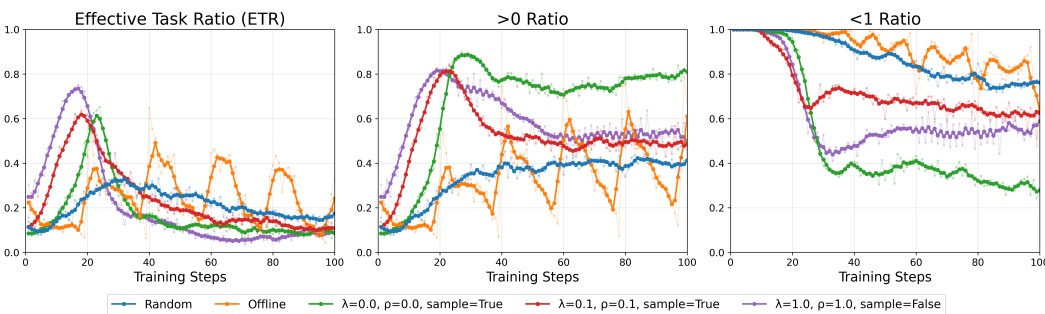

Figure 21: **Qwen2.5-1.5B-Instruct** on **Logic**. Ratio of sampled training tasks with different passing rates over training steps.

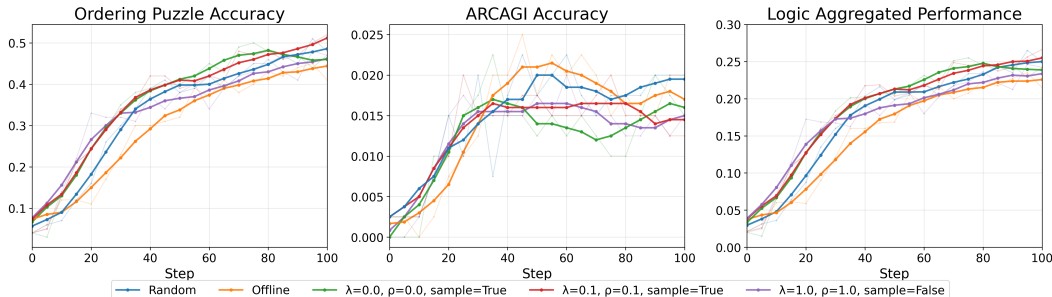

Figure 22: **Qwen2.5-1.5B-Instruct** on **Logic**. Performance on downstream logic benchmarks (Ordering Puzzle, ARCAGI) and their aggregation.

| Benchmark | Ordering Puzzle | | | | | | ARCAGI | | | | | | Aggregated | | | | | |
|---|---|---|---|---|---|---|---|---|---|---|---|---|---|---|---|---|---|---|
| Metric | TTB (↓) | | | BSF (↑) | | | TTB (↓) | | | BSF (↑) | | | TTB (↓) | | | BSF (↑) | | |
| Method (↓), % (→) | 50% | 75% | 100% | 25% | 50% | 100% | 50% | 75% | 100% | 25% | 50% | 100% | 50% | 75% | 100% | 25% | 50% | 100% |
| Random | 1.00 | 1.00 | **1.00** | 1.00 | 1.00 | 1.00 | **1.00** | 1.00 | 1.00 | 1.00 | 1.00 | 1.00 | 1.00 | 1.00 | **1.00** | 1.00 | 1.00 | 1.00 |
| Offline | 1.24 | 1.55 | - | 0.65 | 0.78 | 0.88 | 1.64 | 1.16 | 0.94 | 0.67 | **1.11** | **1.11** | 1.23 | 1.36 | - | 0.67 | 0.78 | 0.89 |
| BOTS-MoPPS | 0.87 | 1.02 | - | 1.12 | 0.98 | 0.98 | 1.23 | 1.04 | **0.78** | 1.00 | 1.00 | 1.00 | 0.87 | 1.04 | - | 1.13 | 0.96 | 0.96 |
| BOTS-DOTS | **0.65** | 1.31 | - | **1.27** | 0.93 | 0.94 | **1.00** | 0.87 | - | 1.17 | 0.78 | 0.89 | **0.66** | 1.32 | - | **1.28** | 0.92 | 0.93 |
| **BOTS** | 0.85 | **0.93** | 1.03 | 1.15 | **1.02** | **1.02** | 1.16 | **0.83** | - | **1.33** | 0.89 | 0.89 | 0.85 | **0.94** | 1.05 | 1.19 | **1.01** | **1.00** |

Table 12: **Qwen2.5-1.5B-Instruct** on **Logic**. TTB and BSF evaluated on downstream logic benchmarks.

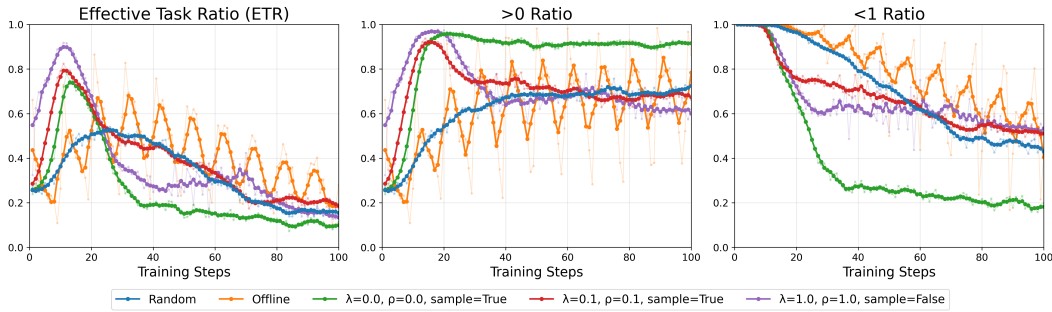

Figure 23: **Qwen2.5-7B** on **Logic**. Ratio of sampled training tasks with different passing rates over training steps.

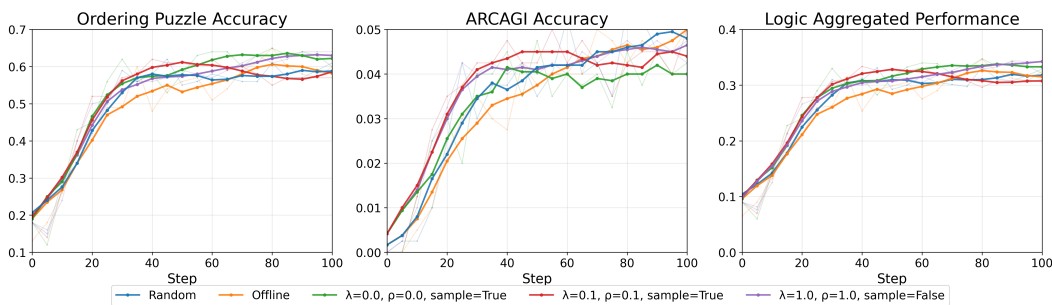

Figure 24: **Qwen2.5-7B** on **Logic**. Performance on downstream logic benchmarks and their aggregation.

| Benchmark | Ordering Puzzle | | | | | | ARCAGI | | | | | | Aggregated | | | | | |
|---|---|---|---|---|---|---|---|---|---|---|---|---|---|---|---|---|---|---|
| Metric | TTB (↓) | | | BSF (↑) | | | TTB (↓) | | | BSF (↑) | | | TTB (↓) | | | BSF (↑) | | |
| Method (↓), % (→) | 50% | 75% | 100% | 25% | 50% | 100% | 50% | 75% | 100% | 25% | 50% | 100% | 50% | 75% | 100% | 25% | 50% | 100% |
| Random | 1.00 | 1.00 | 1.00 | 1.00 | **1.00** | 1.00 | 1.00 | 1.00 | **1.00** | **1.00** | 1.00 | **1.00** | 1.00 | 1.00 | 1.00 | 1.00 | 1.00 | 1.00 |
| Offline | 1.07 | 1.21 | 1.69 | 0.98 | 0.97 | **1.05** | 1.13 | 1.81 | - | 0.71 | 1.00 | 0.91 | 1.06 | 1.23 | 0.95 | 0.96 | 0.95 | 1.04 |
| BOTS-MoPPS | **0.75** | 0.89 | **0.67** | **1.12** | **1.00** | **1.05** | 0.84 | 1.47 | - | 0.71 | **1.12** | 0.86 | **0.75** | 0.92 | 0.69 | **1.07** | **1.00** | **1.05** |
| BOTS-DOTS | 0.80 | 0.98 | 1.61 | 1.02 | 0.95 | 1.03 | 0.80 | **0.99** | - | **1.00** | 1.00 | 0.86 | 0.79 | 0.91 | 0.91 | 1.02 | 0.95 | 1.03 |
| **BOTS** | 0.80 | **0.77** | 0.89 | 1.04 | **1.00** | 1.00 | **0.72** | 1.17 | - | 0.88 | **1.12** | 0.91 | 0.79 | **0.78** | **0.50** | 1.03 | **1.01** | 1.00 |

Table 13: **Qwen2.5-7B** on **Logic**. TTB and BSF evaluated on downstream logic benchmarks.

# H USAGE OF LARGE LANGUAGE MODELS

We employed Large Language Models solely for the purpose of polishing the writing in this manuscript. Their function was limited to tasks such as correcting grammatical errors, rephrasing sentences to enhance clarity and flow, and ensuring the consistent use of terminology. The LLMs had no role in the ideation of the research, the development of the **BOTS** framework, the experimental design, or the analysis of results.

