# OpenReview forum: "BOTS: A Unified Framework for Bayesian Online Task Selection in LLM Reinforcement Finetuning"
_ICLR.cc/2026/Conference — ICLR 2026 Poster_

### Official Review · Reviewer_SYAn · 2025-10-30

**Soundness:** 3
**Presentation:** 2
**Contribution:** 2
**Rating:** 4
**Confidence:** 3

**Summary:**

The paper proposes a unified Bayesian online task selection framework for reinforcement fine-tuning of large language models. Specifically, it builds on Bayesian inference using a Beta distribution to model the policy’s success rate, incorporating both explicit evidence from batch reward signals and implicit evidence from inter-task relationships to update the parameters. The authors provide theoretical justification for the posterior inference and analyze how parameters influence the learning dynamics. Empirically, they conduct extensive experiments demonstrating the effectiveness of the proposed method and the impact of its key hyperparameters in terms of multiple metrics, including effective task ratio, time-to-baseline, and best-so-far across domains and models.

**Strengths:**

1. The proposed method is technically sound and principled. Incorporating both explicit and implicit evidence within recursive Bayesian inference framework for RL online task selection is well-motivated.
2. The paper provides both theoretical and empirical justifications for the key hyperparameters $\lambda$ and $\rho$, which is commendable.
3. The experimental analysis is thorough, and the metric design is reasonable, showing consistent gains across multiple domains and model scales.

**Weaknesses:**

1. Given that the recursive Bayesian framework has been proposed in prior work, and that the proposed estimation of implicit evidence via interpolation with the reference models is relatively simple, the overall contribution beyond prior studies [1][2] appears incremental.
2. A major issue in the paper is the discussion of prior mixing or temporal discounting, i.e., the interpretation of $\lambda$, which is confusing.
- First, it is unclear why interpolation with the prior distribution $\pi_0$ is introduced in the Bayesian sequential update. While such interpolation might be beneficial for non-stationary processes, the paper should provide appropriate references or theoretical justification.
- Several statements regarding $\lambda$ are inconsistent and potentially contradictory, raising concerns about possible implementation errors.
(1) In Proposition 2 (lines 239–242), the effective sample size $n_t$ depends inversely on $\lambda$, implying that $n_t$ increases as $\lambda$ decreases. This contradicts the claim in lines 1139–1140 that smaller $\lambda$ leads to a smaller effective sample size and higher variance.
(2) In Section 4.3, lines 350–352 state that smaller $\lambda$ emphasizes recent evaluations while larger $\lambda$ focuses on accumulated history. However, lines 373–375 claim that $\lambda \rightarrow 0$ corresponds to short-term memory and $\lambda \rightarrow 1$ to long-term memory. These descriptions are unclear and inconsistent.
- Given the known distributional form, it would be helpful to include toy examples illustrating the effects of $\lambda$ and observation noise to clarify its role.
3. In general, the analysis would benefit from separating the discussion of uncertainty estimation and success-rate estimation, rather than conflating them. The paper should more clearly define and distinguish terms such as “memory,” “forgetting,” and “exploration.”
4. The argument for low computational cost of the interpolation method in Section 3.3 is not convincing. The method requires reference labels from two models across the entire training dataset, which limits its scalability and undermines its practicality for new models and domains.

[1] Qu, Yun, et al. "Can prompt difficulty be online predicted for accelerating rl finetuning of reasoning models?." arXiv preprint arXiv:2507.04632 (2025).
[2] Sun, Yifan, et al. "Improving Data Efficiency for LLM Reinforcement Fine-tuning Through Difficulty-targeted Online Data Selection and Rollout Replay." arXiv preprint arXiv:2506.05316 (2025).

**Questions:**

1. Overall, based on Figure 3, when $\lambda$ is small, the sampled $p$ from the posterior estimation tends to be smaller than the ground truth, whereas when $\lambda$ is large, the sampled $p$ tends to be larger than the ground truth. Could you provide a consistent explanation for this behavior from both the mean and variance estimation perspectives?
2. I appreciate that the authors discuss the selection of the reference model and the potential bias in estimating $\mu_B$ in the appendix. How does the choice of reference model affect empirical performance? Is it robust? Is there any guidance or heuristic for selecting it in practice?
3. How is the prior distribution (i.e., $\alpha_0$ and $\beta_0$) for each sample initialized?
4. Could you report the correlation between the interpolated average success rate and the ground-truth average success rate across training rounds to demonstrate the accuracy of the implicit evidence estimation?

---

> ### Author Response · Authors · 2025-11-22
>
> We sincerely thank the reviewer for the insightful feedback! We are particularly grateful for your detailed critique regarding the interpretation of $\lambda$ and the suggestion to use a toy example to clarify its role.
>
> We extended our empirical study by conducting experiments on **different reference models**, and further revised the manuscript following your suggestions by providing a **consistent and comprehensive discussion on the impact of $\lambda$ and $\rho$** in Section 3.5, rephrasing ambiguous terms, adding a **toy example** to illustrate the impact of $\lambda$ on difficulty estimation, updating the **analysis on $\lambda$** in Section 4.3, extending the discussion on **the cost of reference evaluation** in Section 3.6.
>
> Below are our point-by-point responses to your Weaknesses (W) and Questions (Q).
>
> ---
>
> ### W1: Regarding marginal contribution of BOTS over prior works
>
> > *'Given that the recursive Bayesian framework has been proposed in prior work, ... the overall contribution beyond prior studies [1][2] appears incremental.'*
>
> Thanks you for raising this point, and we respectfully would like to take this opportunity to re-emphasize the contribution lies in several key aspects that advance beyond prior works. BOTS can indeed be interpreted as (i) extending explicit-only approaches such as MoPPS by additionally incorporating implicit evidence, and symmetrically, (ii) extending implicit-only approaches such as DOTS by adding explicit evidence.
>
> The core contribution of BOTS is to show—conceptually and empirically—that ***both*** sources of evidence matter, and to provide a principled Bayesian framework that unifies them in a compatible and tunable way. Our interpolation-based difficulty estimator is introduced not because it is intrinsically powerful, but to demonstrate that even a ***simple and lightweight*** form of implicit evidence is sufficient to compensate for the limitations of explicit-only methods within the BOTS framework.
>
> ---
>
> ### W2(1): Regarding the prior distribution
>
> > *'the interpretation of $\lambda$': unclear why interpolation with the prior distribution is introduced in the Bayesian sequential update*
>
> Incorporating a prior into posterior updates is not unique to our method—prior interpolation is also adopted in earlier work [1] and a standard approach for classic non-stationary bandit [2].
>
> Notice the equivalence between:
>
> (a) $s_0 = \alpha_0$, $f_0 = \beta_0$, $s_t = (1 - \lambda) s_{t-1} + \lambda \alpha_0 + ...$, $f_t = (1 - \lambda) f_{t-1} + \lambda \beta_0 + ...$, posterior Beta($s_t$, $s_t$)
> (b) $s_0 = 0$, $f_0 = 0$, $s_t = (1 - \lambda) s_{t-1} + ...$, $f_t = (1 - \lambda) f_{t-1} + ...$, posterior Beta($s_t$ + $\alpha_0$, $f_t$ + $\beta_0$)
>
> In our experiments, we use the standard uninformative Beta(1,1) prior (a uniform distribution). Combined with prior-interpolated updated rule, it guarantees $\alpha_t \ge 1$, $\beta_t \ge 1$ and thus prevents the posterior distribution from collapsing.
>
> Following the suggestion from the reviewer, we cited [2] in Section 3.2 when introducing the update rule.
>
> [1] Can Prompt Difficulty be Online Predicted for Accelerating RL Finetuning of Reasoning Models? https://arxiv.org/abs/2507.04632
>
> [2] Taming Non-stationary Bandits: A Bayesian Approach, https://arxiv.org/abs/1707.09727

---

> ### Author Response · Authors · 2025-11-22
>
> ### W2(2): Regarding the issue of $\lambda$
>
> > *'the interpretation of $\lambda$': Several statements regarding $\lambda$ are inconsistent and potentially contradictory in Proposition 2... and Section 4.3...*
>
> The reviewer correctly pointed out several inconsistent descriptions of the role of $\lambda$ in our manuscript. We sincerely thank the reviewer for this careful reading. These inconsistencies arose because we performed a notation change late in the editing process (from $1-\lambda$ to $\lambda$) to simplify the presentation in Proposition 2. Unfortunately, we did not thoroughly verify all associated analyses afterward, which resulted in contradictory statements (e.g., “smaller” vs. “larger” $\lambda$). We apologize for the confusion this caused. In the revised version, we have carefully re-checked and corrected the relevant discussions.
>
> We provide a **consistent and comprehensive discussion on the impact of $\lambda$ and $\rho$** in Section 3.5. In short, $\lambda$ and $\rho$ play roles in shaping both the **difficulty estimation** and **uncertainty profile** that drives task selection.
>
> **Impact on difficulty estimation**
> The parameter $\lambda$ controls the balance between adaptivity and stability in Bayesian belief updating.
>
> - A **larger** $\lambda$ places more weight on recent evaluations, enabling the posterior to rapidly adapt to the model's evolving capability. However, this increased adaptivity also makes the estimate more sensitive to noise in the most recent observations, reducing stability.
>
> - Conversely, a **smaller** $\lambda$ assigns greater weight to accumulated history, reducing variance in the estimated probability of success. This leads to more stable but less responsive difficulty estimates.
>
> See an illustrating toy example in Figure 4.
>
> The parameter $\rho$ governs how explicit and implicit evidence are fused.
>
> - A **larger** $\rho$ biases the update toward implicit evidence, causing tasks without direct evaluations to inherit stronger difficulty signals inferred from tasks with direct evaluations.
>
> - A **smaller** $\rho$, in contrast, favors explicit evidence, making the posterior rely primarily on direct observations and thus behave more conservatively when a task has not been recently evaluated.
>
> **Impact on selection-time uncertainty**
>
> In terms of balancing exploration toward tasks whose current success probability estimates deviate from the target and exploitation of tasks estimated to be near the target success probability, $\lambda$ controls the overall scale of the effective sample size $n_t = \alpha_t + \beta_t$, which directly dictates the variance of the posterior distribution.
>
> - A **larger** $\lambda$ accelerates discounting of older evidence, reducing $n_t$ and increasing posterior uncertainty. Under Thompson Sampling, this results in more exploratory behavior, particularly toward tasks whose current success-probability estimates deviate from the target difficulty.
>
> - A **smaller** $\lambda$ enlarges $n_t$, decreasing posterior variance and nudging the strategy toward exploitation of tasks estimated to be near the target success probability.
>
> The parameter $\rho$ primarily affects the effective sample sizes of tasks that have not been directly evaluated at the current step.
>
> - A **larger** $\rho$ increases these tasks’ effective sample sizes via stronger pseudo-count updates, reducing their posterior uncertainty and making them less likely to be explored.
>
> - A **smaller** $\rho$ decreases their effective sample sizes, increasing posterior variance and allowing tasks with difficulty estimates farther from the target to be selected, thereby improving long-run exploration.
>
> ---
>
> ### W2(3): Regarding a toy example illustrating the role of $\lambda$ in difficulty estimation
>
> > *'the interpretation of $\lambda$': would be helpful to include toy examples illustrating the effects of $\lambda$ and observation noise to clarify its role.*
>
> We thank the reviewer for the constructive suggestion. In Figure 4, Appendix E.1, we add a toy example illustrating the role of $\lambda$ in difficulty estimation. We use a quadratic function to serve as the \textbf{Ground Truth}, simulating a continuously evolving success probability. The \textbf{Observations} add Gaussian noise to mimic estimation errors from explicit/implicit evidence. We plot the posterior means obtained by BOTS over 50 update steps for three settings of $\lambda \in \{0.1, 0.2, 0.5\}$. A larger $\lambda$ (e.g., 0.5) reacts quickly to rapid changes in the underlying difficulty but is highly sensitive to noise. A smaller $\lambda$ (e.g., 0.1) yields smoother and more stable estimates but lags behind when the true difficulty increases rapidly. The intermediate setting $\lambda=0.2$ balances the two behaviors, offering better stability than $\lambda=0.5$ and better adaptivity than $\lambda=0.1$.

---

> ### Author Response · Authors · 2025-11-22
>
> ---
>
> ### W3: Regarding the separated discussion of uncertainty estimation and success‐rate estimation, and confusing term usage
>
> > *'analysis would benefit from separating the discussion of uncertainty estimation and success-rate estimation...  and should more clearly define and distinguish terms such as “memory,” “forgetting,” and “exploration.”'*
>
> We thank the reviewer for the valuable feedback that helps improve the clarity of our presentation.
>
> We provide a **consistent and comprehensive discussion on the impact of $\lambda$ and $\rho$** in Section 3.5.
>
> Regarding the usage of potentially confusing terms, we have made the following changes: (1) we revised Section 4.3 to avoid the use of the term ***memory***; (2) we revised Sections 3, 4, and Appendix A to avoid the use of the term ***forgetting***; and (3) we carefully double-checked and corrected all contexts involving ***exploration*** and ***exploitation***, ensuring that these terms are explicitly tied to the task selection—rather than reinforcement learning itself.
>
> ---
>
> ### W4: Regarding additional cost due to preference evaluation collection
>
> > *'... low computational cost of the interpolation method in Section 3.3 is not convincing...'*
>
> We expand the discussion from Section 3.3 in the original submission in **Section 3.6, paragraph 2**. Overall, our clarification is as follows:
>
> 1. While evaluating reference models requires upfront rollouts, **such evaluations are increasingly routine in modern RL dataset construction (e.g., GURU)**, and serve as reusable difficulty meta-tags;
> 2. Even when such evaluations are not readily available, they constitute a **one-time, amortizable cost**—distinct from oversampling-based online methods that repeatedly incur extra rollouts during training; and
> 3. These evaluations not only support BOTS but also provide additional benefits, such as **enabling offline difficulty-aware filtering** and **supporting RL algorithms requiring difficulty estimation** (e.g., SPO [1]).
>
> [1] Single-stream Policy Optimization, https://arxiv.org/abs/2509.13232
>
> ---
>
> ### Q1: Regarding the impact of $\lambda$
>
> > *'based on Figure 3... Could you provide a consistent explanation for this behavior from both the mean and variance estimation perspectives?'*
>
> ---
>
> We thank the reviewer for the careful inspection. We would like to clarify that Figure 3 alone does **not** support the conclusion that “small $\lambda$ underestimates task success probabilities whereas large $\lambda$ overestimates them.” We suspect this interpretation stems from the observed pattern that “small $\lambda$ selects fewer tasks with success probability =0 and more tasks with success probability =1, while large $\lambda$ shows the opposite behavior.”
>
> This phenomenon is driven by two distinct factors:
>
> (1) **Impact on difficulty estimation (posterior mean)**
> A very small $\lambda$ weakens the adaptivity of difficulty estimation, causing the posterior to rely too heavily on outdated historical evidence. As illustrated in the toy example in Appendix E.1 (Figure 4), this makes the estimator slow to react when a task’s true success probability changes (e.g., when it increases from $<1$ to $=1$), which can lead to misclassification and erroneous selection.
>
> (2) **Impact on task-selection uncertainty (posterior variance)**
> $\lambda$ also affects the effective sample size of the Beta posterior. A very large $\lambda$ yields a much smaller effective count, increasing posterior variance and thus randomness in Thompson Sampling. This pushes the behavior toward near-uniform random sampling, rather than reflecting over- or under-estimation of success probabilities.
>
> Inspired by the reviewer’s observation, we have revised the analysis in Section 4.3 to more clearly separate and explain these two effects.

---

> ### Author Response · Authors · 2025-11-22
>
> ### Q2:Regarding the choice of reference models
>
> > *'How does the choice of reference model affect empirical performance? Is it robust? Is there any guidance or heuristic for selecting it in practice?'*
>
> We are happy to know that the reviewer appreciates our discussion on the bias in estimating $\mu$.
>
> In Appendix I, we extend our study by introducing **Qwen2.5-1.5B-Instruct** as an additional reference model and evaluating three combinations of reference models: {1.5B & 7B}, {1.5B & 30B}, and {7B & 30B}. We assess both (i) the quality of interpolation-based implicit evidence (via Pearson correlation and ROC AUC), and (ii) the resulting impact on BOTS performance.
>
> *Overall, our findings are as follows:* (1) Reference-model pairs that **more closely bracket** the capability of the training model (e.g., {1.5B & 7B} or {1.5B & 30B}) yield substantially **higher-quality** implicit evidence and correspondingly stronger BOTS performance. (2) Even in **extrapolation-heavy** settings (e.g., {7B & 30B} used for a much weaker 1.5B model), where implicit evidence quality is noticeably degraded, BOTS still delivers **significant improvements** over the random baseline.
>
> These results show that BOTS is ***robust*** to reference-model choice, yet benefits further from well-chosen, capability-bracketing reference models.
>
> Based on the findings above, we additionally provides a discussion on **Choice of Reference Models** in Section 3.6.
>
> When a dataset provides pre-computed base model evaluation tags, as in GURU, we recommend directly using these tags as reference-model signals. This avoids any additional rollout cost, even though it may introduce extrapolation -- while extrapolation can reduce the accuracy of implicit evidence, BOTS remains robust, making the computational savings generally worthwhile. When multiple reference-model pairs are available, we suggest choosing models with a **clear capability gap**, which provides stronger discrimination for interpolating task difficulty. For instance, GURU’s choice of Qwen2.5-7B-Instruct and Qwen3-30B-A3B yields informative difficulty signals across a wide range.
> Finally, although the interpolation-based estimator supports extrapolation in principle, our results show that avoiding extrapolation produces more accurate implicit evidence and slightly stronger overall BOTS performance.
>
> ---
>
> ### Q3: Regarding the setting of prior
>
> > *'How is the prior distribution for each sample initialized?'*
>
> In our experiments, we simply use the standard uninformative Beta prior with $\alpha=1$ and $\beta=1$, although this choice is not essential.
>
> ---
>
> ### Q4: Regarding the quality of implicit evidence
>
> > *'Could you report the correlation between ... to demonstrate the accuracy of the implicit evidence estimation?'*
>
> In the original submission, we evaluated the quality of interpolation-based implicit evidence in Appendix F.2 by measuring the Pearson correlation and ROC AUC between the interpolated estimates (using {7B, 30B} as reference models) and the ground-truth success probabilities during training of both the 1.5B and 7B models. To ensure unbiased estimates of these online success probabilities, training was performed under ***random task sampling***, so that each batch evaluation provides an unbiased estimate of dataset-level performance.
>
> In the revised manuscript, Appendix I further extends this study by introducing **Qwen2.5-1.5B-Instruct** as an additional reference model and evaluating all three combinations: {1.5B & 7B}, {1.5B & 30B}, and {7B & 30B}. Across these combinations, we assess (i) the predictive quality of implicit evidence (via Pearson correlation and ROC AUC) and (ii) the resulting impact on BOTS performance.
>
> The key conclusion is as follows: Reference-model pairs that better bracket the capability of the training model (e.g., {1.5B & 7B} or {1.5B & 30B}) yield ***substantially higher-quality*** implicit evidence. However, even when implicit-evidence quality is noticeably degraded in heavy extrapolation settings (e.g., {7B & 30B} used for 1.5B training), BOTS still delivers ***consistent and significant improvements*** over the random baseline. Overall, BOTS is not overly sensitive to the quality of implicit evidence, while better reference models can further enhance its performance.
>
> ---
>
> ### Closing Remarks
>
> We sincerely thank you for raising such thoughtful and important concerns. We hope these clarifications can address your concerns, and we respectfully ask you to consider a higher accessment and confidence rating on our paper.

---

> > ### Comment · Reviewer_SYAn · 2025-11-23
> >
> > Thank you for the detailed response. My concerns have been addressed, and I will raise my scores accordingly. Overall I find the proposed unified, principled, and lightweight Bayesian framework for online task selection compelling, and I appreciate the thoughtful discussion supporting it. I am inclined to recommend acceptance.

---

> > > ### Author Response · Authors · 2025-11-25
> > >
> > > We are glad to hear that our responses have addressed the reviewer's concerns. We sincerely thank the reviewer again for the constructive feedback and insightful discussions that greatly helped us improve the work. If the reviewer has any further questions or would like to discuss additional aspects of BOTS, we would be more than happy to continue the conversation.

---

### Official Review · Reviewer_UaYb · 2025-10-31

**Soundness:** 2
**Presentation:** 4
**Contribution:** 3
**Rating:** 4
**Confidence:** 4

**Summary:**

This paper proposes a principled Bayesian framework to improve task selection efficiency during RFT of large language models. Instead of uniformly sampling tasks, which wastes computation on trivial or unsolvable ones, BOTS maintains Bayesian posteriors over task difficulties that evolve with model capability. It fuses explicit evidence (direct evaluation results) and implicit evidence (inferred task difficulty via inter-task relationships) using generalized Bayesian updates, and employs Thompson sampling to balance exploration and exploitation. The authors further introduce an ultra-light interpolation plug-in that estimates unselected task difficulty without extra rollouts. Experiments across math, code, and logic domains, using Qwen models of different scales, show that BOTS consistently improves data efficiency and training speed with negligible computational overhead.

**Strengths:**

1. The paper presents an interesting and original idea of combining both explicit and implicit evidence for online task selection, and unifies them under a principled Bayesian framework.

2. The interpolation-based implicit evidence plug-in is novel and practical, enabling efficient estimation without additional rollouts.

3. The writing is very clear and well-organized, with a solid and detailed appendix that strengthens the paper’s technical quality.

4. The experimental section is thorough: it clearly introduces evaluation metrics, systematically analyzes the effects of ρ and λ, and then presents comprehensive main results demonstrating certain improvements.

**Weaknesses:**

1. A major issue lies in the analysis of λ. The statement at line 244 that “smaller values accelerate forgetting and increase posterior uncertainty” appears inconsistent with Equation (1) and Proposition 2. Based on the update rule, smaller λ should slow down forgetting and reduce posterior uncertainty due to larger effective sample size. Moreover, line 372 later describes λ → 0 as “long memory,” which directly contradicts the earlier statement. The paper should carefully revisit this part.

Relatedly, the use of terms such as forgetting, exploration, and exploitation is somewhat confusing. It is not always clear what exactly is being “forgotten” or what dimensions are being “explored” versus “exploited,” especially since these terms may differ from their usage in reinforcement learning. Clarifying these concepts would improve readability and conceptual precision.

Overall, I suggest the authors carefully revisit the analysis of how λ and ρ affect the algorithm. The current explanations are confusing and inconsistent, making it difficult to assess whether the paper’s claims about their impact are correct.

2. On the experimental side, the evaluation on math benchmarks is limited. Only MATH500 and AIME24 are used, which seems insufficient to demonstrate generality. Moreover, AIME24 may be too difficult for small models, leading to unstable results. For example, as shown in the appendix figures, the 1.5B model’s AIME24 accuracy remains near zero throughout training, making it difficult to draw meaningful conclusions from such results.

---

I would be open to adjusting my rating depending on how the authors address these issues.

**Questions:**

1. Regarding the interpolation-based implicit evidence plug-in in Section 3.3: (1) What happens if there are multiple candidate weak and strong models? Would the resulting interpolations be consistent across different reference pairs?
(2) I saw the empirical results in the appendix, but the reported correlation coefficients (around 0.5) seem rather low. Could you elaborate on why this level of correlation is sufficient to justify the estimator’s reliability?
(3) If part of a dataset is either too easy or too hard—so that both weak and strong models perform similarly high or low—would the interpolation still behave well? I understand you could in principle extrapolate, but is there any empirical evidence showing this works reliably?
(4) Also, while you claim efficiency due to pre-computed references in existing datasets, would new datasets still require additional one-time rollouts for the reference models? It would be helpful to discuss this limitation explicitly.

2. In Section 4.2, you state that implicit evidence “fails to recognize when tasks are fully mastered.” Could you provide further explanation or empirical evidence to support this claim?

3. (Minor) For the BSF metric, why do you report results at 25% and 50% of total training steps? Wouldn’t BSF at 100% be sufficient to summarize the final performance?

4. (Minor) I suggest adding a complete pseudocode of the full algorithm in the appendix. This would help readers clearly follow the overall logic and implementation flow.

5. (Optional) Including error bars for key results—at least for the 1.5B model—would make the experimental analysis more convincing, if computational resources permit.

---

> ### Author Response · Authors · 2025-11-22
>
> We sincerely thank the reviewer for your insightful feedback and constructive suggestions, which have helped us further clarify and strengthen our work.
>
> We extended our empirical study by including **AMC23** for evaluating 1.5B model, conducting experiments on **different reference models**, repeating core experiments by **three** independe runs and reporting the **confidence interval**.
>
> We further revised the manuscript following your suggestions by providing a **consistent and comprehensive discussion on the impact of $\lambda$ and $\rho$** in Section 3.5, rephrasing ambiguous terms, including further discussion on the **limitation of interpolation-based estimator** in Appendix E.3, extending the discussion on **the cost of reference evaluation** in Section 3.6, updating the **analysis on large $\rho$** in Section 4.2, and including pseudo codes in Appendix H.
>
> We will address each of your Weaknesses (W) and Questions (Q) below.
>
> ---
>
> ### W1: Regarding the issue of $\lambda$
>
> > *'A major issue lies in the analysis of λ. The statement at line 244 appears inconsistent with Equation (1) and Proposition 2...'*
>
> The reviewer correctly pointed out several inconsistent descriptions of the role of $\lambda$ in our manuscript. We sincerely thank the reviewer for this careful reading. These inconsistencies arose because we performed a notation change late in the editing process (from $1-\lambda$ to $\lambda$) to simplify the presentation in Proposition 2. Unfortunately, we did not thoroughly verify all associated analyses afterward, which resulted in contradictory statements (e.g., “smaller” vs. “larger” $\lambda$). We apologize for the confusion this caused. In the revised version, we have carefully re-checked and corrected the relevant discussions.
>
> We provide a **consistent and comprehensive discussion on the impact of $\lambda$ and $\rho$** in Section 3.5. In short, $\lambda$ and $\rho$ play roles in shaping both the **difficulty estimation** and **uncertainty profile** that drives task selection.
>
> **Impact on difficulty estimation**
> The parameter $\lambda$ controls the balance between adaptivity and stability in Bayesian belief updating.
>
> - A **larger** $\lambda$ places more weight on recent evaluations, enabling the posterior to rapidly adapt to the model's evolving capability. However, this increased adaptivity also makes the estimate more sensitive to noise in the most recent observations, reducing stability.
>
> - Conversely, a **smaller** $\lambda$ assigns greater weight to accumulated history, reducing variance in the estimated probability of success. This leads to more stable but less responsive difficulty estimates.
>
> See an illustrating toy example in Figure 4.
>
> The parameter $\rho$ governs how explicit and implicit evidence are fused.
>
> - A **larger** $\rho$ biases the update toward implicit evidence, causing tasks without direct evaluations to inherit stronger difficulty signals inferred from tasks with direct evaluations.
>
> - A **smaller** $\rho$, in contrast, favors explicit evidence, making the posterior rely primarily on direct observations and thus behave more conservatively when a task has not been recently evaluated.
>
> **Impact on selection-time uncertainty**
>
> In terms of balancing exploration toward tasks whose current success probability estimates deviate from the target and exploitation of tasks estimated to be near the target success probability, $\lambda$ controls the overall scale of the effective sample size $n_t = \alpha_t + \beta_t$, which directly dictates the variance of the posterior distribution.
>
> - A **larger** $\lambda$ accelerates discounting of older evidence, reducing $n_t$ and increasing posterior uncertainty. Under Thompson Sampling, this results in more exploratory behavior, particularly toward tasks whose current success-probability estimates deviate from the target difficulty.
>
> - A **smaller** $\lambda$ enlarges $n_t$, decreasing posterior variance and nudging the strategy toward exploitation of tasks estimated to be near the target success probability.
>
> The parameter $\rho$ primarily affects the effective sample sizes of tasks that have not been directly evaluated at the current step.
>
> - A **larger** $\rho$ increases these tasks’ effective sample sizes via stronger pseudo-count updates, reducing their posterior uncertainty and making them less likely to be explored.
>
> - A **smaller** $\rho$ decreases their effective sample sizes, increasing posterior variance and allowing tasks with difficulty estimates farther from the target to be selected, thereby improving long-run exploration.

---

> ### Author Response · Authors · 2025-11-22
>
> ### W2: Regarding confusing term usage
>
> > *'Relatedly, the use of terms such as forgetting, exploration, and exploitation is somewhat confusing...'*
>
> We thank the reviewer for the valuable feedback that helps improve the clarity of our presentation. In the revised manuscript, we have made the following changes: (1) we revised Sections 3, 4, and Appendix A to avoid the use of the term ***forgetting***, which is not essential to our presentation and may cause unintended ambiguity; and (2) we carefully double-checked and corrected all contexts involving ***exploration*** and ***exploitation***, ensuring that these terms are explicitly tied to the task selection—rather than reinforcement learning itself. These revisions substantially improve conceptual clarity and reduce potential confusion for readers.
>
> ---
> ### W3: Regarding limited benchmarks for math performance evaluation
>
> > *'...the evaluation on math benchmarks is limited. Only MATH500 and AIME24 are used... Moreover, AIME24 may be too difficult for small models...'*
>
> We thank the reviewer for the constructive suggestion. In the revised manuscript, we have included **AMC23** as an additional **medium-difficulty** benchmark in all math experiments presented in Section 4, Appendix I, J for the 1.5B and 3B models. Across these expanded evaluations, we observe trends consistent with those reported in the main text, further strengthening the reliability and comprehensiveness of our empirical study. We hope this extension addresses the reviewer’s concern regarding benchmark coverage.
>
> ---
>
> ### Q1(1): Regarding reference models
>
> > *'What happens if there are multiple candidate weak and strong models? Would the resulting interpolations be consistent across different reference pairs?'*
>
> We thank the reviewer for raising this important point. In Appendix I, we extend our study by introducing **Qwen2.5-1.5B-Instruct** as an additional reference model and evaluating three combinations of reference models: {1.5B & 7B}, {1.5B & 30B}, and {7B & 30B}. We assess both (i) the quality of interpolation-based implicit evidence (via Pearson correlation and ROC AUC), and (ii) the resulting impact on BOTS performance.
>
> *Overall, our findings are as follows:* (1) Reference-model pairs that **more closely bracket** the capability of the training model (e.g., {1.5B & 7B} or {1.5B & 30B}) yield substantially **higher-quality** implicit evidence and correspondingly stronger BOTS performance. (2) Even in **extrapolation-heavy** settings (e.g., {7B & 30B} used for a much weaker 1.5B model), where implicit evidence quality is noticeably degraded, BOTS still delivers **significant improvements** over the random baseline.
>
> These results show that BOTS is ***robust*** to reference-model choice, yet benefits further from well-chosen, capability-bracketing reference models. We explicitly mention this as a practical consideration in Section 3.6.
>
> ---
>
> ### Q1(2): Regarding the quality of implicit evidence and its impact on BOTS
>
> > *'the reported correlation coefficients (around 0.5) seem rather low. Could you elaborate on why this level of correlation is sufficient to justify the estimator’s reliability?'*
>
> We again direct the reviewer to the new results provided in Appendix I. The key finding is that ***even when the interpolation-based implicit evidence is of relatively low quality***—such as in extrapolation-heavy settings where the reference models are much stronger than the training model—**BOTS still consistently outperforms the random baseline**. While higher-quality implicit evidence (e.g., from better-bracketed reference-model pairs) further improves performance, our empirical results suggest that **BOTS is not particularly sensitive to the exact accuracy of implicit evidence**. In short, implicit evidence of even moderate quality is already sufficiently informative to boost training efficiency, and BOTS remains robust across a wide range of reference-model choices.

---

> ### Author Response · Authors · 2025-11-22
>
> ### Q1(3): Regarding the extreme case that the dataset is too easy or too hard for reference models
>
> > *'If part of a dataset is either too easy or too hard—so that both weak and strong models perform similarly high or low—would the interpolation still behave well?... is there any empirical evidence showing this works reliably? '*
>
>
> This is an excellent point. Indeed, if both reference models perform identically on a task (e.g., both achieving success probability 0 or 1), then interpolation degenerates: the estimated difficulty becomes independent of the training model’s capability and collapses to the reference models’ performance. This introduces noise into implicit evidence.
>
> In our experiments, this issue did not occur because the GURU dataset performs difficulty-aware preprocessing, removing tasks that the strong reference model always fails on or that the weak reference model always solves. In other domains, similar preprocessing can be applied to avoid such degenerate cases, though we acknowledge this constrains dataset construction.
>
> The good news is that under BOTS, such tasks still have a ***non-zero*** probability of being selected due to Thompson sampling. Once selected, explicit evidence from direct rollouts compensates for the failure of implicit evidence, allowing the posterior to recover.
>
> Nonetheless, we agree that this represents a limitation of interpolation-based difficulty estimation. We have updated Appendix E.3 to explicitly discuss this scenario and clarify the behavior of BOTS under such cases.
>
> ---
> ### Q1(4): Regarding additional cost due to preference evaluation collection
>
> > *'while you claim efficiency due to pre-computed references in existing datasets, would new datasets still require additional one-time rollouts for the reference models?'*
>
> Yes, collecting preference evaluations does require additional rollouts when applying BOTS to a new dataset.
>
> We expand the discussion from Section 3.3 in the original submission in **Section 3.6, paragraph 2**. Overall, our clarification is as follows:
>
> 1. While evaluating reference models requires upfront rollouts, **such evaluations are increasingly routine in modern RL dataset construction (e.g., GURU)**, and serve as reusable difficulty meta-tags;
> 2. Even when such evaluations are not readily available, they constitute a **one-time, amortizable cost**—distinct from oversampling-based online methods that repeatedly incur extra rollouts during training; and
> 3. These evaluations not only support BOTS but also provide additional benefits, such as **enabling offline difficulty-aware filtering** and **supporting RL algorithms requiring difficulty estimation** (e.g., SPO [1]).
>
> [1] Single-stream Policy Optimization, https://arxiv.org/abs/2509.13232

---

> ### Author Response · Authors · 2025-11-22
>
> ### Q2: Regarding the explanation for large $\rho$ values
>
> > *'In Section 4.2... “fails to recognize when tasks are fully mastered.” Could you provide further explanation or empirical evidence to support this claim?'*
>
> The original claim in Section 4.2—that implicit evidence “fails to recognize when tasks are fully mastered’’—was inferred from the observation that the proportion of partially solved tasks ($<1$ ratio) decreases more rapidly under large $\rho$ compared to other settings. However, upon further reflection, we acknowledge that this explanation is not rigorous.
> Different $\rho$ values influence the stochasticity of task sampling and thus alter the training dynamics in ways that cannot be cleanly isolated. As a result, we cannot conclusively attribute the observed degradation to an inherent inability of implicit evidence to detect fully mastered tasks.
>
> Since we are currently unable to disentangle these interacting effects to obtain a stronger conclusion, we have revised the wording in Section 4.2 to remove the over-interpretation and instead focus solely on reporting the empirical phenomena.
>
>
> ---
>
> ### Q3: Regarding the BSF metric
>
> > *'For the BSF metric, why do you report results at 25% and 50% of total training steps? Wouldn’t BSF at 100% be sufficient to summarize the final performance?'*
>
> BSF(100%) captures the performance improvement at a ***single*** checkpoint, specifically, the relative best performance achieved when both methods consume the same number of training tasks. However, this metric alone does not reflect how task selection strategies influence the ***entire*** training trajectory. To obtain a more comprehensive understanding of training dynamics, we therefore report BSF at multiple checkpoints, allowing us to examine performance gains throughout different stages of training rather than only at the endpoint.
>
> ---
>
> ### Q4: Regarding the request for algorithm pseudocode
>
> > *'suggest adding a complete pseudocode of the full algorithm in the appendix...'*
>
> Thanks for the constructive suggestion. We have added pseudocode for both the full BOTS framework (Algorithm 1) and the interpolation-based implicit evidence estimator (Algorithm 2). These have been included in Appendix H.
>
> ---
>
> ### Q5: Regarding statistical significance tests
>
> > *'Including error bars for key results...'*
>
> We have expended experiments in Section 4.2, 4.3, and 1.5B-math experiments in Section 4.4, each conducted with 3 independent runs using different random seeds. All curve plots now include 95% confidence intervals, and all table entries are computed using the mean over these 3 runs. Figure 2-3, Table 1-3 are updated accordingly. Such a practice is also applied to the new experimental results reported in Appendix I & J.
>
> Overall, the extended results are consistent with the main conclusions reported in the paper, including the effects of $\lambda$ and $\rho$ in BOTS and the performance gains of BOTS over baselines. We hope these additional results alleviate concerns regarding the statistical robustness of our empirical findings.
>
> ---
>
> ### Closing Remark
>
> We thank you once again for your valuable feedback, which will significantly help us improve our work. We will integrate all new experiments and analyses into the revision of our manuscript. We hope our responses and new empirical evidence have addressed your concerns and would be grateful if you would reconsider your assessment.

---

> > ### Comment · Reviewer_UaYb · 2025-11-23
> >
> > Thank you very much for the detailed and thoughtful responses. First, I want to commend the authors for the care and effort put into the rebuttal. I also appreciate the academic rigor reflected in your replies—for instance, acknowledging the earlier flaw in Q1(3) and the potential over-interpretation in Section 4.2.
> >
> > Regarding W1, I now understand your explanation, which aligns with my own reasoning. While λ and ρ remain conceptually tricky and may still pose challenges for readers, I believe your current clarification and presentation are correct.
> >
> > For W2, the results are still somewhat limited, as the AIME performance remains very weak (especially for smaller models), making it unclear whether it provides a meaningful signal. Thus, the conclusions are still effectively based on two datasets, and the overall gains remain modest. That said, I appreciate that you added error bars—this shows strong scientific integrity. I also understand that RFT inherently involves substantial noise, and asking for more benchmarks would mainly add experimental burden rather than yield qualitatively new insights. Therefore, although the results are not entirely satisfying, I find the idea itself sensible, the method well-designed, and the presentation and discussion already of high quality.
> >
> > On Q1 (1)–(2), thank you for the additional experiments. Please correct me if I misunderstood—if the trained model is 1.5B, how can the same 1.5B model be used as a reference model?
> >
> > For Q1 (3), I appreciate your thoughtful discussion; I agree that this is a difficult open question and could be future work.
> >
> > For Q1 (4), I read your discussion. I think it would be better to simply acknowledge this limitation more directly — it is an understandable one and does not require a defensive tone. That said, I do think your reasoning in the discussion is sound and well-justified.
> >
> > For the remaining points, thank you for your clarifications; the pseudocode addition is indeed helpful.
> >
> > Overall, I look forward to your clarification on Q1 (1)–(2) (I may have missed some detail—please correct me if so). **I believe the paper is now moving in the right direction, and I am raising my score to 6.** If time allows, I will also review the other reviewers’ comments — some have raised questions about novelty relative to prior work — and may adjust my score further. That said, in my view, the current version of the paper already meets a solid standard of quality.

---

> > > ### Author Response · Authors · 2025-11-25
> > >
> > > We again thank the reviewer for the constructive and insightful comments, which have greatly helped us improve the quality and clarity of our work. We also sincerely appreciate the reviewer’s time in reading our responses and examining our revised manuscript, as well as the encouraging feedback on our revision efforts.
> > >
> > > ---
> > >
> > > **Regarding Q1(1)(2)**, we further clarify how the 1.5B model can serve as its own weak reference model by describing the full setup explicitly:
> > >
> > > (a) Let the initialization of the 1.5B training model (either any checkpoint or the standard *Qwen2.5-1.5B-Instruct* parameters) be denoted as $\theta_0$. Before training begins, we evaluate this model on all tasks and record the empirical success probabilities (with extra cost, as GURU did not provide the "Qwen2.5-1.5B-Instruct passing rate" meta-tag). These evaluations constitute the **weak reference model** signals. (Intuitively, we assume the model improves during training, so all future checkpoints surpass $\theta_0$, making it a valid weak reference. These checkpoints are **not** identical to $\theta_0$)
> > >
> > > (b) We then choose a second model as the **strong reference model** (e.g., *Qwen3-30B-A3B*) and perform the same evaluation. Alternatively, one may conceptualize an oracle model that achieves success probability 1 on all tasks.
> > >
> > > (c) Training is then carried out using BOTS for online task selection. As training progresses, the capability of the model is expected to exceed that of the weak reference model, implying that the capability coefficient ($\mu$) remains positive. Under this setup, the interpolation-based estimator naturally predicts task success probabilities strictly larger than those of the weak reference model.
> > >
> > > Importantly, our framework does not impose **any** structural assumptions on the relationship between the reference models and the training model. Formally, one could even choose an extremely weak model (success probability 0 on all tasks) and an oracle model (success probability 1 on all tasks) as the reference pair. However, such choices provide no meaningful differentiation between easy and hard tasks, causing the implicit evidence to become uninformative. This highlights why we recommend selecting reference models that provide meaningful difficulty contrast, though the BOTS framework itself remains agnostic to this choice.
> > >
> > > ---
> > >
> > > **Regarding Q1(4)**, we apologize for our misunderstanding of the reviewer’s original intention in the first-round response. We initially interpreted the comment as a concern about the **practicality** of the additional rollout cost, and thus our reply focused on discussing practical considerations. In fact, as the reviewer correctly pointed out, we explicitly acknowledge this cost in Section 3.6 (Line 280) and subsequently provide a detailed discussion of its implications and tradeoffs.
> > >
> > > We hope that the expanded analysis and clarifications in the revised manuscript adequately address the reviewer’s concern regarding this computational cost.
> > >
> > > ---
> > >
> > > We hope that the above clarification fully addresses the reviewer’s concern. Once again, we sincerely thank the reviewer for the thoughtful comments and valuable feedback, which have greatly improved the quality and clarity of this work.

---

> > > > ### Comment · Reviewer_UaYb · 2025-11-25
> > > >
> > > > Thank you for the detailed clarification. I understand now.
> > > >
> > > > No need to apologize at all; you’ve already done an excellent job in addressing the questions and refining the explanation :)

---

### Official Review · Reviewer_u4yu · 2025-10-31

**Soundness:** 2
**Presentation:** 3
**Contribution:** 3
**Rating:** 4
**Confidence:** 2

**Summary:**

The paper proposes BOTS, a Bayesian framework for online task selection in reinforcement finetuning (RFT). It maintains posterior estimates of task difficulty and uses Thompson sampling to select tasks near a target difficulty (in practice, $p^*=0.5$).  BOTS aims to balance data efficiency and adaptability by combining explicit evidence from direct rollouts and implicit evidence inferred through an interpolation-based plug-in. Experiments on the GURU benchmark with Qwen2.5 models demonstrate improved sample efficiency compared to uniform random and other baselines.

**Strengths:**

- Provides a principled Bayesian formulation of online task selection that generalizes prior methods.
- Fuses explicit and implicit evidence to improve stability and cold-start performance.
- Includes ablations on $\lambda$ (forgetting) and $\rho$ (evidence fusion), providing good insight into the method’s behavior.
- The interpolation-based implicit evidence is computationally efficient (<0.2% overall overhead).

**Weaknesses:**

- Experiments are limited to two LLM scales (1.5B and 7B) of Qwen2.5 on a single benchmark (GURU), raising concerns about generalizability.
- The reported improvements over baselines, appear modest and sometimes inconsistent across domains and target fractions and no confidence intervals or statistical tests are provided.
- The method assumes access to reference models for implicit evidence, the practicality of this in new domains is not fully addressed.

**Questions:**

- Were any statistical significance tests (e.g., across seeds) conducted for the reported improvements? Confidence intervals or variance would strengthen claims of robustness.
- Since $\lambda$ and $\rho$ control forgetting and evidence fusion and seem to balance early and long-term performance, have you considered adaptive or schedule-based versions that evolve over training rather than remaining fixed?
- Correct me if I'm wrong, but there seems to be a minor inconsistency in Section 3.4: the paper defines utility as the distance $\| p_k - p^* \|$ from the target difficulty but states that tasks with the highest utility are selected, which appears opposite to the stated goal of favoring tasks difficulty near $p^*$.

---

> ### Author Response · Authors · 2025-11-22
>
> We sincerely thank the reviewer for the constructive feedback! We take your concerns regarding the generalizability of our results and the lack of statistical significance tests very seriously. **Motivated by your comments, we have further broadened the scope and rigor of our empirical study**. Specifically, we have: (1) re-run core experiments with **3 independent seeds to provide 95% confidence intervals**, ensuring the robustness of our reported improvements; and (2) expanded our evaluation to a new model scale (**Qwen2.5-3B**) and a new dataset (**DeepScaleR**). We believe these extensive additions directly address your concerns about generalizability and statistical strength. Below, we provide detailed responses to your raised Weaknesses (W) and Questions (Q) point by point.
>
> ---
>
> ### W1: Regarding the generalizability of the empirical study
>
> > *'Experiments are limited to two LLM scales (1.5B and 7B) of Qwen2.5 on a single benchmark...'*
>
> We have expanded our empirical evaluation to further assess the generalizability of BOTS. Specifically, Appendix J.6 now includes results for Qwen2.5-3B-Instruct on the math domain, and Appendix J.7 reports a comparison between BOTS and the Random baseline on the DeepScaleR dataset [1]. We hope these additional experiments help address the reviewer's concerns regarding the breadth and generalizability of our empirical study.
>
> [1] Deepscaler: Surpassing o1-preview with a 1.5b model by scaling rl
>
> ---
>
> ### W2: Regarding inconsistent improvements across domains and target fractions
>
> > *'The reported improvements... appear modest and sometimes inconsistent... no confidence intervals or statistical tests are provided.'*
>
> We agree that the improvements of BOTS vary across domains and target fractions. However, this variability largely stems from the heterogeneity of datasets and the substantially different learning dynamics across models and tasks. Given these complexities, ***it is extremely challenging for any general-purpose method to consistently and significantly outperform all baselines in every setting***.
>
> Another criterion is whether a method can ***reliably outperform the random baseline*** (i.e., TTB<1 and BSF>1). Under this criterion, BOTS improves training efficiency more consistently than other baselines across domains and target fractions. Considering that BOTS introduces almost no engineering burden (default hyperparameters are used for all settings, without tuning) and incurs negligible computational overhead, we believe BOTS provides a practical and robust general framework for online task selection in RFT.
>
> ---
>
> ### W3: Regarding the practicality of reference models in new domains
>
> > *'... the practicality of this in new domains is not fully addressed.'*
>
> We expand the discussion from Section 3.3 in the original submission in **Section 3.6, paragraph 2**. Overall, our clarification is as follows:
>
> 1. While evaluating reference models requires upfront rollouts, **such evaluations are increasingly routine in modern RL dataset construction (e.g., GURU)**, and serve as reusable difficulty meta-tags;
> 2. Even when such evaluations are not readily available, they constitute a **one-time, amortizable cost**—distinct from oversampling-based online methods that repeatedly incur extra rollouts during training; and
> 3. These evaluations not only support BOTS but also provide additional benefits, such as **enabling offline difficulty-aware filtering** and **supporting RL algorithms requiring difficulty estimation** (e.g., SPO [2]).
>
> [2] Single-stream Policy Optimization, https://arxiv.org/abs/2509.13232

---

> ### Author Response · Authors · 2025-11-22
>
> ### Q1: Regarding statistical significance tests
>
> > *'Were any statistical significance tests conducted for the reported improvements? ...'*
>
> Thanks for your suggestions. We have expended experiments in Section 4.2, 4.3, and 1.5B-math experiments in Section 4.4, each conducted with 3 independent runs using different random seeds. All curve plots now include 95% confidence intervals, and all table entries are computed using the mean over these 3 runs. Figure 2-3, Table 1-3 are updated accordingly. Such a practice is also applied to the new experimental results reported in Appendix I & J.
>
> Overall, the extended results are consistent with the main conclusions reported in the paper, including the effects of $\lambda$ and $\rho$ in BOTS and the performance gains of BOTS over baselines. We hope these additional results alleviate concerns regarding the statistical robustness of our empirical findings.
>
> ---
>
> ### Q2: Regarding adaptive $\lambda$ and $\rho$
>
> > *'... have you considered ($\lambda$ and $\rho$) adaptive or schedule-based versions that evolve over training rather than remaining fixed?'*
>
> We agree that designing adaptive or schedule-based versions of $\lambda$ and $\rho$ is a very interesting direction. As shown in Section 4, $\lambda$ and $\rho$ influence BOTS differently across training stages. This naturally motivates adaptive strategies.
>
> As mentioned in Appendix E.4.2, we indeed view adaptive update rules as a promising extension of BOTS. Early in the project, we experimented with simple schedulers, for example, gradually reducing $\rho$ in later training stages since implicit evidence may become less reliable once sufficient explicit evaluations accumulate. However, the observed gains were marginal, largely because our default setting already places significant weight on explicit evidence, which limits the negative impact of imperfect implicit evidence. To avoid adding unnecessary engineering overhead (e.g., manually designing schedules), we deliberately kept $\lambda$ and $\rho$ fixed in this work.
>
> Nonetheless, we believe adaptive strategies remain a compelling future direction. One intuitive possibility is to ***increase*** $\lambda$ when the estimated success probabilities deviate substantially from online evaluations (i.e., speed up adaptation), or to ***reduce*** $\rho$ when implicit evidence appears less accurate. We plan to explore such mechanisms in future iterations of BOTS.
>
> ---
>
> ### Q3: Regarding the typo in the definition of the utility function
>
> > *'a minor inconsistency in Section 3.4... utility as the distance from the target difficulty but states that tasks with the highest utility are selected...'*
>
> You are absolutely correct. We thank you for carefully spotting this inconsistency.
>
> We have fixed this issue in the revised manuscript by adding a minus sign in the definition of the utility function so that higher utility corresponds to tasks whose estimated success probabilities are ***closer*** to the target difficulty.
>
> ---
>
> ### Closing Remark
>
> Thank you again for your insights and feedback! We kindly hope these responses and the new revision can address your concerns and we would be grateful if you would consider re-evaluating your assessment of our work.

---

> > ### Comment · Reviewer_u4yu · 2025-11-26
> >
> > Thank you for the clarifications and the additional experiments. I believe my concerns have been adequately addressed, and I am therefore raising my score.

---

### Official Review · Reviewer_kBRo · 2025-10-31

**Soundness:** 3
**Presentation:** 3
**Contribution:** 3
**Rating:** 6
**Confidence:** 3

**Summary:**

The paper proposes a unified framework for Bayesian Online Task Selection in LLM reinforcement finetuning, named BOTS. The method jointly incorporates explicit evidence the selected tasks and implicit evidence from unselected tasks, with Thompson sampling to pick tasks which is close to a target difficulty. The method's effectiveness is empirically demonstrated on several benchmarks.

**Strengths:**

1. The paper is well-written and easy to follow.

2. The proposed idea is interesting and novel.

3. The empirical results demonstrates the effectiveness of the proposed method.

**Weaknesses:**

1. The proposed method is only evaluated on binary rewards. It is not clear whether it is applicable on more complex tasks.

2. The comparisons are internal. Lack of comparisons with other alternatives.

3. There are some related work on task selection are missed:

[1] DATS: Difficulty-Aware Task Sampler for Meta-Learning Physics-Informed Neural Networks.

[2] Model predictive task sampling for efficient and robust adaptation

**Questions:**

In table 1, it seems that the proposed method is sensitive to the hyperparameter values and there is no consistent conclusion. Is there any common principle to set the hyperparameters according to the task?

---

> ### Author Response · Authors · 2025-11-22
>
> Thanks for your valuable feedback! We are deeply encouraged by your recognition of our work's **novelty** and **effectiveness**.
>
> Motivated by your insightful comments on generalizability and external comparisons, we have further expanded our empirical evaluation to a new **external baseline (Dynamic Sampling)**. We further revised the manuscript following your suggestions by including the **related works** you mentioned and adding **Practical Consideration** (Section 3.6) to improve the readability. We will address each of your Weaknesses (W) and Question (Q) below.
>
> ---
>
> ### W1: Regarding non-binary rewards
>
> > *'The proposed method is only evaluated on binary rewards... whether it is applicable on more complex tasks.'*
>
> It is true that we evaluate BOTS primarily in the binary-reward setting. We believe this setting is already highly relevant and practically valuable, since the dominant RLVR training tasks today—such as math and code—use correct/incorrect rewards.
>
> Nonetheless, BOTS is not limited to binary rewards. In Appendix E.4.1, we discuss how BOTS naturally extends to more general reward distributions, including Gaussian and Categorical models, by replacing the Beta–Bernoulli update with the corresponding conjugate Bayesian updates. We fully agree that evaluating BOTS on non-binary reward tasks is an important next step, and we would be eager to investigate such settings in future work when suitable benchmarks become available.
>
> ---
>
> ### W2: Regarding external comparison
>
> > *'The comparisons are internal...'*
>
> As a unified framework, BOTS subsumes several advanced task-selection strategies as special cases. Accordingly, a substantial portion of our evaluations focuses on **within-framework comparisons**, which reveal how different evidence sources and update rules behave under a common Bayesian formulation.
>
> For **external baselines**, beyond the easy-to-hard offline curriculum included in the original submission, we additionally compare BOTS against **Dynamic Sampling** [1] and report the results in Appendix J.2. Overall, BOTS outperforms Dynamic Sampling in the studied setting, while requiring **substantially less computation**.
>
> Below we clarify why certain other alternatives could not be included at this stage:
>
> **Original DOTS [2].** We are unable to directly compare BOTS with the original DOTS for two reasons. First, as discussed in Appendix B.6, DOTS incorporates **Rollout Replay**, a mechanism that reuses historical trajectories. This lies outside the scope of our work, which focuses strictly on online task selection without trajectory reuse.
> Second, even for the task-selection component, DOTS cannot be faithfully reproduced: the authors have not released the training data or the trained adaptor weights required for their attention-based task difficulty estimator (see their GitHub issue: https://github.com/ASTRAL-Group/data-efficient-llm-rl/issues). We plan to evaluate BOTS with DOTS-style implicit evidence once the adaptor becomes reproducible.
>
> **SPEED-RL [3].** We do not compare BOTS with SPEED-RL because the public arXiv manuscript (https://arxiv.org/pdf/2506.09016) currently includes an author-issued notice:
> **“NOTE (July 9, 2025): This preprint contains an issue that we are currently investigating. Results and conclusions may change.”**
> We will consider benchmarking BOTS against SPEED-RL once the issue is resolved and the method stabilizes.
>
> [1] DAPO: An Open-Source LLM Reinforcement Learning System at Scale.
>
> [2] Improving Data Efficiency for LLM Reinforcement Fine-tuning Through Difficulty-targeted Online Data Selection and Rollout Replay.
>
> [3] SPEED-RL: Faster Training of Reasoning Models via Online Curriculum Learning.
>
> ---
>
> ### W3: Regarding missed related works
>
> > *'related work on task selection are missed'*
>
> We thank the reviewer for pointing out the missing related works. We have added the suggested references in Appendix C.

---

> ### Author Response · Authors · 2025-11-22
>
> ### Q1: Regarding Hyperparameter Sensitivity
>
> > *'... Is there any common principle to set the hyperparameters according to the task?'*
>
> As shown in Sections 4.2 and 4.3, BOTS is not sensitive to the choice of $\lambda$ and $\rho$ as long as they remain near the default value $0.1$. Across all experiments, settings such as $0.05$ and $0.2$ consistently outperform the random baseline, while only extreme values (very close to $0$ or $1$) lead to noticeable degradation.
>
> Moreover, in all scenarios evaluated in Section 4.4, we simply reused the default values $\lambda = 0.1, \rho = 0.1$ without any hyperparameter tuning, yet BOTS still exhibits robust and strong performance. Sections 4.2 and 4.3 also provide detailed analyses of how $\lambda$ and $\rho$ influence task selection dynamics. Based on these findings, we recommend that practitioners begin with the default setting $\lambda = 0.1, \rho = 0.1$ in new applications and optionally adjust them following the insights provided in Section 3.5. We explitly mention the strategy in Section 3.6 in our revision.
>
> ---
>
> ### Closing Remark
>
> Once again, we sincerely thank you for your recognition and invaluable feedback. We hope our responses have thoroughly addressed your concerns regarding experiments' generalizability.
>
> We kindly hope that these clarifications and additions might further strengthen your confidence in our work, your support is immensely encouraging to us. If you have any additional concerns or queries, we warmly invite you to share them with us.

---

> > ### Comment · Reviewer_kBRo · 2025-11-26
> >
> > I thank the authors for the answers, which solved all my concerns. I will keep my original score and suggest the paper is above the acceptance threshold.

---

### Official Review · Reviewer_e5tW · 2025-11-02

**Soundness:** 2
**Presentation:** 3
**Contribution:** 3
**Rating:** 4
**Confidence:** 4

**Summary:**

This paper studies the problem of online task selection in reinforcement finetuning (RFT) for Large Language Models (LLMs). The authors introduce BOTS, a unified Bayesian framework that aims to actively select tasks for each step that are of appropriate difficulty at the current stage. BOTS models task difficulty using a Beta distribution posterior and adaptively updates its beliefs by fusing two sources of information: explicit evidence from direct evaluations of selected tasks and implicit evidence from unselected tasks. The framework includes a lightweight interpolation-based method to generate implicit evidence with negligible overhead. Task selection is performed via Thompson sampling to balance exploration and exploitation. Experiments across various domains and model scales show that BOTS improves data efficiency and model performance compared to random selection.

**Strengths:**

1. The studied online task selection problem is crucial for the LLM reinforcement finetuning problem, and BOTS's formulation of it as a unified Bayesian inference problem is elegant and well-suited for the task.
2. BOTS avoids the need for additional model rollouts during training, and the computational overhead is shown to be negligible ($\le 0.2\%$) while consistently showing performance gains over random selection, making the framework efficient and effective in adoption.
3. The idea of trying to balance explicit and implicit evidence is practical for stabilizing the effective sample size $n_t$, as proved in Proposition 2. And the authors show it is beneficial for cold start in Section 4.2.

**Weaknesses:**

1. Although quite efficient, the linear interpolation function in BOTS may be weak as an instantiation of $\tilde{p}(k, \mathcal{B}_t)$. Results in Figure 5 show that the Pearson correlation between estimated and empirical difficulties is weak, and the ROC AUC is also not good enough (especially for Qwen2.5-1.5B-Instruct). And there is no theoretical analysis to justify why a linear interpolation alone can adequately capture the complex information needed for task difficulty estimation.
2. The training time is not reported in the main results. Although BOTS generates the same number of rollouts and trains for the same number of steps compared to random sampling, the actively selected tasks usually generate longer rollouts that actually need more computational resources. Thus, a fairer comparison should let random sampling generate more rollouts per task to have a similar training time as BOTS.
3. The comparisons to prior work DOTS [1] are done via "within-framework" ablations (BOTS-DOTS) rather than against their original implementations (Section 4.4, Appendix B.6). While this is sufficient for case studies within the BOTS framework, it does not fairly compare the performance between BOTS and DOTS. A comparison between BOTS and the original DOTS is needed. Additionally, more related baselines should be included for comparison to highlight BOTS's advantage (e.g., Dynamic Sampling [2], SPEED [3]).
4. The framework mainly studies two important hyperparameters, $\lambda$ and $\rho$. The ablation studies show that the model's performance is sensitive to these choices (Section 4.2, 4.3). The paper does not provide a clear methodology for setting these hyperparameters beyond empirical sweeps, which could be computationally expensive for new applications. Additionally, the momentum coefficient $\gamma$ is a crucial hyperparameter for interpolation, but the impact of it is not discussed in the paper.
5. It seems BOTS mainly works on math problems; there's no significant performance gain compared to random sampling in other cases, especially on experiments using Qwen2.5-7B on Code and Logic. The generalization is somewhat limited.
6. Minor suggestions: providing an algorithm pseudocode can help readers understand the workflow of BOTS more quickly.

[1] Improving Data Efficiency for LLM Reinforcement Fine-tuning Through Difficulty-targeted Online Data Selection and Rollout Replay.

[2] DAPO: An Open-Source LLM Reinforcement Learning System at Scale.

[3] SPEED-RL: Faster Training of Reasoning Models via Online Curriculum Learning.

**Questions:**

1. This work seems to be an extension of MoPPS [4], with new consideration on the balance of explicit and implicit evidence. But I don't understand why it can additionally capture "cross-task relationship". In my understanding, BOTS follows the same Thompson Sampling strategy as MoPPS, where different tasks are all treated independently. I believe BOTS aims to synchronizingly update tasks' difficulty that are not sampled at the current stage, to get a more precise estimation of these tasks for later sampling. Please correct me if I'm wrong.
2. How are $\bar{p}^k_w$ and $\bar{p}^k_s$ computed? I'm confused according to the description in the main text. From Appendix E.3, it seems that it directly uses precomputed results, then it may have large errors since training details are quite different. Moreover, I think the ablation of reference model choices is necessary and is not adequate for future work. I wonder if better reference model choices can indicate better Pearson correlation and ROC AUC?
3. Why is the result of "Random" not reported in Figure 15/16?

[4] Can prompt difficulty be online predicted for accelerating RL finetuning of reasoning models?

---

> ### Author Response · Authors · 2025-11-22
>
> We sincerely thank the reviewer for recognizing the efficiency and design of BOTS! Your probing questions regarding the robustness of implicit evidence and the need for stronger baselines have driven the improvement of our paper. To thoroughly address these critical points, we undertook a substantial expansion of our empirical evaluation. This includes: (1) implementing and comparing against **Dynamic Sampling** as a new baseline; (2) running core experiments with **3 independent random seeds** to report **95% confidence intervals**, ensuring statistical rigor; and (3) extending the implicit evidence analysis to a **wider range of reference model pairs**, including extrapolation settings. We believe this extensive new evidence further solidifies our conclusions, and directly addresses your weaknesses (W) and Questions(Q) one by one.
>
> ---
>
> ### W1: Regarding the implicit evidence estimation
>
> >  *'... the linear interpolation function in BOTS may be weak as an instantiation... the ROC AUC is also not good enough ... And there is no theoretical analysis to justify why a linear interpolation alone can adequately capture ...'*
>
> We would like to clarify that this issue arises primarily in **extrapolation** settings. To provide additional empirical support for this point, we added new results in Appendix I (Figure 22). The poor predictive quality observed for the {7B & 30B} $\rightarrow$ Qwen1.5B-Instruct configuration is a direct consequence of using two much stronger reference models to estimate the behavior of a significantly weaker training model. When more appropriate preference-model pairs are used—such as {1.5B & 7B} or {1.5B & 30B}—the predictive quality improves substantially, achieving an average Pearson correlation of ~0.6 and an average ROC AUC of ~0.8.
>
> Importantly, we further observe that although the quality of implicit evidence is indeed sensitive to the choice of reference models, **even relatively** poor implicit evidence still leads to significant improvements in training efficiency. As shown in Figure 23 and discussed in Appendix I, BOTS continues to outperform random sampling even under extrapolation-heavy conditions. This demonstrates both the robustness of BOTS and the value of implicit evidence—even when imperfect.
>
> The reviewer further questioned the lack of theoretical justification for the linear interpolation–based task difficulty estimator. We acknowledge that this choice is heuristic rather than derived from first principles. To narrow this gap, we added a theoretical discussion in Appendix E.3 to clarify **when** linear interpolation is exact and **why** it becomes imperfect in practice. In short, perfect interpolation implicitly assumes that all tasks progress at the same learning pace. While this assumption does not strictly hold—different tasks indeed evolve at different rates—there remains strong empirical correlation across tasks: as the model improves on some tasks, it often improves on related ones as well. This shared progression structure provides intuition for why linear interpolation can still be empirically effective.
>
> ---
>
> ### W2: Regarding the extra cost due to longer rollouts
>
> > *'The training time is not reported in the main results ... the actively selected tasks usually generate longer rollouts that actually need more computational resources'*
>
> In practice, we did not observe any meaningful increase in rollout length compared to random task selection. Across three random seeds, the average rollout lengths (with 95% confidence intervals) are:
>
> - **Qwen2.5-1.5B-Instruct on Math:**
>   - BOTS: $734 \pm 45$ vs. Random: $710 \pm 52$.
> - **Qwen2.5-7B on Math:**
>   - BOTS: $926 \pm 23$ vs. andom: $941 \pm 26$.
>
> These differences fall well within natural variance and show no systematic trend of longer rollouts under BOTS. Thus, we conclude that BOTS **does not** introduce noticeable additional computation through rollout length.

---

> ### Author Response · Authors · 2025-11-22
>
> ### W3: Regarding comparison against DOTS, Dynamic Sampling, SPEED-RL
>
> > *'The comparisons to prior work ... rather than against their original implementations... it does not fairly compare the performance between BOTS and DOTS... more related baselines should be included for comparison'*
>
> - `Original DOTS.` We are unable to directly compare BOTS with the original DOTS for two reasons. First, as discussed in Appendix B.6, DOTS incorporates **Rollout Replay**, a mechanism that reuses historical trajectories. This lies outside the scope of our work, which focuses strictly on online task selection without trajectory reuse. Second, even for the task-selection component, DOTS cannot be faithfully reproduced: the authors have not released the training data or the trained adaptor weights required for their attention-based task difficulty estimator (see their GitHub issue: https://github.com/ASTRAL-Group/data-efficient-llm-rl/issues). We plan to evaluate BOTS with DOTS-style implicit evidence once the adaptor becomes reproducible.
>
> - `SPEED-RL.` We do not compare BOTS with SPEED-RL because the public arXiv manuscript (https://arxiv.org/pdf/2506.09016) currently includes an author-issued notice: ***“NOTE (July 9, 2025): This preprint contains an issue that we are currently investigating. Results and conclusions may change.”*** We will consider benchmarking BOTS against SPEED-RL once the issue is resolved and the method stabilizes.
>
> - `Dynamic Sampling.` We fully agree that Dynamic Sampling is an important baseline, and we have expanded our comparative evaluation by adding a dedicated experiment and analysis in the revised manuscript (see Appendix J.1 and Figure 24).
>
> Overall, BOTS outperforms Dynamic Sampling in the studied setting, despite requiring substantially less computation. This additional comparison further highlights BOTS's efficiency advantages.
>
> ---
>
> ### W4: Regarding sensitivity to hyperparameters
>
> > *'The paper does not provide a clear methodology for setting these hyperparameters beyond empirical sweeps... but the impact of $\gamma$ is not discussed in the paper'*
>
> - **Regarding $\lambda$ and $\rho$**. As shown in Sections 4.2 and 4.3, BOTS is not sensitive to the choice of $\lambda$ and $\rho$ as long as they remain near the default value $0.1$. Across all experiments, settings such as $0.05$ and $0.2$ consistently outperform the random baseline, while only extreme values (very close to $0$ or $1$) lead to noticeable degradation. Moreover, in all scenarios evaluated in Section 4.4, we simply reused the default values $\lambda = 0.1, \rho = 0.1$ without any hyperparameter tuning, yet BOTS still exhibits robust and strong performance. Sections 4.2 and 4.3 also provide detailed analyses of how $\lambda$ and $\rho$ influence task selection dynamics. Based on these findings, we recommend that practitioners begin with the default setting $\lambda = 0.1, \rho = 0.1$ in new applications and optionally adjust them following the insights provided in Section 3.5. We explitly mention the strategy in Section 3.6 in our revision.
>
> - **Regarding $\gamma$**. To evaluate the sensitivity of BOTS to the momentum coefficient used in interpolation-based implicit evidence, we conducted additional experiments with different $\gamma$ values. The detailed results and discussion are provided in Appendix J.2. Overall, we observe that interpolation-based difficulty estimation is not sensitive to $\gamma$: the performance differences between $\gamma \in \{0.8, 0.9, 0.95\}$ are minimal. Since the default $\gamma = 0.9$ consistently yields robust estimates and strong downstream performance, we recommend using this default value in practice to avoid unnecessary hyperparameter tuning.
>
> ---
>
> ### W5: Regarding the non-significant performance gain in domains other than math
>
> > *'...there's no significant performance gain compared to random sampling in other cases...'*
>
> As shown in Table 3 and Table 4, BOTS outperforms the random baseline on most metrics (TTB $<1$, BSF $>1$). Several results outside the math domain are particularly noteworthy: for **1.5B + Code**, BOTS achieves a TTB (100%) of $0.77$, indicating a **23%** reduction in the computation required to reach the baseline’s best performance; for **7B + Logic**, BOTS achieves a TTB (100%) of $0.50$, yielding a striking **50%** reduction in required computation. Given that BOTS adds only negligible overhead, we believe these improvements and the consistency across domains are meaningful.
>
> We acknowledge that the gains for 7B models are generally smaller than those for 1.5B models. This is expected: the GURU dataset was preprocessed using difficulty signals to filter out tasks that are too easy or too hard for 7B model, which inherently reduces the headroom for task-selection methods like BOTS. Even under this less favorable setup, BOTS consistently improves over the random baseline, highlighting its generality and robustness across domains and model scales.

---

> ### Author Response · Authors · 2025-11-22
>
> ### W6: Regarding the request for algorithm pseudocod
>
> > *'... providing an algorithm pseudocode...'*
>
> Thanks for your constructive suggestion, we have added pseudocode for both the full BOTS framework (Algorithm 1) and the interpolation-based implicit evidence estimator (Algorithm 2). These have been included in Appendix H.
>
> ---
>
> ### Q1: Regarding the contribution of BOTS and cross-task relationship
>
> > *'... I don't understand why it can additionally capture "cross-task relationship"... '*
>
> Thank you for the accurate summary of BOTS. Your understanding is correct: BOTS can indeed be viewed as (i) extending explicit-only approaches such as MoPPS by additionally incorporating implicit evidence, and symmetrically, (ii) extending implicit-only approaches such as DOTS by adding explicit evidence. The central contribution of BOTS is to demonstrate—both conceptually and empirically—that ***both sources of evidence are important, and to provide a principled Bayesian framework that integrates them in a compatible and tunable manner***.
>
> Regarding your question on cross-task relationships: BOTS ***does not*** introduce cross-task coupling inside the Thompson sampling stage—the sampling policy remains identical to MoPPS and treats tasks independently. Instead, cross-task signals arise in the ***implicit evidence*** module. Our interpolation-based estimator leverages the relationship between tasks’ success probabilities under the current model and the two reference models. Intuitively (and as discussed in Appendix E.3), tasks that are similarly “positioned” between the weak and strong references tend to remain similarly positioned for the current model. Linear interpolation captures this coarse-grained relationship and transfers information from directly evaluated tasks to unevaluated ones. This type of cross-task relationship is thus expressed through the evidence update step, not the selection policy itself.
>
> Importantly, linear interpolation is only one possible instantiation of implicit evidence within BOTS. As stated in Appendix E.4.3, BOTS is designed as an extensible framework: more sophisticated plug-ins capable of capturing richer and more fine-grained cross-task relationships can be incorporated in future work. The interpolation-based estimator is chosen for its extreme efficiency (no extra rollouts) and surprisingly strong performance despite its simplicity.
>
> ---
>
> ### Q2: Regarding the success probabilities of reference models
>
> > *'How are success probabilities computed? ... I wonder if better reference model choices can indicate better Pearson correlation and ROC AUC?'*
>
> Yes, in Section 4 we use the empirical success rates of **Qwen2.5-7B-Instruct** and **Qwen3-30B-A3B** pre-tagged in the GURU dataset. This is appropriate because BOTS only relies on ***rollout-based*** task success statistics of the reference models; these do not depend on our training configuration, and the interpolation-based estimator imposes no assumptions that couple the reference models’ training dynamics with those of the finetuned model. Thus, using these precomputed tags is safe.
>
> To address the reviewer’s concern regarding the choice of reference models, we added new ablations in Appendix I (Figures 22–23). We introduce **Qwen2.5-1.5B-Instruct** as an additional reference model and evaluate two new reference pairs: {1.5B, 7B} and {1.5B, 30B}.
>
> Findings are (i) Better-bracketed reference models yield higher predictive quality (avg. Pearson Correlation $\approx 0.6$, avg. ROC AUC $\approx 0.8$);
> (ii) Even when interpolation is inaccurate (e.g., extrapolating with {7B, 30B} $\rightarrow$ 1.5B), BOTS still significantly improves training efficiency;
> (iii) Across all reference choices, BOTS consistently outperforms the random baseline.
>
> These results confirm that (i) reference model choice affects the quality of implicit evidence, and (ii) BOTS remains robust across choices. We refer the reviewer to Appendix  I for full results and the **Choice of Reference Models** paragraph in Section 3.6 for practical considerations.

---

> ### Author Response · Authors · 2025-11-22
>
> ### Q3: Why is the result of 'Random' not reported in Figure 15/16?
>
> We thank the reviewer for the careful inspection. The omission of the **Random** curve in the legend was caused by a visualization-script bug when parsing log files to generate plot legends. The **Random** baseline was included in the underlying data but its label was not rendered. We have fixed this issue and updated the figures in the revised manuscript accordingly.
>
> ### Closing Remarks
>
> Thank you again for your insights and feedback! We hope these responses and the new revision can address your concerns and enhance your confidence in the acceptance of this paper. If you have any additional concerns or queries, we warmly invite you to share them with us.

---

### Comment · Area_Chair_3F6S · 2025-11-26
**Reminder of the Author-Reviewer discussions**

Hi Authors and Reviewers,

Please remember that the manuscript is able to update. And it would be appreciated if you can attend the author-reviewer discussion. The ddl is Dec. 3rd.

Best,
AC

---

### Author Response · Authors · 2025-12-01
**Rebuttal Status Summary (2/2)**

### 2.2 Writing Improvements

- R1. We added Section 3.5 to provide a clearer and more systematic discussion of the roles of $\lambda$ and $\rho$ in both difficulty estimation and selection-time uncertainty.

- R2. We added Section 3.6 to consolidate practical considerations—hyperparameter choices, computational cost, and reference-model selection—for practitioners using BOTS.

- R3. We included pseudocode for both BOTS and the interpolation-based implicit evidence estimator in Appendix H to increase clarity and reproducibility.

- R4. We updated Appendix E.3 to include new theoretical results that formally discuss the intuition behind linear interpolation and clarify its limitations.

- R5. We made various wording and presentation refinements throughout the paper to improve readability.

- R6. We updated Appendix E.1 with a toy example illustrating the role of $\lambda$ in difficulty estimation.

---

# 3. Reviewer Response Summary

---

### 3.1 Rating/Confidence Changes

(original rating, original confidence) -> (updated rating, updated confidence)

- Reviewer e5tW did not reply to our rebuttal responses (4, 4) -> **(4, 4)**
- Reviewer kBRo acknowledged that all concerns have been solved and kept the original rating (6, 3) -> **(6, 3)**
- Reviewer u4yu acknowledged that all concerns have been adequately addressed (4, 2) -> **(6, 2)**
- Reviewer UaYb acknowledged that the revised paper already meets a solid standard of quality, and further questioned about "how to utilize a 1.5B model as a reference model for training a 1.5B model itself", for which we provided further explanation. (4, 4) -> **(6, 5)**
- Reviewer SYAn acknowledged that the concerns have been addressed and is inclined to recommend acceptance. (4, 3) -> **(6, 3)**

Reviewer kBRo, u4yu, UaYb, SYAn **explicitly mentioned their decisions to keep/increase the rating** in their responses.

---

### 3.2 Timeline (UTC)

**\[Nov 22, 16:30 ~ 16:43\]** We posted our rebuttal responses to all reviewers.

**\[Nov 23, 0:17\]** **Reviewer UaYb** replied to our rebuttal responses with
> *Thank you very much for the detailed and thoughtful responses... **I believe the paper is now moving in the right direction, and I am raising my score to 6**... **That said, in my view, the current version of the paper already meets a solid standard of quality**.*

**\[Nov 23, 3:01\]** **Reviewer SYAn** replied to our rebuttal responses with
> *Thank you for the detailed response. **My concerns have been addressed, and I will raise my scores accordingly**. Overall I find the proposed unified, principled, and lightweight Bayesian framework for online task selection compelling, and I appreciate the thoughtful discussion supporting it. **I am inclined to recommend acceptance**.*

**\[Nov 25, 9:54\]** We replied to **Reviewer UaYb** to provide furrher explanation on "how to utilize a 1.5B model as a reference model for training a 1.5B model itself".

**\[Nov 25, 19:28\]** **Reviewer UaYb** replied to our further explanation with
> ***Thank you for the detailed clarification. I understand now**.
No need to apologize at all; you’ve already done an excellent job in addressing the questions and refining the explanation :)*

**\[Nov 26 13:15\]** **Reviewer kBRo** replied to our rebuttal responses with
> *I thank the authors for the answers, **which solved all my concerns**. **I will keep my original score and suggest the paper is above the acceptance threshold**.*

**\[Nov 26 13:20\]** **Reviewer u4yu** replied to our rebuttal responses with
> *Thank you for the clarifications and the additional experiments. **I believe my concerns have been adequately addressed, and I am therefore raising my score**.*

---

If you have any further questions or suggestions regarding BOTS, we would be very happy to continue the discussion before the discussion period ends. Thank you again for your time and effort.

---

### Author Response · Authors · 2025-12-01
**Rebuttal Status Summary (1/2)**

Dear AC, SAC, and PCs,

Thank you very much for taking the time to handle the review process for our paper. We fully understand that the current situation significantly increases your workload. To assist you as much as we can, we have prepared a concise summary of the rebuttal status as of **Nov 26, 14:00 UTC**. Our summary consists of three parts:
1. **Review Summary**: We consolidate all reviewer comments, grouping identical or closely related ones, and annotate each concern with the corresponding reviewer and item (R(eviewer)xxxx-W(eakness)#y/Q(uestion)#z).
2. **Revision Summary**: We summarize all added experiments and revisions (E(xperiment)#a, R(evision)#b).
3. **Reviewer Response Summary**: We summarize reviewers’ follow-up responses to our rebuttal, including **rating/confidence updates**.

We hope this summary is helpful for your assessment.

---

# 1. Review Summary

---

### 1.1 Common/Major Comments

Comments raised by multiple reviewers or those we believe are most important for revision. For these, we **added new experiments or made revisions** (summarized in the next section, referenced as \[E#, R#\]) and responsed to reviewers accordingly.

We highlight **Re5tW** because all **OTHER** reviewers have **explicitly acknowledged that their concerns have been addressed**.

- C1. Regarding the quality of implicit evidence and the choice of reference models (**Re5tW**-W1, **Re5tW**-Q2, RUaYb Q1(1)(2), RSYAn-Q2Q4)\[We addressed them in E1, R2, R4\]
- C2. Regarding hyperparameter sensitivity and selection strategy, and the corresponding explanations (**Re5tW**-W4, RkBRo-Q1, RUaYb W1, RSYAn-W2(2))\[E3, R1, R2\]
- C3. Regarding the cross-domain/scale/dataset/benchmark generalization of BOTS (**Re5tW**-W5, Ru4yu-W1, RUaYb W3)\[E3\]
- C4. Regarding comparison to other out-of-framework baselines (**Re5tW**-W3, RkBRo-W2)\[E3\]
- C5. Regarding statistical consistency (Ru4yu-W2Q1, RUaYb Q5)\[E2\]
- C6. Regarding the practicality and the cost of reference model evaluation (Ru4yu-W3, RUaYb Q1(4), RSYAn-W4)\[R2\]
- C7. Suggestions on writing, including term usage (RUaYb W2, RSYAn-W3)\[R5\], pseudo code (RUaYb Q4)\[R3\], toy example illustrating the impact of $\lambda$(RSYAn-W2(3))\[R6\], separated discussion of uncertainty estimation and success‐rate estimation (RSYAn-W3)\[R1\]

---

### 1.2 Clarification

Some points raised by reviewers involved questions or requested clarifications. We **responded to each of them point-by-point** and made corresponding refinements to the relevant parts of the manuscript.

- C8. Regarding the contribution of BOTS over previous works (RSYAn-W1)
- C9. Extension of BOTS, including non-binary scenarios (RkBRo-W1), adaptive hyperparameters (Ru4yu-Q2), potential collapse mode and solution (RUaYb Q1(3))
- C10. Questions on conception and algorithmic details, including the rollout length change and potential cost (**Re5tW**-W2), cross-task relationship (**Re5tW**-Q1), reference evaluation collection (**Re5tW**-Q2), explanation for empirical results with large $\rho$ (RUaYb Q2), BSF metric (RUaYb Q3), prior setting (RSYAn-W2(1)Q3), explanation for empirical results with various $\lambda$ (RSYAn-Q1)

---

### 1.3 Minor

Typos and related works. We **revised the paper accordingly**.

- C11. Missed related works (RkBRo-W3)
- C12. typo in text and plot (**Re5tW**-Q3, Ru4yu-Q3, RUaYb W1, RSYAn-W2(2))

---

# 2. Revision Summary

---

### 2.1 Experimental Enhancements

- E1. We added new experiments in Appendix I to study different combinations of reference models and their effects on (i) the quality of implicit evidence and (ii) the resulting performance of BOTS. Our findings show that while avoiding extrapolation improves the quality of implicit evidence and in turn improves BOTS a bit, BOTS remains robust and consistently outperforms baselines even when implicit evidence is imperfect.

- E2. For the 1.5B model on the Math domain (Sections 4.2, 4.3, and 4.4), we reran all experiments with three independent random seeds and updated all relevant plots, both in the main text and in Appendix J.3–J.5, to include mean curves with 95% confidence intervals, improving statistical reliability.

- E3. We added several additional experiments in Appendix J, including:
  - Dynamic Sampling as a new external baseline (J.1),
  - the impact of the momentum coefficient $\gamma$ in interpolation-based difficulty estimation (J.2),
  - results using Qwen2.5-3B-Instruct as a new base model (J.6), and
  - results on the DeepScaler dataset (J.7).

(To be continued)

---

### Meta-Review · Area_Chair_MrzZ · 2026-01-15

**Summary:**

This paper proposes BOTS, a unified Bayesian online task-selection framework for LLM reinforcement fine-tuning that maintains per-task difficulty posteriors, fuses explicit rollout evidence with implicit (interpolated) evidence for unevaluated tasks, and uses Thompson sampling to select tasks near a target difficulty (e.g., 0.5) to improve training efficiency.

Key strengths include a new and clean Bayesian formulation for task selection in LLM reinforcement finetuning, a lightweight implicit-evidence plug-in designed to avoid extra online rollouts, and comprehensive experimental results with ablations and multiple efficiency/performance metrics across domains and model scales.

The main weaknesses raised in review were initially (1) limited generalization/benchmark breadth and modest or inconsistent gains, (2) missing uncertainty/robustness reporting, (3) confusing or inconsistent representations, terminology, and hyperparameters, and (4) concerns about the simplicity/quality of the linear interpolation and the need for stronger external baselines and fair cost comparisons.

The author made significant efforts during the rebuttal period and added multiple new experiments, and received positive feedback from most reviewers. After revision, the paper presents a clear and principled framework empirically supported by stronger ablations, improved statistical reporting for key results, added baselines, and substantially improved clarity around previously confusing points. While some external comparison and breadth limitations remain, the core contribution is technically coherent and practically motivated. The AC thus recommends the acceptance of this paper.

**Reviewer Concerns:**

Most substantive concerns were addressed in the rebuttal and revision via additional experiments and clarifications: the authors added explicit discussions of the roles of the key hyperparameters (and corrected previously inconsistent explanations), added practical guidance on reference-model selection and the cost tradeoff, provided algorithm pseudocode, and greatly expanded empirical evidence (multi-seed runs with confidence intervals for core settings, additional baselines such as Dynamic Sampling, and added scope via additional model scale/dataset experiments). In addition, presentation issues (terminology, organization, pseudo-code) were acknowledged and corrected in the revised manuscript.

The remaining outstanding issues are concentrated in one non-responding reviewer. The key skepticism about whether linear interpolation is an adequately justified instantiation of implicit evidence is mitigated but not fully eliminated (the authors added discussion and evidence, but it remains a heuristic plug-in rather than a principled estimator). Likewise, while the authors added at least one stronger external baseline (Dynamic Sampling) and explained why some comparisons (e.g., original DOTS, SPEED-RL) are not currently feasible or stable, the broader comparison picture is still not exhaustive, and the gains, though consistent enough for acceptance, are sometimes benchmark-dependent.

**Reviewer Scores:**

The authors have conducted an effective rebuttal. Reviewer u4yu, UaYb and SYAn have acknowledged that their concerns were mostly addressed and their scores are likely to move to 6.

Reviewer e5tW: score 4 (confidence 4) and did not participate further in the discussion. Given that the rebuttal directly targeted this reviewer’s main points (reference-model ablations, added baseline, added discussion of practicality, and additional clarifications), it is plausible that the reviewer would have moved to a borderline score around 5 (or possibly 6).

---

### Decision · Program_Chairs · 2026-01-26

Accept (Poster)